# TCF1 and LEF1 promote B-1a cell homeostasis and regulatory function

Qian Shen[1✉], Hao Wang[1,19], Jonathan A. Roco[1,2,19], Xiangpeng Meng[2,19], Marita Bosticardo[3], Marie Hodges[4], Michael Battaglia[5], Zhi-Ping Feng[6], Benjamin James Talks[7,8], Jason Powell[8], Vijaya Baskar Mahalingam Shanmugiah[9], Julia Chu[10], Najib M. Rahman[11], Alguili Elsheikh[11], Probir Chakravarty[1], Amalie Grenov[1], Max Emmerich[1,12], Ottavia M. Delmonte[3], Alexandra F. Freeman[3], Michael D. Keller[13], Brahim Belaid[14], Ilenia Papa[1], James C. Lee[1], Pablo F. Cañete[15,16], Paula Gonzalez-Figueroa[1,2], Yaoyuan Zhang[2], Hai-Hui Xue[17], Samra Turajlic[1,12,18], Luigi D. Notarangelo[3], Muzlifah Haniffa[9], Lee Ann Garrett-Sinha[5], Helen M. Parry[4], Nikolaos I. Kanellakis[10] & Carola G. Vinuesa[1✉]

B-1 cells are innate-like immune cells abundant in serosal cavities with antibodies enriched in bacterial recognition, yet their existence in humans has been controversial[1–3]. The CD5+ B-1a subset expresses anti-inflammatory molecules including IL-10, PDL1 and CTLA4 and can be immunoregulatory[4–6]. Unlike conventional B cells that are continuously replenished, B-1a cells are produced early in life and maintained through self-renewal[7]. Here we show that the transcription factors TCF1 and LEF1 are critical regulators of B-1a cells. LEF1 expression is highest in fetal and bone marrow B-1 progenitors, whereas the levels of TCF1 are higher in splenic and peritoneal B-1 cells than in B-1 progenitors. TCF1–LEF1 double deficient mice have reduced B-1a cells and defective B-1a cell maintenance. These transcription factors promote MYC-dependent metabolic pathways and induce a stem-like population upon activation, partly via IL-10 production. In the absence of TCF1 and LEF1, B-1 cells proliferate excessively and acquire an exhausted phenotype with reduced IL-10 and PDL1 expression. Furthermore, adoptive transfer of B-1 cells lacking TCF1 and LEF1 fails to suppress brain inflammation. These transcription factors are also expressed in human chronic lymphocytic leukaemia B cells and in a B-1-like population that is abundant in pleural fluid and circulation of some patients with pleural infection. Our findings define a TCF1–LEF1-driven transcriptional program that integrates stemness and regulatory function in B-1a cells.

B-1a cell development is instructed by B cell receptor (BCR) signal strength and selection is based on BCR recognition of components of bacterial and senescent red blood cell membranes, as well as apoptotic cells and oxidized lipids[8]. B-1a cells have a restricted BCR repertoire[9] and persist through adulthood due to their capacity to self-renew, a process that requires sustained but limited proliferation to maintain a stable cell pool over time[10]. Because of these properties, B-1a cells have been referred to by some as 'natural memory' cells[11]. B-1a cells in the peritoneal cavity have slow turnover rates compared with B-2 cells[12], and can completely reconstitute host tissues lacking these cells after adoptive transfer, persisting for many months with little contribution from host-derived cells[13]. B-1a cells are potent repressors of autoreactive and inflammatory immune responses and tissue damage, largely due to their ability to produce IL-10 (refs. 14,15). The existence of B-1 cells in humans has, however, been controversial. Here we show that TCF1 and LEF1—known to confer stem-cell like properties in various cell types including CD8+ T cells[16]—are expressed in mouse B-1 cells and human B-1-like cells. In vivo experiments in mice showed that TCF1 and LEF1 promote B-1 maintenance and regulatory ability via enhancing IL-10 production.

[1]Francis Crick Institute, London, UK. [2]Division of Immunology and Infectious Disease, John Curtin School of Medical Research, Australian National University, Canberra, Australian Capital Territory, Australia. [3]Laboratory of Clinical Immunology and Microbiology, National Institute of Allergy and Infectious Disease, National Institutes of Health, Bethesda, MD, USA. [4]School of Infection, Inflammation and Immunology, University of Birmingham, Birmingham, UK. [5]Department of Biochemistry, State University of New York at Buffalo, Buffalo, NY, USA. [6]ANU Bioinformatics Consultancy, John Curtin School of Medical Research, Australian National University, Canberra, Australian Capital Territory, Australia. [7]Biosciences Institute, William Leech Building, Newcastle University, Newcastle Upon Tyne, UK. [8]Translational and Clinical Research Institute, Newcastle University, Newcastle upon Tyne Hospitals NHS Foundation Trust, Newcastle Upon Tyne, UK. [9]Wellcome Sanger Institute, Wellcome Genome Campus, Cambridge, UK. [10]CAMS Oxford Institute, Nuffield Department of Medicine, University of Oxford, Oxford, UK. [11]Oxford Centre for Respiratory Medicine, Churchill Hospital, Oxford University Hospitals NHS Foundation Trust, Oxford, UK. [12]Renal and Skin Unit, The Royal Marsden Hospital, London, UK. [13]Division of Allergy and Immunology, Children's National Hospital, Washington, DC, USA. [14]Department of Medical Immunology, Beni Messous University Hospital Center, Faculty of Pharmacy, University of Algiers, Algiers, Algeria. [15]Frazer Institute, Faculty of Medicine, The University of Queensland, Brisbane, Queensland, Australia. [16]Ian Frazer Centre for Children's Immunotherapy Research, Child Health Research Centre, Faculty of Medicine, The University of Queensland, Brisbane, Queensland, Australia. [17]Centre for Discovery and Innovation, Hackensack University Medical Center, Nutley, NJ, USA. [18]CAPTURE Consortium, https://www.royalmarsden.nhs.uk/capture. [19]These authors contributed equally: Hao Wang, Jonathan A. Roco, Xiangpeng Meng. ✉e-mail: qian.shen@crick.ac.uk; carola.vinuesa@crick.ac.uk

## TCF1–LEF1 are expressed in B-1 cells

To understand the transcriptional program governing B-1 cell homeostasis, we performed single-cell RNA sequencing (scRNA-seq) on sorted peritoneal CD19[+] cells from adult mice. The determinants of the two major subclusters after removing immunoglobulin genes were the B-2 gene *Fcer2a* (encoding CD23) and the B-1a gene *Cd5*. *Tcf7* (encoding TCF1) was expressed in the *Cd5* subcluster, together with *Bhlhe41*, a regulator of B-1a cell development[17] (Fig. 1a and Extended Data Fig. 1a). ImmGen also revealed high *Tcf7* expression in B-1 cells (Extended Data Fig. 1b). B-1 cells in both the peritoneal cavity and the spleen identified as CD19[high]B220[low] (refs. 17,18) expressed TCF1 by flow cytometry (Extended Data Fig. 1c–e). LEF1 is a transcription factor that interacts with TCF1 (ref. 19), sharing overlapping yet not redundant functions in many biological processes[20]. Although peritoneal cavity B-1a (PC B-1a) cells expressed low amounts of *Lef1* mRNA by scRNA-seq, flow cytometric analysis revealed that the LEF1 protein shared the same pattern of expression as TCF1 (Fig. 1b,c and Extended Data Fig. 1c–e). *Lef1* was also expressed in splenic T3 B cells, and both transcription factors were also highly expressed in memory B cells according to Immgen (Extended Data Fig. 1b). B-1a cells are enriched in phosphatidylcholine (PtC) reactivity[1]. We found that peritoneal B-1a cells expressing the lowest amount of B220 contained more PtC-binding B cells and expressed the highest amounts of TCF1 and LEF1 (Fig. 1d). TCF1 and LEF1 were also detected in bone marrow pro-B cells that give rise to both B-1 and B-2 cell progenitors, as well as B-1 progenitor (B-1P) cells[21] (Extended Data Fig. 1f,g).

In humans, rare circulating B cells bearing B-1 markers have been thought to be pre-plasmablasts[22]. A recent study sequencing human fetal tissues has identified a B-1-like cell subset expressing CD5 and CD43 (ref. 23). In search for a human adult B-1-like population that might co-express TCF1 and LEF1, we stained cells from pleural effusions drained from patients with bacterial pleural infection. Paired CTV-labelled peripheral blood mononuclear cells and unlabelled pleural fluid cells from the same donor were stained in the same tube, and gatings were applied to exclude CD38[+] plasmablasts (Fig. 1e,f). We found a population bearing the B-1 markers CD43 and CD5 and co-expressing TCF and LEF1 that constituted over 80% of all B cells in pleural fluid and up to 60% of peripheral blood B cells in some patients (Fig. 1e–g and Extended Data Fig. 1h,i). Despite this B-1-like population being rare in blood from healthy donors (Fig. 1f,h), it was still enriched in PtC reactivity and expressed higher levels of TCF1 and LEF1 than CD43[−]CD5[+] cells and the bulk of CD43[−]CD5[−] cells (Fig. 1h and Extended Data Fig. 1j). Expression of CD27 and CD24 was heterogenous in human CD43[+]CD5[+] B-1-like cells (Extended Data Fig. 1j). Although the majority of CD43[+]CD5[+] cells were IgM[+], up to 6% expressed IgA (Extended Data Fig. 1j). Of note, PtC-reactive cells were also found in the CD38[+] population (Extended Data Fig. 1k). A CD43[+]CD5[+] phenotype also characterized the neoplastic chronic lymphocytic leukaemia (CLL) B cells[24], which also co-expressed LEF1, a diagnostic marker for CLL[25] and/or TCF1 (Fig. 1e,i). Together, these data show that human CD38[−] B cells that co-express both CD43 and CD5 express higher levels of LEF1 and TCF1 than other mature B cell subsets and are enriched in PtC reactivity, thus displaying a phenotype that resembles mouse B-1 cells.

## TCF1–LEF1 promote B-1a formation

To determine whether TCF1 and LEF1 have a role in B-1 cell homeostasis, we generated mice lacking these transcription factors only in B cells by crossing *Tcf7*-floxed and *Lef1*-floxed mice to mice expressing Cre under the control of the Mb1 (*Cd79a*) promoter (Cre[Mb1])[26]. Although B-2 and B-1b cells were comparable in all groups, mice double deficient in TCF1 and LEF1 (TCF1[Δ]LEF1[Δ]) had a 45% reduction in total peritoneal B-1 cells, a 71% reduction in peritoneal B-1a cells and a 67% reduction in splenic B-1 cells compared with Cre[Mb1]-expressing littermate controls

(TCF1[WT]LEF1[WT]; Fig. 1j–m and Extended Data Fig. 2a,b). Deficiency of TCF1 alone only decreased splenic B-1 cells. TCF1–LEF1 deficiency reduced CD5 expression in B-1a cells but these remained identifiable (Extended Data Fig. 2c). To establish whether TCF1 and LEF1 influence homeostasis of both fetal liver-derived and bone marrow-derived B-1a cells, we reconstituted sublethally irradiated *Rag1*[−/−] mice with either embryonic day 14.5 fetal liver cells or adult bone marrow cells from mice sufficient or deficient in TCF1 and/or LEF1. B-1a cells were decreased by 70% and 60% in mice receiving TCF1[Δ]LEF1[Δ] fetal liver and bone marrow cells, respectively, compared with recipients of wild-type cells (Fig. 1n and Extended Data Fig. 2d,e).

B-1P cells identified as Lin[−]CD93[+]IgM[−]CD19[+]B220[neg-low] peak during late gestation and differentiate into both B-1a and B-1b cells or mature in the spleen through a transitional (TrB-1a) cell stage[27] (Extended Data Fig. 3a). TCF1 and LEF1 deficiency resulted in an increased frequency of B-1P cells in embryonic day 18.5 fetal liver and neonatal (days 1, 3 and 9) bone marrow (Extended Data Fig. 3b,c), yet splenic transitional B-1a cells (TrB-1a; CD93[+]IgM[+]CD19[+]B220[low]CD5[+]) known to exclusively generate B-1a cells[27], as well as their likely counterparts in the bone marrow[28]–B-1Ps (BM B-1Ps)–appeared significantly reduced (Extended Data Fig. 3d,e). Although this suggests that TCF1–LEF1 are involved in B-1a cell development, the decrease in CD5 expression observed in the absence of these transcription factors may underestimate the number of progenitors. Together, our results indicate that TCF1 and LEF1 are required for the formation of a replete peripheral mature B-1a cell pool.

## TCF1–LEF1 maintain B-1a cells

TCF1 and LEF1 are important for proliferation and self-renewal of memory CD8[+] T cells[29,30] and stem cells[29]. A median of 90% of TrB-1a cells were Ki-67 positive–a marker of proliferation–compared with 77% of TrB cells (Extended Data Fig. 3f,g). Both the proliferative rate and the expression of TCF1 and LEF1 were significantly higher in young (4 weeks of age) than adult (10–16 weeks of age) mice (Fig. 2a,b and Extended Data Fig. 3h). Peritoneal B-1a cells lacking TCF1 and LEF1 failed to expand in adult life compared with the steady increase seen in littermate controls (Fig. 2c). To formally evaluate self-renewal, we adoptively transferred peritoneal cells from TCF1[WT]LEF1[WT] or TCF1[Δ]LEF1[Δ] (CD45.2) donors into congenic CD45.1 recipients, who then received 5-bromodeoxyuridine (BrdU) in the drinking water for 12 days. Deficiency in TCF1 and LEF1 led to a greater reduction in the proportion of donor-derived BrdU-labelled B-1a cells ($P = 0.0210$) than B-2 cells ($P = 0.0948$; Fig. 2d). In a complementary approach, we adoptively transferred 50:50 mixes of CD45.2 TCF1[Δ]LEF1[Δ]:CD45.1 TCF1[WT]LEF1[WT]-sorted or control CD45.2 TCF1[WT]LEF1[WT]:CD45.1 TCF1[WT]LEF1[WT]-sorted peritoneal B-1a cells into unirradiated *Rag1*[−/−] mice that cannot repopulate the B-1a cell pool. A smaller fraction of donor-derived CD45.2 B-1a cells was found in both the peritoneal cavity and the spleen when donor B-1a cells lacked TCF1 and LEF1 (Extended Data Fig. 3i).

Consistent with the well-described constitutive BCR signalling in B-1a cells[31], phosphorylated SYK, BTK and BLNK could be readily detected ex vivo in B-1a cells in the absence of exogenous stimulation; this was diminished in TCF1[Δ]LEF1[Δ] B-1a cells (Extended Data Fig. 4a). Downregulation of κ-light and λ-light chains that characterizes B-1a cells was also increased in TCF1[Δ]LEF1[Δ] B-1a cells (Extended Data Fig. 4b). CD19 remained unchanged in both fetal liver B-1P (FL B1P) and PC B1a cells lacking TCF1 and LEF1 (Extended Data Fig. 4c). BCR scRNA-seq of peritoneal CD19[+] cells from adult TCF1–LEF1 sufficient versus deficient mice revealed three highly expanded clonal clusters (Fig. 2e) expressing B-1a-associated immunoglobulin V genes composed of heavy-chain and light-chain pairs *Ighv11-2/Igkv14-126* (c1), *Ighv12-3/Igkv4-91* (c2) and *Ighv9-3/Iglv2* (c3) in mice from both genotypes, albeit reduced in TCF1–LEF1-deficient mice (Fig. 2f). Although the BCR repertoire of the total peritoneal CD19[+] B cell population from TCF1–LEF1-deficient mice appeared more diverse (Extended Data Fig. 5a, top), this was

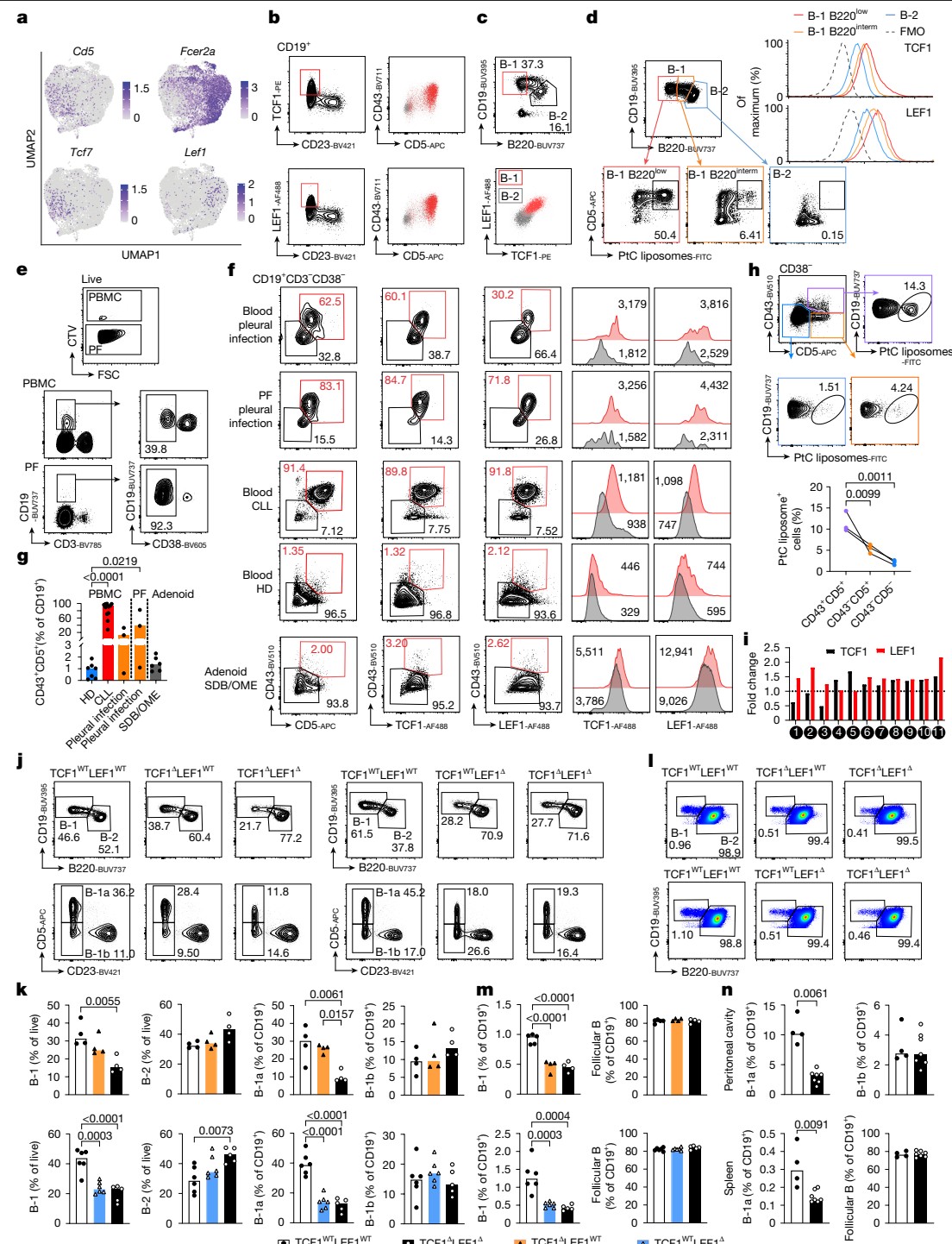

**Fig. 1 | TCF1 and LEF1 are highly expressed in B-1 cells and B-1a cells are reduced in TCF1–LEF1-deficient mice. a**, Gene expression in the different clusters projected on uniform manifold approximation and projection (UMAP) of scRNA-seq from total peritoneal B cells. **b,c**, Flow cytometry plots showing TCF1 or LEF1 expression (**b**) or both (**c**) in peritoneal cavity B cells. **d**, Flow cytometry plots showing gating strategy of B220[low], B220[intermediate] and B220[high] cells from peritoneal cavity B cells, CD5 and PtC expression and histogram of TCF1 and LEF1. FMO, fluorescence minus one. **e–g**, Flow cytometry plots showing CD38[+] and CD38[–] B cells, CD43, CD5, TCF1 and LEF1 expression from the indicated compartments (**e,f**) and quantification of the percentage of CD43[+]CD5[+] cells within the CD19[+] cells in healthy donor (HD; n = 7), CLL (n = 11), pleural infection (n = 3) and sleep disordered breathing/otitis media with effusion ((SDB/OME); n = 6; **g**). CTV, CellTrace Violet; FSC, forward scatter; PBMC, peripheral blood mononuclear cell; PF, pleural fluid. **h**, Contour plots (top) and quantification (bottom) showing PtC reactivity in the indicated

subsets from freshly processed PBMC of HD (n = 3). **i**, Fold change of expression of TCF1 and LEF1 between CD43[+]CD5[+] cells and CD43[–]CD5[–] cells in patients with CLL. **j–m**, Contour plots and quantification of the percentage of B-1, B-2, B-1a and B-1b cells from the peritoneal cavity (**j,k**); B-1 and follicular B cells from the spleen (**l,m**) from TCF1[WT]LEF1[WT] (n = 4 or 6), TCF1[Δ] (n = 4), LEF1[Δ] (n = 6) or TCF1[Δ]LEF1[Δ] (n = 4 or 5). **n**, Frequency of the indicated peritoneal and splenic B cell populations in sublethally irradiated *Rag1*[−/−] recipient mice 6 weeks after reconstitution with fetal embryonic day 14.5 liver cells from mice of TCF1[WT]LEF1[WT] (n = 4) or TCF1[Δ]LEF1[Δ] (n = 7). Each symbol represents an individual mouse and bars represent median values. Data are representative of n = 5 (**e**), n = 3 (**k–m**) and n = 4 (**n**) experiments. Data are from n = 5 (**g**) experiments and n = 3 donors (**h**). Statistical analysis was performed using one-way analysis of variance (ANOVA) with Tukey multiple-comparison test (**g,k,m**) and two-tailed Mann–Whitney *U*-test (**n**) or two-way ANOVA (**h**). The exact *P* values are shown.

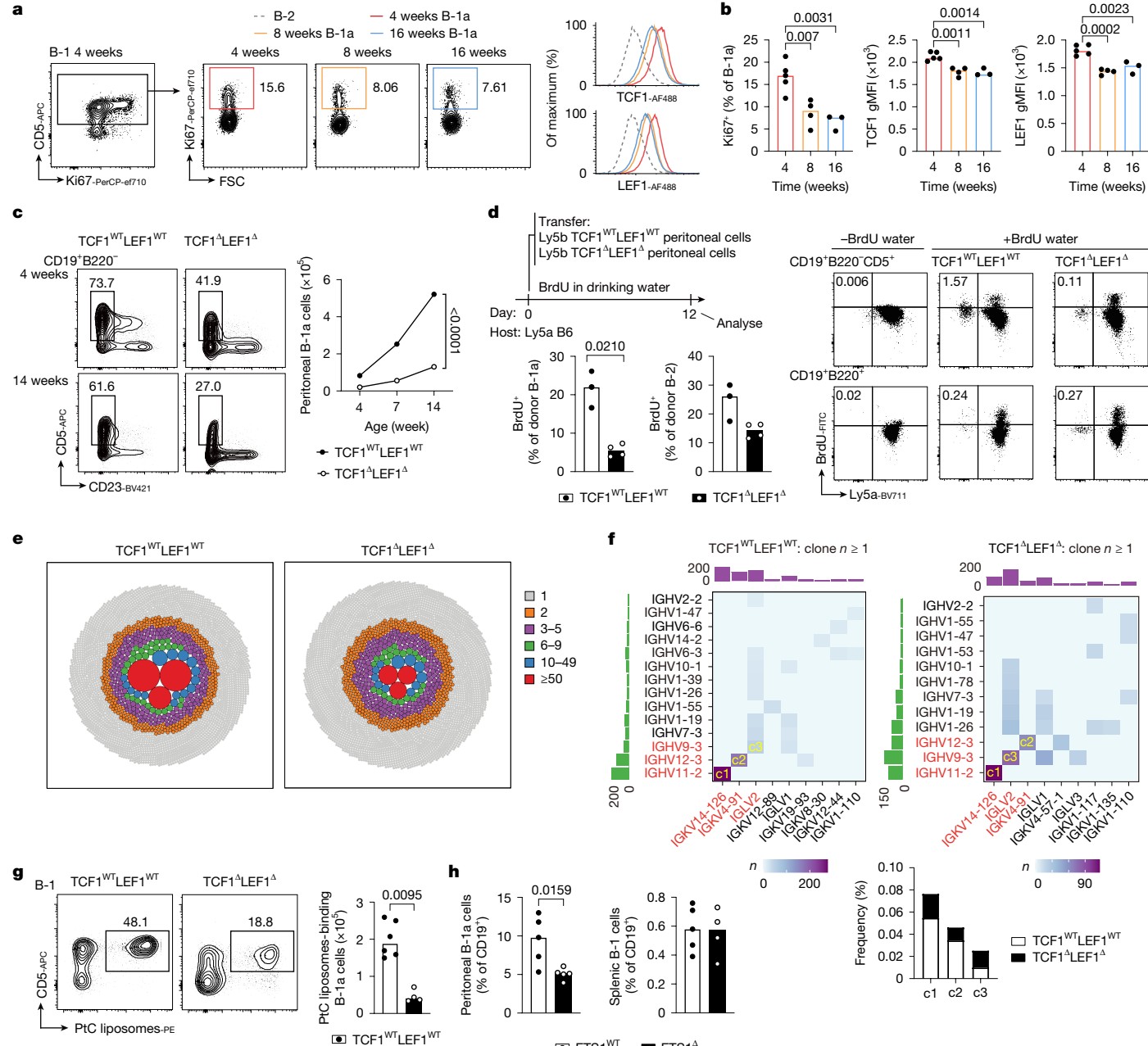

**Fig. 2 | TCF1 and LEF1 are required for B-1a self-renewal. a,b,** Contour plots (**a**) and quantification (**b**) of positive cells and TCF1 or LEF1 expression in peritoneal B-1a cells from 4 (*n* = 5), 8 (*n* = 4) and 16 (*n* = 3) weeks of age mice. **c**, Contour plots showing B-1a cells from TCF1^WT^LEF1^WT^ and TCF1^Δ^LEF1^Δ^ mice at 4 (top) and 14 (bottom) weeks of age, gated on peritoneal CD19^+^B220^−^ cells (left) and quantification (right). **d**, Experimental diagram (top left), dot plots (right) and quantification (bottom left) of BrdU-labelled donor-derived B-1a or B-2 cells: TCF1^WT^LEF1^WT^ (*n* = 3) and TCF1^Δ^LEF1^Δ^ (*n* = 4). **e,f,** Circular plots show cell numbers of the different clonotypes (**e**) with corresponding heatmap (number of clones) and quantification of peritoneal B cell clones with a particular V_H_–V_L_ pairing from TCF1^WT^LEF1^WT^ and TCF1^Δ^LEF1^Δ^ mice. Clones composed of

*Ighv11-2/Igkv14-126*, *Ighv12-3/Igkv4-91* and *Ighv9-3/Iglv2* were labelled as c1, c2 and c3, respectively (**f**, top) and their frequency quantified (**f**, bottom). **g**, Contour plots (left) and quantification (right) of PtC liposome-binding CD5^+^ PC B-1 cell numbers from TCF1^WT^LEF1^WT^ (*n* = 6) and TCF1^Δ^LEF1^Δ^ (*n* = 4) mice. **h**, Quantification of the percentage of peritoneal and splenic B-1 cells from ETS1^WT^ (*Ets1^+/+^*.Cre^Cd19^; *n* = 5) and ETS1^Δ^ (*Ets1^flox/flox^*.Cre^Cd19^; *n* = 4) mice. Each symbol represents an individual mouse, and the bars indicate median values. Results are representative of *n* = 2 (**a**–**d**,**g**,**h**) experiments. Statistical analysis was performed using two-tailed Welch's *t*-test (**d**), Mann–Whitney *U*-test (**g**,**h**), one-way ANOVA with Tukey multiple-comparison test (**b**) or two-way ANOVA (**c**). The exact *P* values are shown.

probably only due to the overall decrease in the more clonal B-1a cells and overrepresentation of sequenced B-1b and B-2 cells. By contrast, the B-1 cell BCR repertoire of TCF1–LEF1-deficient peritoneal cavity B cells did appear less diverse than that of wild-type counterparts (Extended Data Fig. 5a, bottom, and 5c) despite reduced CD5 expression, which has been associated with clonal dominance[32]. Consistent with a decrease in the absolute numbers of B-1a cells, the number of

PtC-binding peritoneal B cells was decreased in TCF1–LEF1-deficient mice (Fig. 2g), while still representing approximately 42% of B-1a cells, comparable with wild-type littermates (Extended Data Fig. 5c).

We did not detect decreased serum antibodies in mice lacking TCF1 and LEF1 (Extended Data Fig. 5d), suggesting compensation by B-1 cell-derived plasma cells[33]. Indeed, B-1 plasma cells were comparable with those of wild-type mice when quantified as a fraction of total

splenic cells, but B-1 plasmablasts were increased when quantified as a fraction of splenic B-1 cells (Extended Data Fig. 5e). An approximately sevenfold increase in IgG3+ B-1a cells was seen in mice lacking TCF1–LEF1 (Extended Data Fig. 5f).

## TCF1–LEF1 upregulate *Ets1* and MYC targets

To obtain insights into signalling pathways downstream of TCF1–LEF1 in B-1a cells, we performed RNA-seq on purified TCF1$^{WT}$LEF1$^{WT}$ and TCF1$^{\Delta}$LEF1$^{\Delta}$ B-1a cells, and expression of B-1a signature genes was superimposed for comparison[34]. B-1a cell identity was largely maintained within the residual B-1a subset in the absence of TCF1–LEF1 (Extended Data Fig. 6a). Transcripts encoding the transcriptional factors *Ets1* and *Irf4* were downregulated in single and double TCF1–LEF1-deficient B-1a cells (Extended Data Fig. 6b). Given a previous report showing the B cell intrinsic roles of ETS1 on B cell activation and differentiation[35], we investigated whether this transcription factor regulates B-1a cell numbers and found that B-1a cells were reduced in the peritoneal cavity, but not the spleen, of adult mice lacking ETS1 in B cells (Fig. 2h); albeit, this was not evident in young mice (data not shown), suggesting that ETS1 may act downstream of TCF1–LEF1 to promote maintenance of B-1a cells with age.

Gene set enrichment analysis revealed that TCF1–LEF1-dependent pathways were linked to cell cycle, including G2–M checkpoint (*Cdk1* and *Ccnb2*) and E2F targets (*E2f8* and *Cenpm*; Extended Data Fig. 6c,d). The hallmark 'MYC targets' pathway was downregulated in TCF1$^{\Delta}$LEF1$^{\Delta}$ B-1a cells (Fig. 3a). There was also a strong and significant correlation between the expression of MYC—a master regulator of metabolism—as well as its target gene *Bcl2* (ref. 36), and expression of both TCF1 and LEF1 (Fig. 3b). B-1 cells depend on glycolysis, acquisition of exogenous fatty acids and autophagy for self-renewal[37]. Consistent with this, inhibition of fatty acid metabolism, oxidative phosphorylation and glycolysis pathways were also prominent pathways in the residual TCF1$^{\Delta}$LEF1$^{\Delta}$ B-1a cells (Fig. 3a).

## TCF1–LEF1 expression in B-1 development

To delineate unique versus overlapping or shared roles of TCF1 and LEF1 in B-1a cell maturation, we generated scRNA-seq data from four distinct B-1 developmental stages, individually enriched, barcoded and then pooled: (1) FL B1Ps, (2) adult bone marrow B-1 progenitors (BM B1Ps), (3) PC B1 cells, and (4) spleen B-1 cells (SP B1s; Fig. 3c). Trajectory analysis revealed a continuous, well-populated path with FL B1Ps at one terminus of the trajectory, SP B1 and PC B1 clusters positioned close to the other end, and BM B1Ps dispersed between FL B1Ps and SP B1s or PC B1s (Fig. 3d). A small number of SP B1 and PC B1 cells located within the FL B1P cluster (Fig. 3e), supporting the existence of splenic B-1a precursors[38]. *Lef1* was expressed strongly in B-1Ps from both fetal liver and bone marrow and maintained expression, albeit at low levels, in peripheral B-1 cells. *Tcf7* by contrast was low at the progenitor stages, and expression increased in peripheral splenic and peritoneal cavity B-1 cells (Fig. 3f). Molecules associated with B-1 cell development including *Lin28b*[39] and *Arid3a*[40] as well as *Il7r* were highly expressed in fetal B-1Ps, whereas *Bhlhe41* was expressed in BM B1Ps, and more strongly in SP B1 and PC B1 cells. The latter also expressed *Ctla4* (ref. 4) and *Bmi1* (ref. 41) (Fig. 3f).

Interrogation of pathways revealed that IL-6–STAT3, TGFβ and TNF signalling were dysregulated in LEF1-deficient B-1a cells, whereas TCF1 deficiency predominantly affected cell cycle via regulation of E2F targets and G2–M checkpoints. By contrast, deficiency of either transcription factor caused changes in the IL-2–STAT5 signalling pathway (Fig. 3g). Reanalysis of a scRNA-seq atlas of human prenatal tissues[23] revealed that most cells expressing the mouse B-1a marker genes *CD5* and *SPN* (encoding CD43) co-expressed TCF1 and LEF1 alone or in combination. Cells co-expressing both TCF1 and LEF1 also expressed the highest amounts of *CD5*, *SPN*, *IL2RG* and *IL7RA* (Fig. 3h and Extended Data Fig. 6e). As seen in mice (Extended Data Fig. 1g), TCF1 and LEF1 were also expressed in human progenitor B cells, particularly in ProB cells, as well as in cycling B cells, as indicated by *Mki67* expression[23] (Extended Data Fig. 6f).

Given that IL-2 signalling is a TCF1–LEF1-dependent hallmark pathway in mouse B-1a cells (Fig. 3g) and that LEF1–TCF1 co-expressing human B-1 cells express high levels of IL-2Rγ and IL-7RA (Fig. 3h), we asked whether these receptors influence B-1 cell development in humans. We evaluated the B-1-like cells in the circulation of patients with paediatric severe combined immunodeficiency (SCID) with genetic loss-of-function variants in *IL2RG* and *IL7RA*, and sufficient events in the CD19+CD38− gate. Patients with IL-2Rγ deficiency had an 80% reduction in B-1-like cells out of total B cells (Extended Data Fig. 6g). Given that the median age of the healthy donors was 24 years compared with 6 years in the patients, and that adult mice have twofold more B-1a cells than young mice, further work is needed to reach definitive conclusions.

## TCF1–LEF1 promote IL-10 production

Next, we set out to investigate the potential targets of TCF1 and LEF1 in B-1a cells. Owing to low cell numbers and low-level expression of these transcription factors in B-1a cells, CUT&Tag was not successful. Analysis of TCF1 peaks from chromatin immunoprecipitation followed by sequencing datasets of mouse thymocytes[42] and comparison with open chromatin regions identified by assay for transposase-accessible chromatin using sequencing (ATAC-seq) in B-1a cells from *Vh12/Vk4* transgenic mice[17] revealed TCF1 peaks at promoters or enhancers of *Cd5*, *Myc* and *Ets1* that also harboured open chromatin regions in B-1a cells, suggesting that they are TCF1 targets (Extended Data Fig. 6h).

B-1a cells have been described to exert regulatory roles via production of IL-10 (ref. 15). In mice, IL-10-producing B cells referred to as 'B10' co-express B220 and CD1d and lack CD23 and CD21 expression[43]. CD21$^{high}$CD1d$^{high}$ marginal-zone B cells[44] and plasma cells can also produce IL-10 (ref. 45). We scored the expression of 'B10-associated genes' and the related 'negative regulation of immune system genes'[46] in our scRNA-seq analysis of peritoneal B cells. Both gene signatures were upregulated in the B-1a c1, c2 and c3 clonal clusters (Figs. 2f and 4a,b), which also co-expressed *Tcf7* and *Cd5*. These signatures included *CTLA4* and *PDL1*, known to be expressed by B-1a cells and to contribute to various immunoregulatory processes, including promoting regulatory T cell responses, dampening inflammatory macrophage activity, limiting T cell-mediated central nervous system (CNS) damage and maintaining B cell tolerance[4–6,47,48]. Indeed, the PDL1 protein was highly expressed in peritoneal B-1 cells compared with conventional B cells (Fig. 4c). B-1 cells with high TCF1–LEF1 expression produced twofold higher IL-10 upon activation than those expressing low amounts of TCF1–LEF1 (Fig. 4d), and B-1a cells lacking TCF1 and LEF1 produced approximately 40% less IL-10 than LEF1–TCF1-sufficient cells (Fig. 4e and Extended Data Fig. 7a). Intravenous lipopolysaccharide (LPS) injection induces IL-10-producing LAG3+CD138+ cells[49]; the number of these cells was also decreased in mice lacking TCF1$^{\Delta}$LEF1$^{\Delta}$ in B cells (Extended Data Fig. 7b).

Among splenic B cells, only CD19$^{low}$ cells exhibiting a CD5+CD1d+ phenotype characteristic of B10 cells[43] produced IL-10 in the steady state, whereas upon LPS stimulation, we observed an increase in IL-10 production among CD19$^{high}$ cells that displayed a B-1a phenotype (B220$^{low}$CD5$^{high}$CD21$^{low}$CD1d$^{low}$) and higher expression of TCF1 and LEF1 (Fig. 4f,g and Extended Data Fig. 7c). Only the latter were significantly reduced in TCF1$^{\Delta}$LEF1$^{\Delta}$ mice (Fig. 4h). There was a strong correlation between the increase in IL-10-producing cells and TCF1 and LEF1 expression. A positive correlation, albeit weak, was also seen with CD5 expression, consistent with the known role for CD5 in promoting IL-10 production[50], and the observed reduction in CD5 expression in B-1a cells lacking TCF1–LEF1 (Fig. 4i,j). As TCF1+ and LEF1+ fetal B cells express minimal

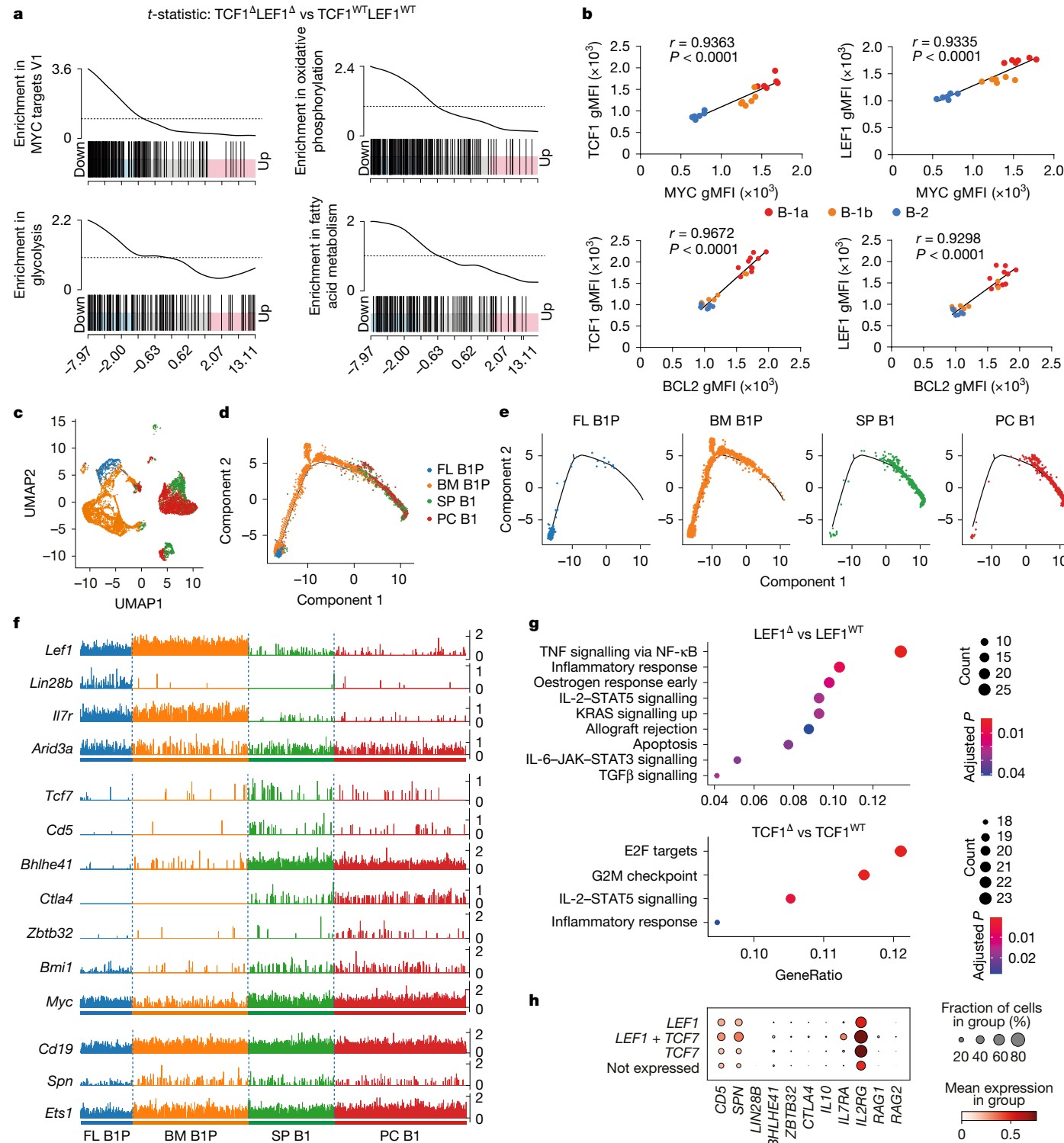

**Fig. 3 | TCF1 and LEF1 in B-1a cell mitosis, development and gene expression.**
**a**, Gene set enrichment analysis of bulk RNA-seq from TCF1$^{WT}$LEF1$^{WT}$ and
TCF1$^{\Delta}$LEF1$^{\Delta}$ B-1a cells shows enrichment for the indicated signalling pathways.
The black dotted lines indicate neutral enrichment. **b**, Correlation analysis
between expression of TCF1 or LEF1 and expression of MYC (top) or BCL2
(bottom) in peritoneal B-1a, B-1b and B-2 cells ($n = 6$). gMFI, geometric mean
fluorescence intensity. **c**, UMAP of scRNA-seq profiles from the four indicated
sorted cell subsets (FL B1P and BM B1P), B-1 cells from the spleen (SP B1)
and peritoneal cavity (PC B1) barcoded and pooled for sequencing.
**d**,**e**, Developmental trajectory of the combined (**d**) or individual (**e**) FL B1P,
BM B1P, SP B1 and PC B1 populations constructed by Monocle2. **f**, Comb plots

displaying the incidence and amplitude of the indicated genes in each subset
shown in panel **d**. **g**, Dot plot presentation of hallmark gene sets upon gene
set enrichment analysis for differentially expressed genes in B-1a cells of the
indicated genotypes. **h**, Dot plot presentation of expression of the indicated
genes in human prenatal B-1 cells according to single, double or no TCF1 and
LEF1 expression. Panel **h** was adapted with permission from ref. 23, AAAS. Data
are representative of $n = 2$ experiments. Statistical analysis was performed
using two-tailed Pearson correlation analysis (**b**). The significant hallmark
gene sets in MSigDB (adjusted $P < 0.05$) for each contrast based on the
hypergeometric testing using cluster profiler package (**g**).

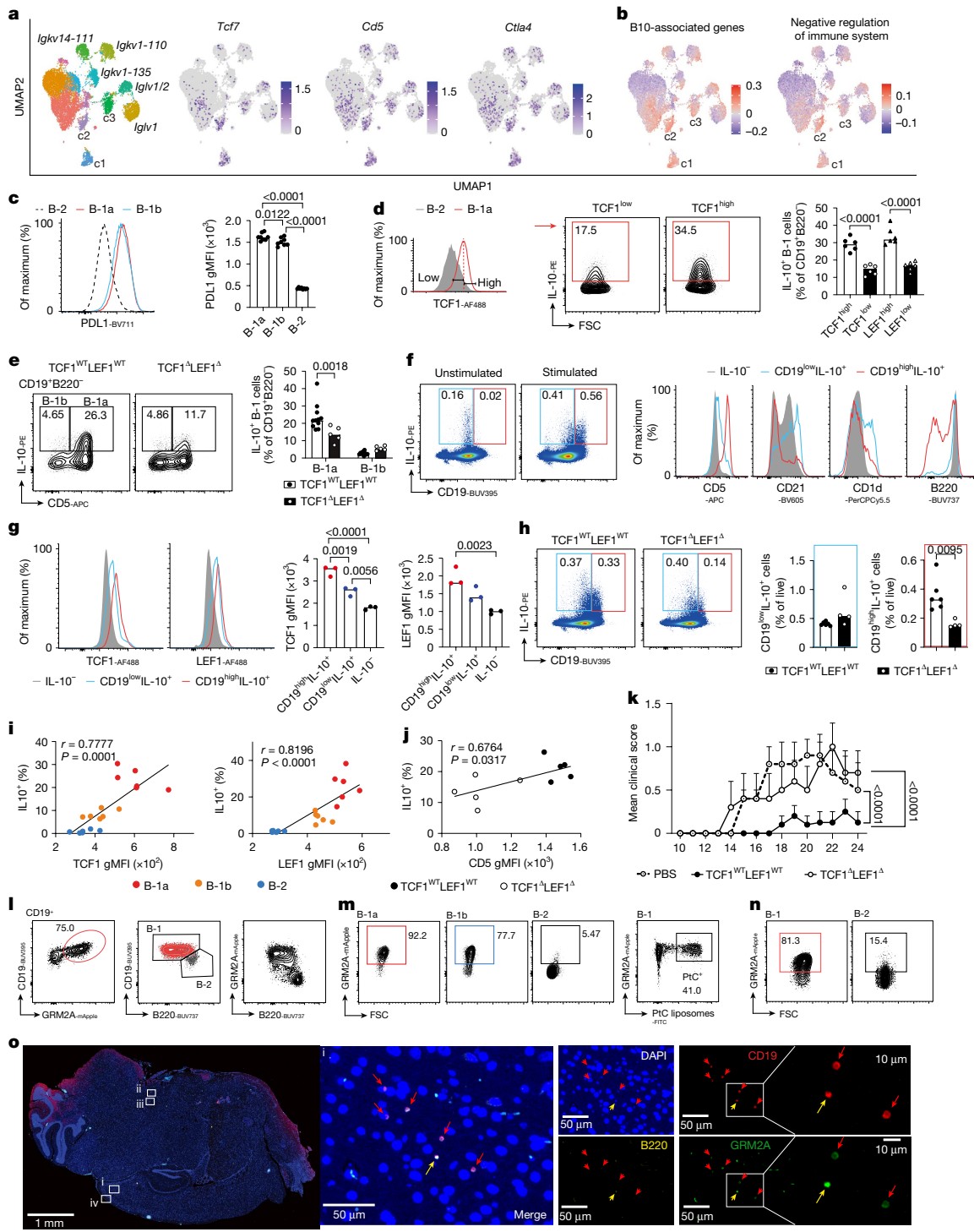

**Fig. 4 | TCF1 and LEF1 promote IL-10 production by B-1a cells and their control of CNS inflammation. a**, UMAP showing clonal BCR usage in clusters of peritoneal B cells and expression of *Tcf7*, *Cd5* and *Ctla4*. **b**, Gene module scores associated with gene signatures. **c**, PDL1 histograms and gMFI in the indicated cell subsets (*n* = 8). **d**,**e**, Flow cytometry plots and quantification of IL-10+ cells in TCF1high or TCF1low PC B1 cells (*n* = 6; **d**) and IL-10+ cells within PC B1 cells from TCF1WTLEF1WT (*n* = 11) and TCF1ΔLEF1Δ mice (*n* = 5; **e**). **f**, Expression of CD5, CD21, CD1d and B220 by CD19lowIL-10+ (blue), CD19highIL-10+ (red) and IL-10− (grey) cells with or without stimulation. **g**, TCF1 and LEF1 histograms (left) and quantification (right) in the cell subset shown in panel **f** (*n* = 3). **h**, Pseudocolour plots and quantification of IL-10+ splenic B cells from TCF1WTLEF1WT (*n* = 6) and TCF1ΔLEF1Δ (*n* = 4) mice. **i**,**j**, Correlation analysis between TCF1 or LEF1 expression and the percentage of IL-10+ cells (*n* = 6; **i**) or expression of CD5 and the percentage of IL-10+ in peritoneal B-1a cells from TCF1WTLEF1WT (*n* = 5) and

TCF1ΔLEF1Δ (*n* = 5; **j**). **k**, Mean clinical scores of mice treated with either PBS (*n* = 5) or adoptively transferred with peritoneal B-1 cells from TCF1WTLEF1WT (*n* = 5) or TCF1ΔLEF1Δ (*n* = 5). **l**–**n**, Contour plots of GRM2A-positive cells within the indicated cell populations from the peritoneal cavity (**l**,**m**) and the spleen (**n**). **o**, Identification of GRM2A-positive B-1 cells in the EAE mouse brain; representative cells are from the area labelled 'i'. Representative cells from the areas labelled 'ii–iv' are shown in Extended Data Fig. 7h. Each symbol represents an individual mouse, and the bars represent the median values (**c**–**e**,**g**,**h**). Data are presented as mean ± s.e.m. (**k**). Data are representative of *n* = 2 experiments (**c**–**e**,**g**–**k**) and *n* = 3 mice (**l**). Statistical analysis was performed using two-tailed Pearson correlation analysis (**i**,**j**), two-tailed Mann–Whitney *t*-test (**h**), one-way ANOVA with Tukey multiple-comparison test (**c**–**e**,**g**) and two-way ANOVA (**k**). The exact *P* values are shown.

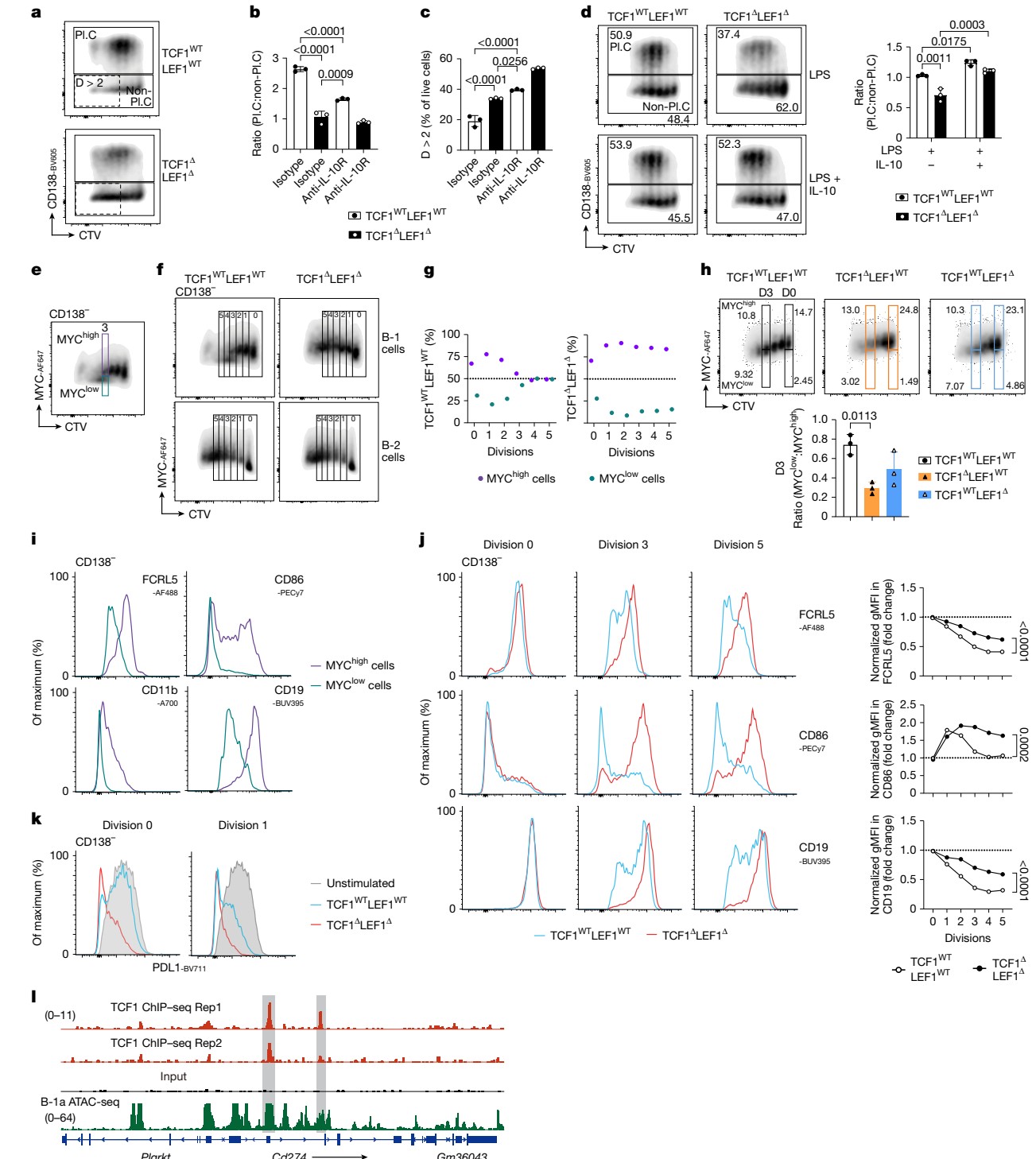

**Fig. 5 | TCF1 and LEF1 deficiency in B-1 cells prevents emergence of stem-like B-1 cells and promotes exhaustion.** Purified peritoneal cavity TCF1$^{WT}$LEF1$^{WT}$ or TCF1$^{\Delta}$LEF1$^{\Delta}$ B-1 cells were cultured with LPS, IL-10R-Fc antibody or IgG1 isotype for 3 days. **a**,**b**, Density plots (**a**) and quantification (**b**) of the mean ratio of plasma cells (Pl.Cs; CD138$^{+}$):non-plasma cells (CD138$^{-}$; **b**). **c**, Percentage of non-plasma cells undergone more than two divisions. **d**, Density plots (left) and mean ratio of plasma cells:non-plasma cells (right) after 3 days of LPS stimulation with or without IL-10. **e**, Representative gating strategy of MYC$^{high}$ and MYC$^{low}$ cells at division 3 (D3). **f**,**g**, Density plots of MYC expression in proliferating cells (**f**) and quantification of the percentage of MYC$^{high}$ and MYC$^{low}$ in each division measured by CTV dilution (**g**). **h**, Representative gating strategy of MYC$^{high}$ and MYC$^{low}$ cells at undivided (D0) and D3 (top) and the ratio of MYC$^{low}$:MYC$^{high}$ (bottom). **i**, Histograms of CD86, FCRL5, CD11b and

CD19 expression among MYC$^{high}$ and MYC$^{low}$ cells. **j**, Histograms (left) and quantification (right) of CD86, FCRL5 and CD19 expression on TCF1$^{WT}$LEF1$^{WT}$ or TCF1$^{\Delta}$LEF1$^{\Delta}$ B-1 cells stimulated with LPS for 3 days that are either D0 or have undergone D3 or D5. **k**, Histogram of PDL1 expression in D0 and D1 of TCF1$^{WT}$LEF1$^{WT}$ or TCF1$^{\Delta}$LEF1$^{\Delta}$ B-1 cells stimulated with or without LPS for 3 days. **l**, Chromatin immunoprecipitation followed by sequencing (ChIP–seq) analysis of TCF1 binding in CD4$^{+}$CD8$^{+}$ thymocytes[42] and ATAC-seq analysis at the *Pdl1* locus on B-1a cells from *Vh12/Vk4* transgenic mice[17]. Panel **l** was adapted from refs. 17,42, Springer Nature Ltd. Data are presented as mean ± s.d. Data are representative of *n* = 2 experiments with a total of 9 mice per genotype, with peritoneal cavity cells from 3 mice being pooled together. Statistical analysis was performed using one-way ANOVA with Tukey multiple-comparison test (**b**–**d**,**h**) or two-way ANOVA (**j**). The exact *P* values are shown.

IL-10 (Fig. 3h), it is likely that these transcription factors promote IL-10 production upon postnatal exposure to commensal bacteria bearing Toll-like receptor (TLR) and B-1a BCR ligands (for example, PtC) that coincides with the largest wave of B-1 cells[51]. Together, our data revealed that TCF1 and LEF1 promote the induction of IL-10[+] B-1a cells in both the peritoneal cavity and the spleen.

## TCF1–LEF1 limit CNS inflammation

B cells have been shown to have protective roles in human multiple sclerosis[52] and its experimental autoimmune encephalomyelitis (EAE) mouse model[53]. In EAE, such regulation requires B-1a cells rather than recirculating follicular B-2 cells[48,54] and is dependent on IL-10 (ref. 53). To test whether TCF1 and LEF1 were required for B-1a-mediated regulation in EAE, an equal number of activated peritoneal B-1 cells from TCF1–LEF1-sufficient or TCF1–LEF1-deficient mice was adoptively transferred 3.5 days after recipient mice were immunized with myelin oligodendrocyte glycoprotein (MOG$_{35-55}$). Clinical disease severity was significantly reduced in mice receiving TCF1$^{WT}$LEF1$^{WT}$ B-1 cells and disease onset was also delayed compared with mice receiving TCF1$^{\Delta}$LEF1$^{\Delta}$ B-1 cells (Fig. 4k and Extended Data Fig. 7d).

We asked whether adoptively transferred B-1 cells can migrate into the CNS. Activation of B-1 cells modulates markers used for their identification[55]. We thus generated *Gramd2a*$^{mApple-cre}$ mice in which B-1 cells can be tracked due to expression of the mApple fluorochrome driven by *Gramd2a* (encoding GRM2A) that we found to be selectively expressed in peripheral B-1 cells (Fig. 4l–n and Extended Data Fig. 7e–g). Immunofluorescence of brain sections 21 days after B-1 cell adoptive transfer in the EAE model (Fig. 4o and Extended Data Fig. 7h) revealed GRM2A$^+$ donor-derived B-1 cells in the brain. These were predominantly located underneath the meninges or nearby parenchyma. Although this suggests that B-1a cells can enter the brain and may thus exert local regulatory effects, we cannot exclude additional distal effects in secondary lymphoid organs.

## IL-10 promotes B-1a stemness

Finally, we considered the possibility that TCF1 and/or LEF1 control B-1 cell expansion and self-renewal via boosting B cell-derived IL-10, as B-1 cell-derived IL-10 has been shown to repress proliferation[56]. In the absence of TCF1–LEF1, LPS-activated B-1 cells proliferated extensively and the ratio of CD138$^+$ plasma cells to non-plasma cells was decreased (Fig. 5a–c). IL-10 blockade using IL-10R-Fc antibody in wild-type B-1 cell cultures mimicked the excessive proliferation seen in TCF1–LEF1-deficient B-1 cells (Extended Data Fig. 7i), whereas the addition of IL-10 to cultures of TCF1–LEF1-deficient B-1 cells partially rescued this phenotype (Fig. 5d).

In CD8$^+$ T cells, TCF1 expression allows the emergence of resting memory 'stem-like' MYC$^{low}$ cells from cycling effector MYC$^{high}$ cells via regulation of proliferation, metabolism and differentiation[30]. A comparable MYC$^{low}$ population appeared among B-1a but not B-2 cells around the third division post-LPS stimulation, and MYC$^{low}$ cells exhibited limited proliferation (Fig. 5e and Extended Data Fig. 7j,k). MYC$^{low}$ cells were not generated in B-1 cell cultures lacking TCF1–LEF1 where cells continued to proliferate extensively without downregulating CD19 (Fig. 5f,g). Loss of MYC1$^{low}$ cells appeared more profound with single TCF1 than single LEF1 deficiency (Fig. 5h). Consistent with being a stem/memory-like population, emerging MYC$^{low}$ B-1a cells at the third division downregulated CD19 and markers associated with B cell activation and exhaustion including CD86, FCRL5 and CD11b (Fig. 5i); this was not seen in TCF1–LEF1-deficient cells (Fig. 5j). Expression of CD11c and ZEB2 that identify 'aged-associated B cells'[57] remained unchanged in TCF1–LEF1-deficient B-1a cells (Extended Data Fig. 7l and data not shown). In the absence of TCF1–LEF1, PDL1 was strongly downregulated in activated B-1 cells independently of cell division (Fig. 5k). Reanalysis of

published ATAC-seq from B-1a cells[17] revealed two chromatin-accessible regions in the first two exons of *Cd274* (encoding PDL1; Fig. 5l), suggesting that PDL1 is a direct target of TCF1 as reported in double-positive thymocytes[42]. Thus, TCF1–LEF1 are likely to also promote the regulatory function of B-1a cells by maintaining PDL1 expression. Together, these data suggest that TCF1–LEF1 promote PDL1 and IL-10 expression that contribute to the regulatory function of B-1a cells while limiting excessive proliferation and exhaustion to allow emergence of a MYC1$^{low}$ stem-like population essential for B-1a maintenance.

Our findings suggest that TCF1 and LEF1 are important for the generation of IL-10-producing regulatory B-1 cells in the periphery. As the major wave of B-1 cell expansion occurs postnatally upon microbiome exposure[58] and bacterial CpG is a potent inducer of IL-10 secretion[59], B-1a regulatory function is probably key to help prevent inappropriate immune responses to commensal bacteria and viruses. Gut IgA plasma cells have been shown to arise from the same haematopoietic progenitors as B-1a cells during ontogeny and produce IgA clones in response to neonatal gut viral infection[60]. Whether early-life B-1a education and their antimicrobial IgA production determines the outcome of protective versus pathogenic or autoimmune reactions to microbial exposures later in life remains to be determined. Our identification of expanded B-1-like cells in the blood of some patients with pleural infection based on CD5, CD43, TCF1 and LEF1 expression suggests that these cells may also serve as a biomarker for early detection of serosal infections (for example, pleuritis and peritonitis) and possibly sepsis. Given the observed gradation in PtC reactivity across CD43$^+$CD5$^+$, CD43$^-$CD5$^+$ and CD43$^-$CD5$^-$ circulating populations in healthy donors—and the fact that a fraction of CD38$^+$ cells also exhibit PtC reactivity—it is likely that surface markers such as CD5, CD43 and CD38 change dynamically as B-1-like cells mature. Similarly, an inter-relationship may exist between CD38$^+$ and CD38$^-$ B-1-like cells, with CD38 expression changing as these cells mature and undergo activation. Our data also suggest that chronic activation of B-1 cells—known to be selected for self-reactivities early in life[32]—or loss of IL-10 or PDL1-mediated regulatory function as observed in the absence of TCF1–LEF1, may contribute to autoimmunity. Although TCF1 and LEF1 deficiency had more pronounced effects on B-1 cells, it is possible that these transcription factors also contribute to the homeostasis of long-lived B-2 cell subsets. We observed a trend towards reduced B-2 cell turnover and diminished surface BCR expression in the absence of TCF1 and LEF1. This effect may be particularly relevant for B-2 cells that rely on self-renewal for their maintenance, such as memory B cells, which exhibit the highest expression levels of both transcription factors among B-2 cell populations. Together, our work paves the way for studies that can harness the TCF1–LEF1 axis to control B cell stemness and regulatory ability, and to further investigate the function of human B-1-like cells in infection, autoimmune diseases and cancer.

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

## Methods

### Mice

*Tcf7*[flox/flox]*Lef1*[flox/flox] mice were generated by H.-H.X. (Center for Discovery and Innovation)[20,61]. *Mb1*[Cre] mice, *Rag1*[−/−] and CD45.1 mice were maintained on a C57BL/6 background and housed in specific pathogen-free conditions at the Australian National University (ANU) Bioscience Facility. *Tcf7*[flox/flox]*Lef1*[flox/flox] mice were backcrossed at least six times onto the C57BL/6 background, then were subsequently crossed to *Mb1*[Cre] mice to generate conditional *Tcf1;Lef1*-knockout mice. Mice were used from 8 to 12 weeks, and they were age matched when comparing the effect of two genotypes, except for ageing assessment (4–21 weeks). The *Gramd2a*-T2A-iCre-P2A-mApple strain was generated by the Genetic Modification Service at the Francis Crick Institute. Following adeno-associated virus (AAV) induction of a packaged rAAV donor molecule CRIPSR–Cas9 reagents were electroporated into fertilized single-cell C57BL/6 zygotes. The donor vector contained an in-frame T2A iCre P2A mApple cassette flanked by 5′ and 3′ regions of homology (400 bp, respectively) designed to insert the donor template in-frame immediately upstream of the *Gramd2a* stop codon in exon 12, such that expression of iCre and mApple was controlled by the endogenous *Gramd2a* machinery. The AAV donor vector was synthesized and packaged into AAV serotype 1 by VectorBuilder. The guide sequence used was 5′-ATGTGCAGGTAACAGTCACT-3′ and this was synthesized as a synthetic guide RNA by IDT. Mouse procedures were approved by the ANU's Animal Experimentation Ethics Committee or UK Home Office under project licence (PP2867252). *Ets1*[flox/flox] mice crossed more than 12 generations onto a C57BL/6 background were obtained from B. Kee at the University of Chicago. CD19–Cre mice on a C57BL/6 background were obtained from Jackson Laboratories and crossed to *Ets1*[flox/flox] mice to generate knockout CD19–Cre *Ets1*[flox/flox] (Ets1[Δ])[62] and control CD19–Cre (*Ets1*[WT]) mice. Mice were used at both 9 weeks and 20 weeks of age to compare phenotypes. Mouse procedures involving *Ets1*[flox/flox] mice were approved by the University at Buffalo Institutional Animal Care and Use Committee.

### Human samples

All patients and healthy controls provided written informed consent. Patients with confirmed bacterial pleural infection were invited to donate samples (Extended Data Table 1); all patients had pleural neutrophil counts of more than 10 million per litre, a neutrophil:lymphocyte ratio of more than 4 and were on antibiotic treatment. All clinical specimens (pleural fluid and blood) were collected for Oxford Radcliffe Pleural Biobank (ethical approval reference: 19/SC/0173, South Central-Oxford C Research Ethics Committee) and the study was approved by the Oxford Radcliffe Biobank Tissue Access Committee (reference number: 22/A093). All donors provided written informed consent.

Patients and healthy donors were also recruited following favourable study approval on 4 March 2024 by the London-Brent Research Ethics Committee (REC reference 24/PR/0218, IRAS 330505) and as part of the CAPTURE (NCT03226886) study, a prospective longitudinal cohort study investigating COVID vaccine responses in patients with cancer (Extended Data Table 2). CAPTURE is sponsored by the Royal Marsden Hospital and was approved as a substudy of TRACERx Renal (NCT03226886) by the National Research Ethics Service Committee London, Fulham on 1 May 2020 (REC 11/LO/1996). Baseline samples before COVID vaccination from patients with untreated CLL were selected. Collected as part of the EXACT study, a prospective longitudinal cohort study investigating response to immune checkpoint inhibitor therapy in patients with cancer. EXACT is sponsored by the Royal Marsden Hospital and was approved by the National Research Ethics Service Committee, West Midlands-Black Country on 22 November 2021 (REC/WM/0251). A baseline sample from one patient with untreated CLL and before immune checkpoint inhibitor therapy was selected.

Patients and healthy donor peripheral blood samples (Extended Data Table 3) were obtained upon written informed consent, according to protocols NCT03394053 (www.clinicaltrials.gov) approved by the Institutional Review Boards of Children's National Hospital and of the National Institutes of Health and 93-I-0119, approved by the Institutional Review Board of the National Institutes of Health.

Adenoid tissue samples were collected from children (2–6 years of age, two girls and four boys) undergoing elective adenoidectomy surgery for either otitis media with effusion (OME) or sleep disordered breathing (SDB) at the Great North Children's Hospital. The parents of all donors provided written consent for tissue donation for research and this project was approved by Newcastle University Biobank (project code: NRTB-1).

Ethical approval to obtain blood from healthy individuals was provided by the London-Brent Regional Ethics Committee (REC: 21/LO/0682).

### Human sample processing

Pleural fluid and blood specimens were transferred and processed in the laboratory within hours after collection. Pleural fluid specimens were centrifuged at 800*g* for 10 min. The acellular supernatant was removed and stored in −80 °C. Then, the cellular pellet was resuspended in red blood cell lysis buffer (5–10 ml; J62990.AK, Thermo Scientific) and incubated for 5 min. The sample was centrifuged at 500*g* for 5 min and the supernatant was discarded. If there were red blood cells, the process was repeated, otherwise the cells were washed in 5 ml of PBS and centrifuged at 500*g* for 5 min. The supernatant was discarded, and the cells were resuspended in 5 ml of RPMI enriched with 10% FBS, passed through a 70-µm filter and counted. Whole blood was collected in EDTA tubes (VWR) and stored at 4 °C until processing or processed freshly. All samples were processed within 24 h. Before processing, tubes were brought to room temperature. Peripheral blood mononuclear cells and plasma were isolated by density-gradient centrifugation using centrifugation tubes (SepMate, StemCell) per the manufacturer's instructions. Peripheral blood mononuclear cells either proceeded to be stained with antibodies directly or resuspended in recovery cell culture freezing medium (Fisher Scientific) containing 10% DMSO, placed overnight in CoolCell freezing containers (Corning) at −80 °C and then transferred to liquid nitrogen for long-term storage. Adenoid tissue samples were collected in PBS and immediately mechanically dissociated into a single-cell suspension using scalpel blades and a 100-µm filter. Cells were then cryopreserved in freezing media (10% DMSO and 90% FBS) in liquid nitrogen before flow cytometry experiments.

### Flow cytometry

Single-cell suspension from mouse peritoneal cavity, spleens and bone marrow were treated with TruStain FcX rat anti-mouse CD16/32 antibodies (101320, BioLegend). The cells were then stained with primary antibodies used for mouse samples along with LIVE/DEAD Fixable Aqua Dead Cell strain (Invitrogen) or eBioscience Fixable eFluor780 viability dye (Invitrogen). The primary antibodies included: B220-BUV737 (RA3-6B2; 612838, BD Horizon), CD19-BV605 (6D5; 115540, BioLegend), CD23-BV421 (B3B4; 101621, BioLegend), CD5-APC (53-7.3; 17-0051-82, eBioscience), IgM-PE-Cy7 (II/41; 25-5790-82, eBioscience), CD19-BUV395 (1D3; 563557, BD Horizon), CD3-BV650 (17A2; 100229, BioLegend), CD21-BV605 (7G6; 747763, BD), IgD-PerCP/Cyanine5.5 (11-26c.2a; 405710, BioLegend), CD93-BV480 (AA4.1; 746239, BD), CD93-PE/Cyanine7 (AA4.1; 136506, BioLegend), CD24-Pacific Blue (M1/69; 101820, BioLegend), CD43-BV605 (S7; 563205, BD), CD3-APC-Cy7 (17A2; 100222, BioLegend), CD4-APC-Cy7 (GK1.5; 100414, BioLegend), CD11b-APC-Cy7 (M1/70; 101226, BioLegend), CD11b-AF700 (M1/70; 101222, BioLegend), TER119-APC-Cy7 (TER-119; 560509, BD), Gr1-APC-Cy7 (RB6-8C5; 108424, BioLegend), Sca-1-BV421 (Ly-6A/E; 108128, BioLegend), Kit-APC (2B8; 105812, BioLegend), CD127-PE-Cy7 (A7R34; 135014, BioLegend), CD16/32-PerCP/Cyanine5.5 (93; 101324, BioLegend), CD135-PE (A2F10; 135306,

BioLegend), CD45.1-BV711 (A20; 110739, BioLegend), CD45.2-BUV737 (104; 612779, BD), CD274-BV711 (B7-H1; 124319, BioLegend), CD86-PE-Cy7 (GL-1; 105014, BioLegend), FCRL5-AF488 (FAB6757G, biotechne), CD223(LAG-3)-PE (C9B7W; 125208, BioLegend), CD138-BV605 (281-2; 142515, BioLegend), CD138-PE (281-2; 142504, BioLegend), CD43-BV711 (S7; 740668, BD), CD43-BV605 (S7; 747726, BD), CD1d-PerCP/Cy5.5 (1B1; 123513, BioLegend), Ig light chain κ-AF700 (RMK-45; 409508, BioLegend), Ig light chain λ-FITC (R26-46; 553434, BD), IgG3-biotin (RMG3-1; 406803, BioLegend), streptavidin-BV605 (405229, BioLegend), BLNK phosphorylated at Tyr84 (J117-1278; 558442, BD), ERK1/2 phosphorylated at Thr202/Tyr204 (20A; 561991, BD), phosphorylated PLCγ2 (K86-1161; 560134, BD), SYK phosphorylated at Tyr525/526 (C87C1; 6485S, Cell Signaling Technology), BTK phosphorylated at Tyr551 (M4G3LN; 12-9015-42, eBioscience), Ki-67-PerCP-eFluorTM710 (SolA15; 46-5698-82, eBioscience), MYC-AF647 (Y69; ab190560, abcam), TCF1/TCF7-AF488 (C63D9; 6444S, Cell Signaling Technology) and LEF1-AF488 (C12A5; 8490S, Cell Signaling Technology). DOPC/CHOL/fluorescein-DHPE (54:45:1) and rhodamine-DHPE liposomes (DOPC/CHOL/rhodamine-DHPE(54:45:1)) were used (F60103F2-R, FormumMax).

For cytokine intracellular staining, the cells were stimulated with phorbol 12-myristate 13-acetate (PMA), ionomycin, brefeldin A and LPS for the terminal 5 h of culture. Cells were harvested and stained for surface markers, including ebioscience Fixable eFluor780 viability dye (Invitrogen) to exclude dead cells before cells were fixed. Intracellular staining was performed with the Cytofix/Cytoperm kit (BD) with IL-10-PE (JES5-16E3; 505008, BioLegend) and IL-10-BV421 (JES5-16E3; 563276, BD) as recommended. For transcription factors, the eBioscience FOXP3/transcription factor staining buffer set (Invitrogen) was used per the manufacturer's instructions. For staining phosphorylated BCR signalling components, peritoneal cells were isolated using ice-cold FACS buffer (2% FBS in PBS) and stained on ice with CD19, B220 and CD5 antibodies, followed by immediately being fixed with pre-warmed 1.6% paraformaldehyde for 10 min at 37 °C. Cells were then permeabilized with PERM III buffer (558050, BD) for 30 min on ice and subsequently stained with phosphorylated BCR signalling components for 40 min at room temperature.

Human samples were treated with human TruStain FcX CD16, CD32 and CD64 antibodies (BioLegend) and then stained with the following anti-human antibodies: CD3-BV785 (HIT3a; 740961, BD Bioscience), CD19-BUV737 (SJ25C1; 612756, BD Bioscience), CD19-PerCP/Cyanine5.5 (SJ25C1; 363016, BioLegend), CD27-PECy7 (M-T271; 560609, BD Bioscience), CD38-BV605 (HIT2; 569699, BD Bioscience), CD43-BV510 (1G10; 563377, BD Horizon), CD43-FITC (1G10; 555475, BD), CD38-BV510 (HIT2; 303540, BioLegend), CD5-APC (UCHT2; 300612, BioLegend), CD5-PE/Cyanine7 (UCHT2; 300622, BioLegend), IgA-PerCP-Vio700 (IS11-8E10; 130-113-478, MACS), CD24-BV711 (ML5; 563401, BD), IgM-BUV395 (G20-127; 563903, BD), TCF1/TCF7 (C63D9; 2203, Cell Signaling Technology), LEF1 (EPR2029Y; ab137872, Abcam), Alexa Fluor-488 donkey anti-rabbit IgG (minimal x-reactivity Poly4064; 406416, BioLegend) and Alexa Fluor-647 donkey anti-rabbit IgG (minimal x-reactivity Poly4064; 406414, BioLegend). Flow cytometers (LSRFortessa X-20, FACSAria II and LSR II; BD) and software (CellQuest and FACSDiva; BD) were used for the acquisition of flow cytometric data, and FlowJo software (Tree Star) was used for analysis.

## Generation of bone marrow and fetal liver chimeras
To generate fetal liver chimera, $1 \times 10^6$ fetal liver cells from $Tcf7^{+/flox}$ $Lef1^{+/flox}$Cre$^{Mb1}$ or $Tcf7^{flox/flox}Lef1^{flox/flox}$Cre$^{Mb1}$ mice (at embryonic day 14.5) were transferred intravenously into sublethally irradiated (500 rad) $Rag1^{-/-}$ recipients. To generate bone marrow chimeras, $2 \times 10^6$ bone marrow-derived haematopoietic stem cells from 8-week-old mice with aforementioned genotypes were transferred intravenously into sublethally irradiated $Rag1^{-/-}$ recipients. Mice were given Bactrim in their drinking water for 48 h before injection and for 6 weeks after injection and housed in sterile cages. After 6 weeks

of reconstitution, mice were taken down for phenotyping by flow cytometry.

## Adoptive cell-transfer experiments
Equal numbers ($1 \times 10^5$ cells) of flow-cytometry-sorted peritoneal CD19$^+$B220$^-$CD5$^+$CD23$^-$ B-1a cells from $Tcf7^{+/+}Lef1^{+/+}$ (CD45.2$^+$) or $Tcf7^{flox/flox}Lef1^{flox/flox}$Cre$^{Mb1}$ (CD45.2$^+$) and wild type (CD45.1$^+$) were adoptively transferred into $Rag1^{-/-}$ recipients intraperitoneally. The frequency of donor B cells among total peritoneal and splenic B cells was analysed at 2 months after transfer.

## Peritoneal cell transfer and in vivo BrdU incorporation assay
Peritoneal cells were harvested from $Tcf7^{+/+}Lef1^{+/+}$ (CD45.2$^+$) or $Tcf7^{flox/flox}Lef1^{flox/flox}$Cre$^{Mb1}$ (CD45.2$^+$) donors by injecting 5 ml of serum-free, DMEM medium without L-glutamine (Gibco) into the peritoneal cavity and withdrew as much fluid as possible, followed by centrifuging to harvest the cells; $3 \times 10^6$ cells were then injected intraperitoneally into the CD45.1 recipients. The recipients were fed with water containing 0.8 mg ml$^{-1}$ BrdU for 12 days. Mice were then killed; the frequency of BrdU-positive cells among indicated donor cells in the peritoneal cavity was analysed using a BrdU staining kit (8811-6600-42, Thermo Fisher) and analysed by flow cytometry.

## ELISA
Flat-bottom 96-well ELISA plates (3855, Thermo Fisher) were coated with goat anti-mouse κ-UNLB (1050-01, Southern Biotech) overnight. The plates were subsequently washed and blocked using 1% bovine serum albumin in 1× PBS for 1.5 h at 37 °C. Serial dilution of the mouse serum was added to the wells and incubated overnight at 4 °C. Targeted antibodies were detected using AP-conjugated goat anti-mouse IgM (1020-04, Southern Biotech) and AP-conjugated goat anti-mouse IgG3 (1100-04, Southern Biotech). Plates were developed using 1 mg ml$^{-1}$ phosphatase substrate tablets (S0942, Sigma-Aldrich), and the absorbances at 405 nm and 605 nm were measured using an Infinite 200 PRO plate reader (Tecan) equipped with i-control (v1.9) software. Serum was added in serial dilution before the addition of secondary antibody: alkaline phosphatase (AP)-conjugated goat anti-mouse IgM (1020-04, Southern Biotech) and AP-conjugated goat anti-mouse IgG3 (1100-04, Southern Biotech).

## Flow cytometric analysis of intracellular IL-10 synthesis
In brief, isolated leukocytes or purified cells were resuspended ($1 \times 10^6$ cells per millilitre) with LPS (10 μg ml$^{-1}$), PMA, ionomycin and brefeldin A (1:500; BioLegend) for 5 h. For IL-10 detection, cells were stained with surface markers followed by fixation, permeabilization with the Cytofix/Cytoperm kit (BD Bioscience) and staining with IL-10-PE (JES5-16E3, BioLegend) or IL10-BV421 (JES5-16E3, BD) according to the manufacturer's instructions.

## RNA-seq library preparation and data analysis
Total RNA was purified from sorted peritoneal B-1a cells (CD19$^+$CD3$^-$B220$^-$CD5$^+$CD23$^-$7AAD$^-$) using the PicoPure RNA isolation kit. Library construction and sequencing were performed in the Biomolecular Resource Facility, the John Curtin School of Medical Research, ANU. The single end reads of 76-bp sequencing were generated on a HiSeq2000 machine with a depth of more than 30 million reads per sample. The raw reads were aligned to the mm10 (GRCm38) genome assembly using hisat2 (ref. 63) and the mapped reads were assigned with FeatureCounts (v2.4)[64] based on the genome-build GRCm38.p4 annotation and NCBI Refseq gene mode by removing ribosomal genes and non-coding RNA, respectively. Differential expression analyses were performed with voom-limma[65], after removal of lowly expressed genes and normalized using the trimmed mean of $M$-values method[66,67]. Significantly differentially expressed genes were identified by applying a Benjamini–Hochberg adjusted $P$ value threshold of 0.05 (ref. 68).

Gene set enrichment or pathway analysis were performed using clusterProfiler[69] and Camera[70] against the Gene Ontology database, KEGG database and HALLMARK C2 and C7 gene sets in the MSigDB (v7.5).

## scRNA-seq

Peritoneal B cells (CD19$^+$CD3$^-$7AAD$^-$) from three 8-week-old mice, B-1Ps (Lin$^-$CD93$^+$IgM$^-$CD19$^+$B220$^{low/-}$) from embryonic day 18.5 fetal livers (three fetal livers were pooled together as one replicate) or bone marrow from three 8-week-old mice, and B-1 cells (CD19$^+$B220$^{low/-}$) from the peritoneal cavity and spleen from three 8-week-old mice were sorted by the FACS Aria II cell sorting system (BD Immunocytometry Systems). Cells ($n = 10,000$–20,000) per sample were run on the 10X Chromium platform (10X Genomics). Library preparation and sequencing were performed by the Australian Cancer Research Foundation Biomolecular Resource Facility, the John Curtin School of Medical Research, ANU or Genomics STP, The Francis Crick Institute according to the manufacturer's instructions for the Chromium Next GEM Single Cell 5′ Kit v2 or v3. Two libraries were generated and the mRNA transcript expression and BCR repertoire were measured. The samples were sequenced using the NovaSeq 6000 (Illumina) system or NovaSeq S2. The FASTQ files were aligned to the mm10 mouse reference genome using the 10X Genomics CellRanger pipeline (v6.0.1 or v7.0.1). Data were processed and analysed using the Seurat package (v5.0.3) in R (v4.4.1). Raw gene expression matrices were filtered to exclude low-quality cells based on the thresholds for mitochondrial gene content (set at 5%) and the number of detected genes per cell, using the PercentageFeatureSet() and subset() functions.

For multiplexed samples labelled with cell hashing antibodies, demultiplexing was performed using the MULTIseqDemux() function in Seurat. Hashtag oligonucleotide counts were normalized using centred log-ratio (CLR) transformation via the NormalizeData() function with normalization.method = 'CLR', and cells were classified as singlets, doublets or negatives based on the hashtag oligonucleotide signal. Only singlets were retained for downstream analysis.

Normalization and integration were performed using the scTransform() workflow, followed by the identification of highly variable features. Dimensionality reduction was performed using principal component analysis via the RunPCA() function, and the top 20–30 principal components were selected to construct a shared nearest neighbour graph using FindNeighbours(). Clustering was conducted using the FindClusters() function. Clusters were visualized using UMAP via the RunUMAP() function. Differential expressed genes for each cluster were identified using the FindAllMarkers() function with test. use set as either 'wilcox' or 'MAST'. Where applicable, cell types were annotated based on canonical marker expression or informed by publicly available datasets.

## Cell trajectory analysis

Cell trajectory analysis was performed using Monocle2 (v2.32.0) in R (v4.4.1). The annotated Seurat object was first transformed into a CellDataSet object using the as.CellDataSet() function from the Seurat package. To filter relevant genes, those expressed in at least 10 cells and with a mean expression value of 0.1 or greater were selected. Next, highly variable genes across cell subsets were identified using the differentialGeneTest() function with the formula fullModelFormulaStr = '~cell_subset'. On the basis of their $q$-value, the top 2,000 genes were selected, and setOrderingFilter() was applied to prioritize them for ordering in pseudo-time analysis. Dimensionality reduction was performed using reduceDimension() with the parameter reduction_method = 'DDRTree'. Cells were subsequently ordered along the trajectory using the orderCells() function and visualized with plot_cell_trajectory(). All other parameters were set to their default values.

## Single-cell BCR sequencing analysis

BCR repertoire analysis was conducted with the Immcantation framework, following the guidelines provided by its developers (http://immcantation.org/). This was implemented in a Python environment (v3.11.10). We began with 10X Genomics BCR sequencing data processed through the CellRanger pipeline (v7.0.1) to obtain annotated clonotype and contig sequences. For V(D)J gene assignment, the AssignGenes.py command from the Change-O toolkit was used, referencing the mouse IMGT and IgBLAST (v1.22.0) databases. MakeDb.py was used to standardize the data to AIRR format. Cells expressing more than one heavy chain or only light chains were removed. Clonotypes were assigned using the DefineClones.py tool based on heavy chain sequences, using the default Hamming distance substitution model. Subsequently, light_cluster.py was applied to refine clonotype grouping by incorporating light chain information. Finally, the resulting datasets were merged with the annotated Seurat file and key metrics, including V(D)J gene usage, CDR and FWR sequences, lengths and mutational content were extracted for downstream analysis and visualization.

For diversity analysis, we used the iNEXT package (v3.0.1) to perform coverage-based rarefaction and extrapolation analyses. We computed the three most common Hill numbers: species richness ($q = 0$), Shannon diversity ($q = 1$) and Simpson diversity ($q = 2$). The resulting sampling curves, which illustrate diversity estimates with respect to sample coverage, were visualized using ggiNEXT() function. In addition, we constructed circle packing plots using the packcircles package (v0.3.6) to visualize the immune repertoire composition of individual samples. These analyses were conducted in R (v4.4.1).

## scRNA-seq analysis on human prenatal B-1 cluster

Processed and annotated scRNA-seq data of fetal immune cells, specifically the haematopoietic stem and progenitor cells and the B cell lineages, in Suo et al.[23] were downloaded from the developmental cell atlas portal (https://developmentcellatlas.cellgeni.sanger.ac.uk/fetal-immune/lymphoid/). The mean expression of *Tcf7* and *Lef1* in the cell types of the above-mentioned lineages are represented as dot plots (Fig. 3h and Extended Data Fig. 6f). The B-1 cells were classified into four clusters based on the expression of LEF1 and TCF7. The total B-1 cells expressing LEF1 exclusively is 463, TCF7 exclusively is 325, both is 73 and neither LEF1 nor TCF7 is 4,789 (Extended Data Fig. 6e). All the analyses were performed using scanpy (v1.9.8).

## ChIP–seq analysis

FASTQ files were downloaded from the NCBI's Gene Expression Omnibus database for TCF1 and LEF1 chromatin immunoprecipitation followed by sequencing (ChIP–seq) data (SRP142342)[42]. FASTQ files were aligned to Enembl's mouse GRCm38 genome using BWA (v0.7.15). The resulting BAM files were sorted, duplicates marked and indexed using Picard (v2.1.1). Peaks were called using MAC2 (v2.1.1) that were enriched in TCF1 or LEF1 relative to input using default parameters. Peaks were annotated using HOMER (v4.8). BAM files were normalized to 10 million reads, and IGVTools (v2.3.75) was used to generate coverage files.

## ATAC-seq analysis

FASTQ files were downloaded from the NCBI's Gene Expression Omnibus database for ATAC-seq samples in B-1a cells (GSM2461745)[17]. FASTQ files were aligned to Ensembl's mouse GRCm38 genome using BWA (v0.7.15). The resulting BAM files were sorted, duplicates marked and indexed using Picard (v2.1.1). IGVTools (v2.3.75) was used to generate coverage file for visualization.

## In vitro stimulation

Cells were cultured in RPMI 1640 medium supplemented with 10% FBS, 2 mM L-glutamine, 100 U penicillin–streptomycin, 0.1 mM non-essential amino acids, 100 mM HEPES and 55 µM 2-mercaptoethanol at 37 °C in

5% $CO_2$. B-1 cells were magnetically purified from the peritoneal cavity using the Pan B cell isolation kit II (130-104-443, Miltenyi Biotec) with anti-mouse CD45R (B220) antibody (130-110-707, Miltenyi Biotec) and labelled with CellTrace Violet (C34557, Thermo Fisher) followed by stimulating with or without 5 µg ml$^{-1}$ LPS (O111:B4; L4391, Sigma), 0.4 µg ml$^{-1}$ IL-10 (210-10-10UG, Thermo Fisher), 10 µg ml$^{-1}$ InVivoMAb anti-mouse IL-10R (CD210; BE0050, BioXCell) and its isotype control, InVivoMAb rat IgG1 isotype control, anti-horseradish peroxidase (HRP; BE0088, BioXCell) for 72 h.

## Cytometric bead array

B-1 cells were magnetically purified from the peritoneal cavity using the Pan B cell isolation kit II (130-104-443, Miltenyi Biotec) with anti-mouse CD45R (B220) antibody (130-110-707, Miltenyi Biotec) and cultured in RPMI 1640 medium supplemented with 10% FBS, 2 mM L-glutamine, 100 U penicillin–streptomycin, 0.1 mM non-essential amino acids, 100 mM HEPES, 55 µM 2-mercaptoethanol and 5 µg ml$^{-1}$ LPS (O111:B4; L4391, Sigma) for 3 days at 37 °C in 5% $CO_2$. Supernatant was collected from cell culture, and IL-10 production was measured by the cytometric bead array mouse IL-10 Flex set (BD).

## EAE

EAE was induced by immunization with $MOG_{35-55}$ peptide, emulsified in Complete Freund's adjuvant (Sigma-Aldrich) and pertussis toxin (Sigma-Aldrich). On day 3.5, $2 \times 10^5$ B-1 cells that had been previously activated with R848 (0.1 µg ml$^{-1}$) for 48 h were transferred intravenously. Mice were examined daily and scored using the following scoring system: 0 for no disease, 1 for loss of tail tonicity, 2 for hindleg weakness, 3 for complete hindleg paralysis, 3.5 for complete hindleg paralysis with partial hind body paralysis, 4 for full hindleg and foreleg paralysis, and 5 for moribund or dead animals[71].

## Immunohistochemistry

Sections (3 µm) were staining on the Leica Bond Rx automated staining platform using sequential application of antibodies with Opal TSA fluorophore detection followed by heat stripping of the antibody complex between stainings using ER1 or ER2 antigen retrieval solution (Leica) for 20 min at 95 °C. The following antibodies, secondaries and opal TSA reagents were used: mApple (1:100; STJ140269, St John's Laboratory), CD45R/B220 (1:750; 553086, BD Biosciences), CD19 (1:200; ab245235, Abcam), Immpress HRP Horse anti-goat IgG polymer (MP-7405-50, Vector), streptomycin-HRP (1:500; P0397, Dako), NovoLink Max Polymer (RE7260-CE, Leica) and Opal 520/570/690 (FP1487001KT/FP1488001KT/FP1497001KT, Akoya Biosciences). Slides were counterstained with DAPI and imaged on the PhenoImager HT (Akoya Biosciences).

## Statistical analyses

Statistical methods used in RNA-seq, scRNA-seq and single-cell BCR-seq are described above. Comparison between groups was performed using parametric $t$-test, two-tailed Welch's $t$-test, Mann–Whitney $t$-test, one-way ANOVA with Tukey multiple-comparison test, two-way ANOVA and two-tailed Pearson correlation analysis from GraphPad Prism10 (GraphPad Software). Sample sizes are provided in the figures, and statistically significant differences are indicated as exact $P$ values.

## Reporting summary

Further information on research design is available in the Nature Portfolio Reporting Summary linked to this article.

## Data availability

Data supporting the findings of this study are available within the paper or its supplementary material. Datasets that support the findings of this study are accessible at the following repositories: RNA-seq data (GSE290505), and scRNA-seq and scBCR-seq data (GSE294717 and GSE298030). ChIP–seq and ATAC-seq data were obtained from publicly available datasets (SRP142342 and GSM2461745). Processed and annotated scRNA-seq data of fetal immune cells, specifically the haematopoietic stem and progenitor cells and the B cell lineages in Suo et al.[23] were downloaded from the developmental cell atlas portal (https://developmentcellatlas.cellgeni.sanger.ac.uk/fetal-immune/lymphoid/). Source data are provided with this paper.

## Code availability

No custom code was used in this study.

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

**Acknowledgements** We thank D. Calado, T. Chen and A. Xu for expert advice; D. Schneider-Luftman, M. Llorian Sopenia and Science Technology Platorms for help with next-generation sequencing analysis; S. Varsani-Brown and J. Olsen from Genetic Modification Service for generating the mouse model; Biological Research Facility for animal care; Experimental Histopathology for experimental tissue analysis in the Francis Crick Institute; the Australian Phenomics Facility for animal care; the CAPTURE Consortium, H. Vohra and M. Devoy from the CHASM facility for assistance with FACS sorting; J. Darcy for sample preparation from the National Institute for Health and Care Research Birmingham Biomedical Research Centre; the personnel of the Australian Cancer Research Foundation Biomolecular Resource Facility for Sanger sequencing; and P. Lane for critical reading of our manuscript. This work was supported by the Francis Crick Institute (CC2228), which receives its core funding from Cancer Research UK, the UK Medical Research Council and the Wellcome Trust; and has also received funding by a Royal Society Wolfson Fellowship, a Wellcome Trust Discovery grant to C.G.V., UK Research and Innovation/BCUK (HP MR/X006891/1) to H.M.P., funds from the National Institute of Allergy and Infectious Diseases Division of Intramural Research to L.D.N., and the Chinese Academy of Medical Sciences Innovation Fund for Medical Science, China (grant number 2024-I2M-2-001-1) to N.I.K.

**Author contributions** Q.S. and C.G.V. conceptualized the study. Q.S. and J.A.R. designed the methodology. Q.S. and J.A.R. conducted the formal analysis. Q.S., H.W., J.A.R. and X.M. performed the investigation. N.M.R., A.E., M.E., O.M.D., A.F.F., M.D.K., B.B., I.P., J.C.L., H.-H.X., S.T., L.D.N., M. Haniffa, L.A.G.-S., H.M.P. and N.I.K. provided resources. J.A.R., M. Bosticardo, M. Hodges, M. Battaglia, Z.-P.F., B.J.T., J.P., V.B.M.S., J.C., P.C., A.G., P.F.C., P.G.-F., Y.Z., H.M.P. and N.I.K. curated the data. Q.S. and C.G.V. wrote the original draft of and reviewed and edited the manuscript. Q.S., H.W., X.M., J.A.R., Z.-P.F., P.C., V.B.M.S., P.F.C., P.G.-F. and Y.Z. performed the visualization. Q.S. and C.G.V. supervised the study. Q.S. and C.G.V. provided project administration. C.G.V. acquired funding.

**Funding** Open Access funding provided by The Francis Crick Institute.

**Competing interests** The authors declare no competing interests.

**Additional information**
**Correspondence and requests for materials** should be addressed to Qian Shen or Carola G. Vinuesa.

Extended Figure 1

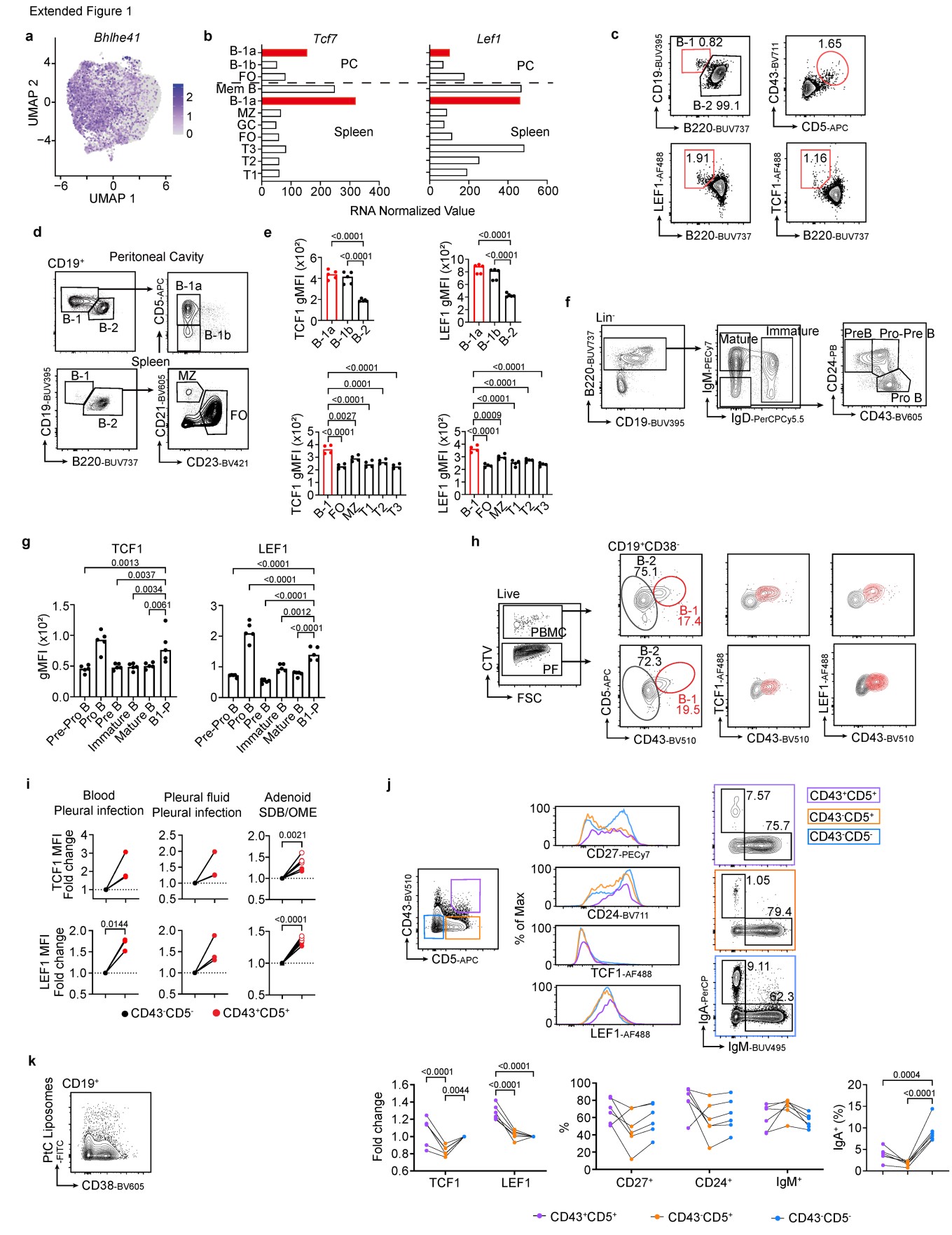

**Extended Data Fig. 1** | See next page for caption.

**Extended Data Fig. 1 | TCF7 and LEF1 expression in mice and human B cells.**
(a) UMAP plots showing the expression of *Bhlhe41* after removing all BCR genes in peritoneal total B cells. (b) Expression of *Tcf7* and *Lef1* across the ImmGen Database. Follicular B cells (FO); memory B cells (Mem B); marginal zone B cells (MZ), germinal centre B cells (GC), transitional 1 (T1), transitional 2 (T2) and transitional 3 (T3) B cells. (c) Flow cytometry plots present TCF1 or LEF1 positive B cells in spleen. (d-e) Gating strategies of B-1a, B-1b and B-2 cells in peritoneal cavity and B-1, B-2, FO and MZ cells in spleen (d) and gMFI quantification of expression of TCF1 and LEF1 in B cell subsets from peritoneal cavity (n = 5) and spleen (n = 4) (e). (f-g) Contour plots (f) and gMFI quantification of TCF1 and LEF1 in B cells subsets from bone marrow (n = 5) (g). (h) Contour plots showing gating strategy and TCF1 and LEF1 expression in CD43$^+$CD5$^+$ B cells from PBMC and pleural fluid (PF) of patient. (i) Fold change of expression of TCF1 and LEF1 between CD43$^+$CD5$^+$ cells and CD43$^-$CD5$^-$ cells in each donor (n = 3). (j) Histogram and contour plots (top) and quantification (bottom) of expression of CD27, CD24, TCF1, LEF1, IgM and IgA in indicated cell subsets. Fold change of expression of TCF1 and LEF1 between CD43$^+$CD5$^{int}$ cells or CD43$^{lo/-}$CD5$^+$ and CD43$^-$CD5$^-$ cells in HD (n = 6). (k). Contour plot showing PtC reactive cells in total B cells from PBMC. Each symbol represents an individual mouse or donor, and bars represent median values. Data are representative of n = 2 experiments. Statistical analysis was performed using one way ANOVA with Tukey multiple-comparison test (e,g), two way ANOVA (j), Paired t test (i). The exact P values are shown.

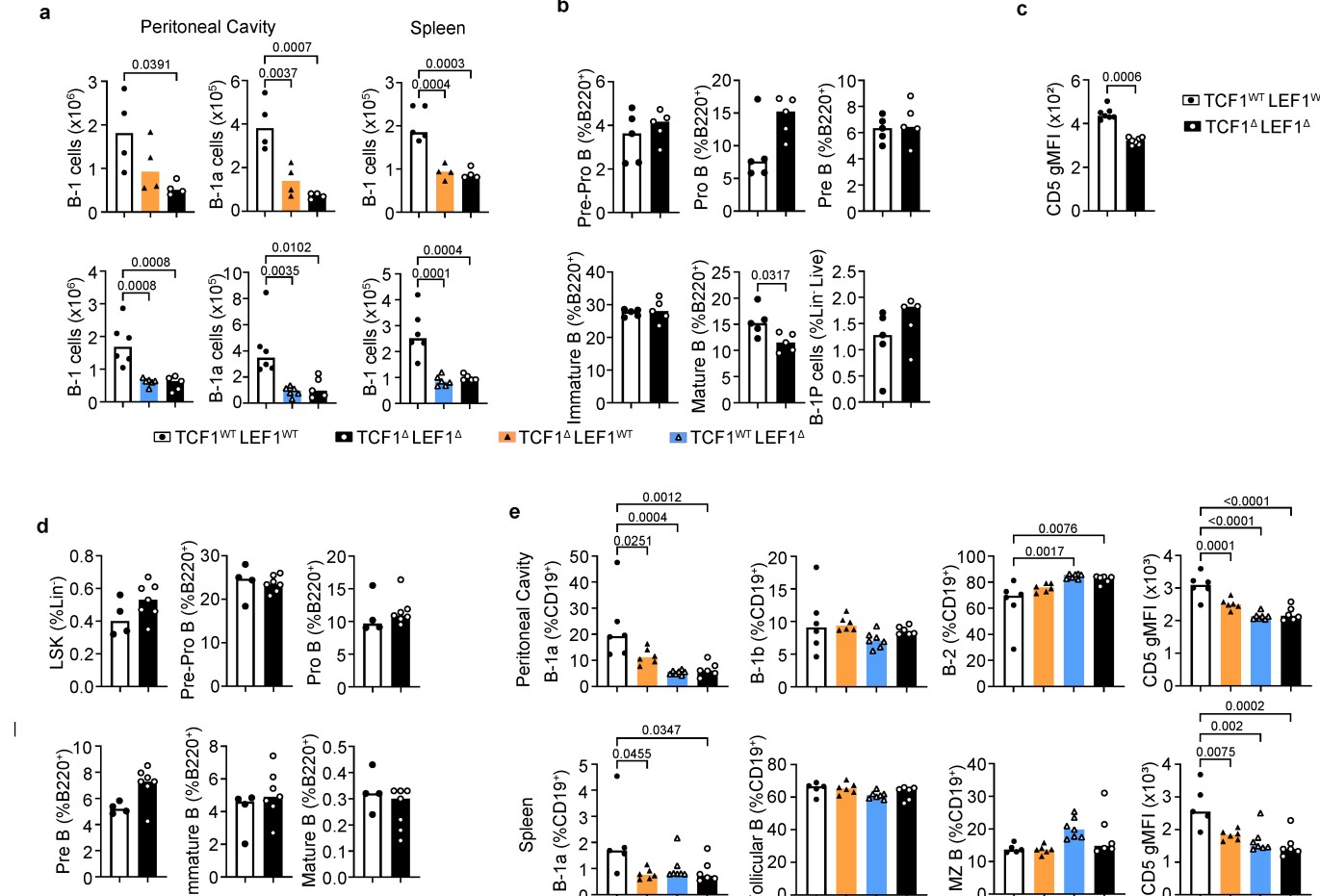

**Extended Data Fig. 2 | TCF1 and LEF1 are required for formation of normal peripheral B-1a numbers.** (a) Quantification of cell numbers of B-1 and B-1a from peritoneal cavity and B-1 cells from spleen in TCF1^WT^LEF1^WT^ (n = 4 or 5), TCF1^Δ^ (n = 4), LEF1^Δ^ (n = 5), TCF1^Δ^LEF1^Δ^ (n = 4 or 5) mice. (b) Quantification of percentage of pre-pro B, pro B, pre-B, immature B, mature B and B-1 progenitors (B-1P) from bone marrow in TCF1^WT^LEF1^WT^ (n = 5) and TCF1^Δ^LEF1^Δ^ (n = 5) mice. (c) gMFI of CD5 expression within B-1a cells in peritoneal cavity between TCF1^WT^LEF1^WT^ (n = 7) and TCF1^Δ^LEF1^Δ^ (n = 7) mice. (d) Frequency of bone marrow (BM) B cell populations from lethally irradiated Rag1^−/−^ recipient mice 6 weeks after transferring foetal liver cells (E14.5) from TCF1^WT^LEF1^WT^ (n = 4) and TCF1^Δ^LEF1^Δ^ (n = 7). (e) Frequency of the indicated peritoneal and splenic B cell populations in sub-lethally irradiated Rag1^−/−^ recipient mice 7 weeks after reconstitution with bone marrow cells from mice with TCF1^WT^LEF1^WT^ (n = 5), TCF1^Δ^ (n = 6), LEF1^Δ^ (n = 7), TCF1^Δ^LEF1^Δ^ (n = 6) mice. Each symbol represents an individual mouse and bars represent median values. Data are representative of n = 3 (a-c), n = 4 (d-e) experiments, Statistical analysis was performed using two-tailed Mann-Whitney U test (b-d), one way ANOVA with Tukey multiple-comparison test (a,e). The exact P values are shown.

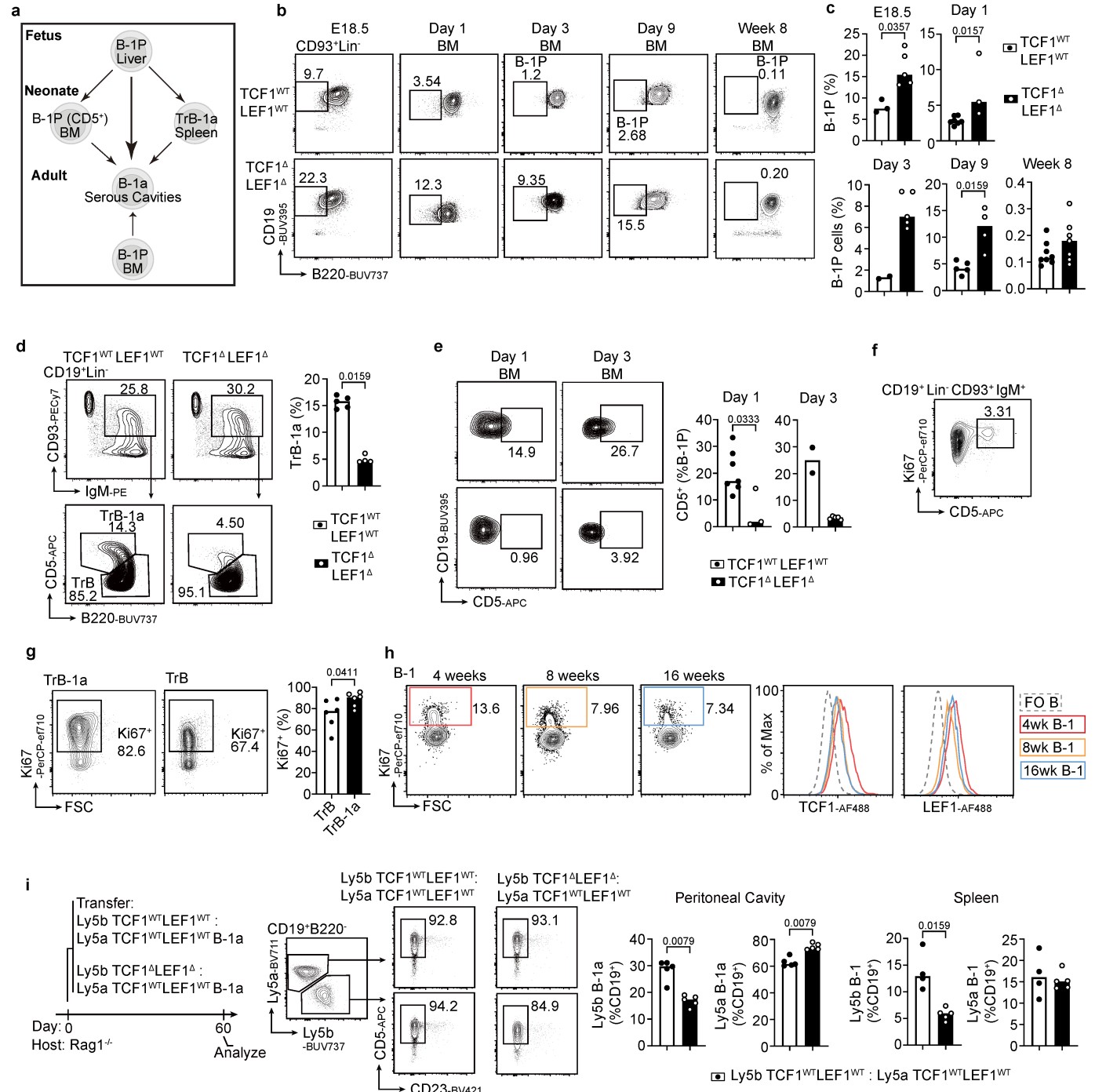

**Extended Data Fig. 3 | TCF1 and LEF1 effect on B-1a development and maintenance.** (a) Diagram depicting key stages of B-1a development. (b-c) Gating strategy (b) and frequency (c) of CD19⁺B220⁻Lin⁻IgM⁻CD93⁺ B-1 progenitors (B-1P) from E18.5 foetal liver cells (TCF1^WT^LEF1^WT^ n = 3; TCF1^Δ^LEF1^Δ^ n = 5) and bone marrow (BM) at postnatal days 1 (TCF1^WT^LEF1^WT^ n = 7; TCF1^Δ^LEF1^Δ^ n = 3), 3 (TCF1^WT^LEF1^WT^ n = 2; TCF1^Δ^LEF1^Δ^ n = 5) and 9 (TCF1^WT^LEF1^WT^ n = 5; TCF1^Δ^LEF1^Δ^ n = 4), and 8 weeks old (TCF1^WT^LEF1^WT^ n = 8; TCF1^Δ^LEF1^Δ^ n = 7). (d) Flow cytometry stains and quantification of TrB-1a identified as CD93⁺IgM⁺ CD19⁺B220^lo^CD5⁺ from spleens of 9-day old mice of TCF1^WT^LEF1^WT^ (n = 5) and TCF1^Δ^LEF1^Δ^ (n = 4). (e) Flow cytometry stains and quantification of the CD5⁺ cells in B-1P from bone marrow at postnatal day 1 (TCF1^WT^LEF1^WT^ n = 7; TCF1^Δ^LEF1^Δ^ n = 3) and 3 (TCF1^WT^LEF1^WT^ n = 2; TCF1^Δ^LEF1^Δ^ n = 5) mice. (f) Gating strategy of CD5⁺ Ki67⁺ cells in CD19⁺ Lin⁻CD93⁺IgM⁺ in TCF1^WT^LEF1^WT^ spleen at neonatal day 9. (g) Ki67⁺ cells within TrB-1a cells and TrB cells as CD93⁺IgM⁺CD19⁺ B220⁺CD5⁻ from spleens of 9-day old mice (n = 6). (h) Flow cytometry plots of Ki67 positive cells and histogram of TCF1 or LEF1 in splenic B-1a cells in 4 weeks old mice (Young, n = 5) and 8-16 weeks old mice (Adult, n = 7). (i) Treatment protocol of mixed sorted B-1a cells of (50% ly5b TCF1^WT^LEF1^WT^ + 50% ly5ba TCF1^WT^LEF1^WT^) and (50% ly5b TCF1^Δ^LEF1^Δ^ + 50% ly5a TCF1^WT^LEF1^WT^) in Rag1^−/−^ recipient mice at 2 months after intraperitoneal injection (left), Flow cytometry plots (middle) and quantification of percentage of B-1a cells in peritoneal cavity (TCF1^WT^LEF1^WT^ n = 5; TCF1^Δ^LEF1^Δ^ n = 5) and spleen (TCF1^WT^LEF1^WT^ n = 4; TCF1^Δ^LEF1^Δ^ n = 5) (right). Each symbol represents an individual mouse and bars represent median values. Data are representative of n = 2 experiments (c-e,g-i). Statistical analysis was performed using two-tailed Mann-Whitney t test (c-e,g-i). The exact P values are shown.

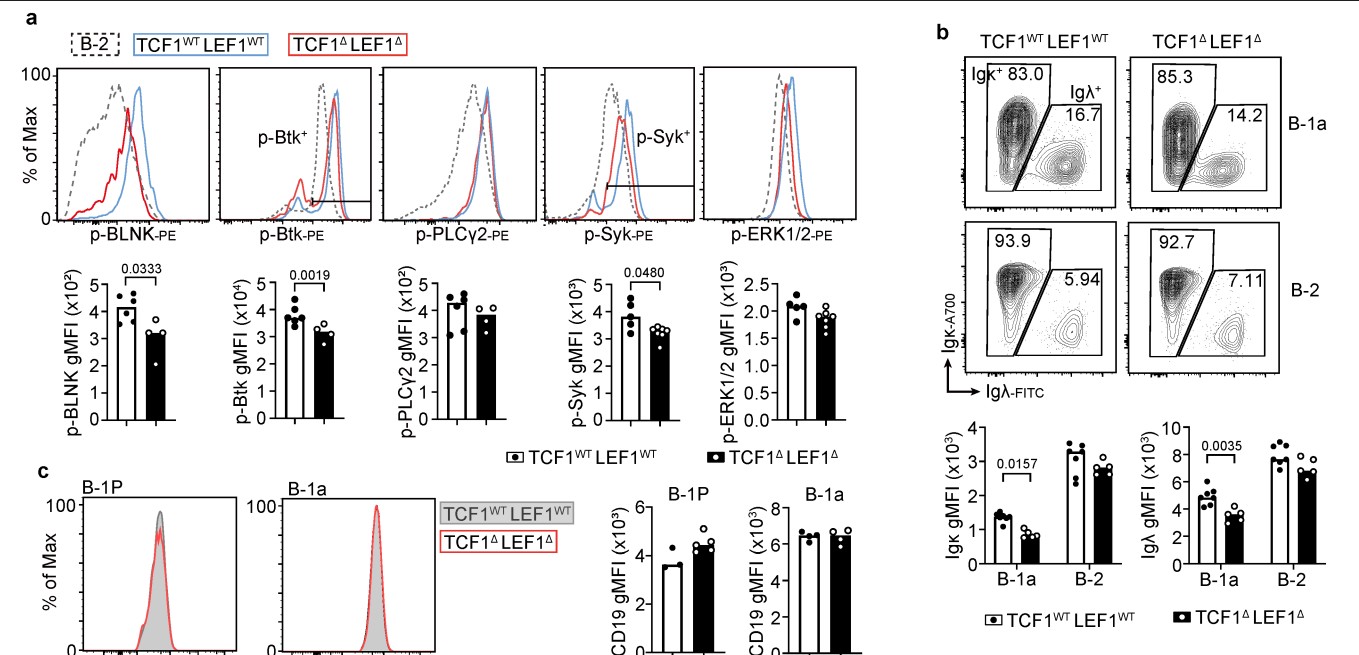

**Extended Data Fig. 4 | TCF1 and LEF1 effect on BCR signalling of B-1 cells.** (a) Histogram (top) and gMFI quantification (bottom) of phosphorylation (p-) status of various BCR signaling components: p-BLNK (TCF1$^{WT}$LEF1$^{WT}$ n = 6; TCF1$^{\Delta}$LEF1$^{\Delta}$ n = 4); p-Btk (TCF1$^{WT}$LEF1$^{WT}$ n = 6; TCF1$^{\Delta}$LEF1$^{\Delta}$ n = 4); p-PLCγ2 (TCF1$^{WT}$LEF1$^{WT}$ n = 6; TCF1$^{\Delta}$LEF1$^{\Delta}$ n = 4); p-Syk (TCF1$^{WT}$LEF1$^{WT}$ n = 5; TCF1$^{\Delta}$LEF1$^{\Delta}$ n = 7); p-ERK1/2 (TCF1$^{WT}$LEF1$^{WT}$ n = 5; TCF1$^{\Delta}$LEF1$^{\Delta}$ n = 7) in B-1a cells. (b) Flow cytometry plots (top) and gMFI (bottom) of Igκ and Igλ in B-1a and B-2 cells

from TCF1$^{WT}$LEF1$^{WT}$ (n = 7) and TCF1$^{\Delta}$LEF1$^{\Delta}$ (n = 5). (c) Histogram (left) and gMFI (right) of CD19 in B-1P from foetal liver (TCF1$^{WT}$LEF1$^{WT}$ n = 3; TCF1$^{\Delta}$LEF1$^{\Delta}$ n = 5) and B-1a cells (TCF1$^{WT}$LEF1$^{WT}$ n = 4; TCF1$^{\Delta}$LEF1$^{\Delta}$ n = 4) from peritoneal cavity. Each symbol represents an individual mouse and bars represent median values. Data are representative of n = 2 experiments (a-c). Statistical analysis was performed using two-tailed Mann-Whitney t test (a,c), one way ANOVA with Tukey multiple-comparison test (b). The exact P values are shown.

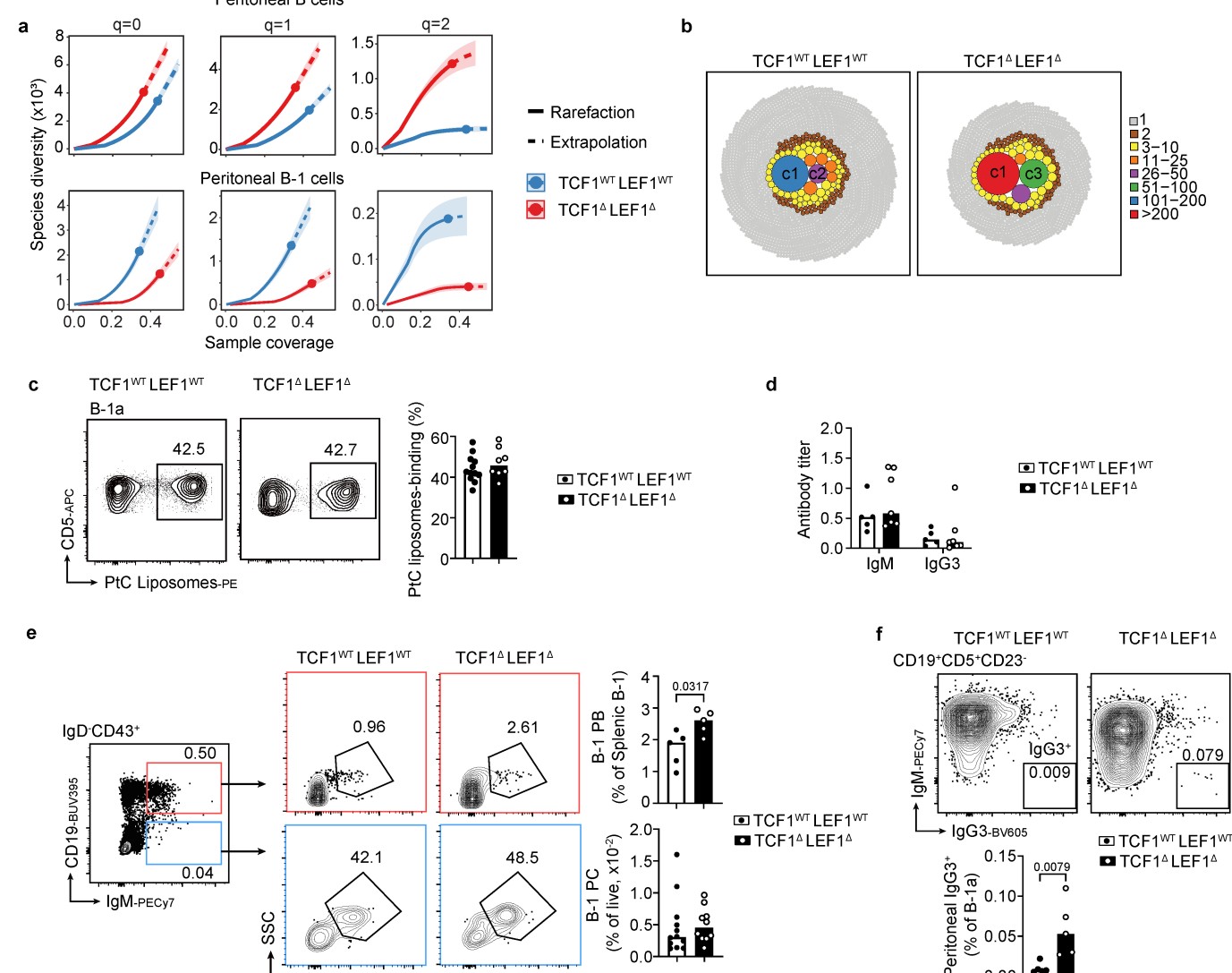

**Extended Data Fig. 5 | TCF1 and LEF1 effect on specificity and clonotype of B-1 cells.** (a) Coverage-based diversity accumulation curve for Hill numbers of order (q = 0, 1 and 2), i.e., species richness, Shannon entropy and Simpson index, respectively[72] in peritoneal B cell (top) or B-1 cells (bottom). (b) Circular plot of cell number of different clonotypes in TCF1[WT]LEF1[WT] and TCF1[Δ]LEF1[Δ] peritoneal B-1 cells. (c) Flow cytometry plots and quantification of frequency of PtC[+]B-1a cells in TCF1[WT]LEF1[WT] (n = 12) and TCF1[Δ]LEF1[Δ] (n = 8) mice. (d) Total serum IgM and IgG3 were determined by ELISA between TCF1[WT]LEF1[WT] (n = 5) and TCF1[Δ]LEF1[Δ] (n = 7) mice. (e) Flow cytometry plots and quantification of percentage of spleen B-1 cell derived plasma cells (B-1 PC) (Dump[−]CD19[−]CD43[+] IgM[+]IgD[−/lo]CD138[+]) (TCF1[WT]LEF1[WT] n = 5; TCF1[Δ]LEF1[Δ] n = 5) and B-1 plasmablasts (B-1 PB) (Dump[−]CD19[+]CD43[+]IgM[+]IgD[−/lo]CD138[+]) (TCF1[WT]LEF1[WT] n = 12; TCF1[Δ]LEF1[Δ] n = 10) in B-1 cells. (f) Flow cytometry plots and quantification of percentage of IgG3[+] from peritoneal B-1a cells in TCF1[WT]LEF1[WT] (n = 5) and TCF1[Δ]LEF1[Δ] (n = 5) mice. Each symbol represents an individual mouse and bars represent median values. Data from n = 3 experiments in total (c, e), or representative of n = 3 experiments (d-f). The lines represent the diversity estimates. The shaded area around the curve represents the 95% confidence intervals (95% CI) (a). Statistical analysis was performed using two-tailed Mann-Whitney U test (c,e-f), one way ANOVA with Tukey multiple-comparison test (d). The exact P values are shown.

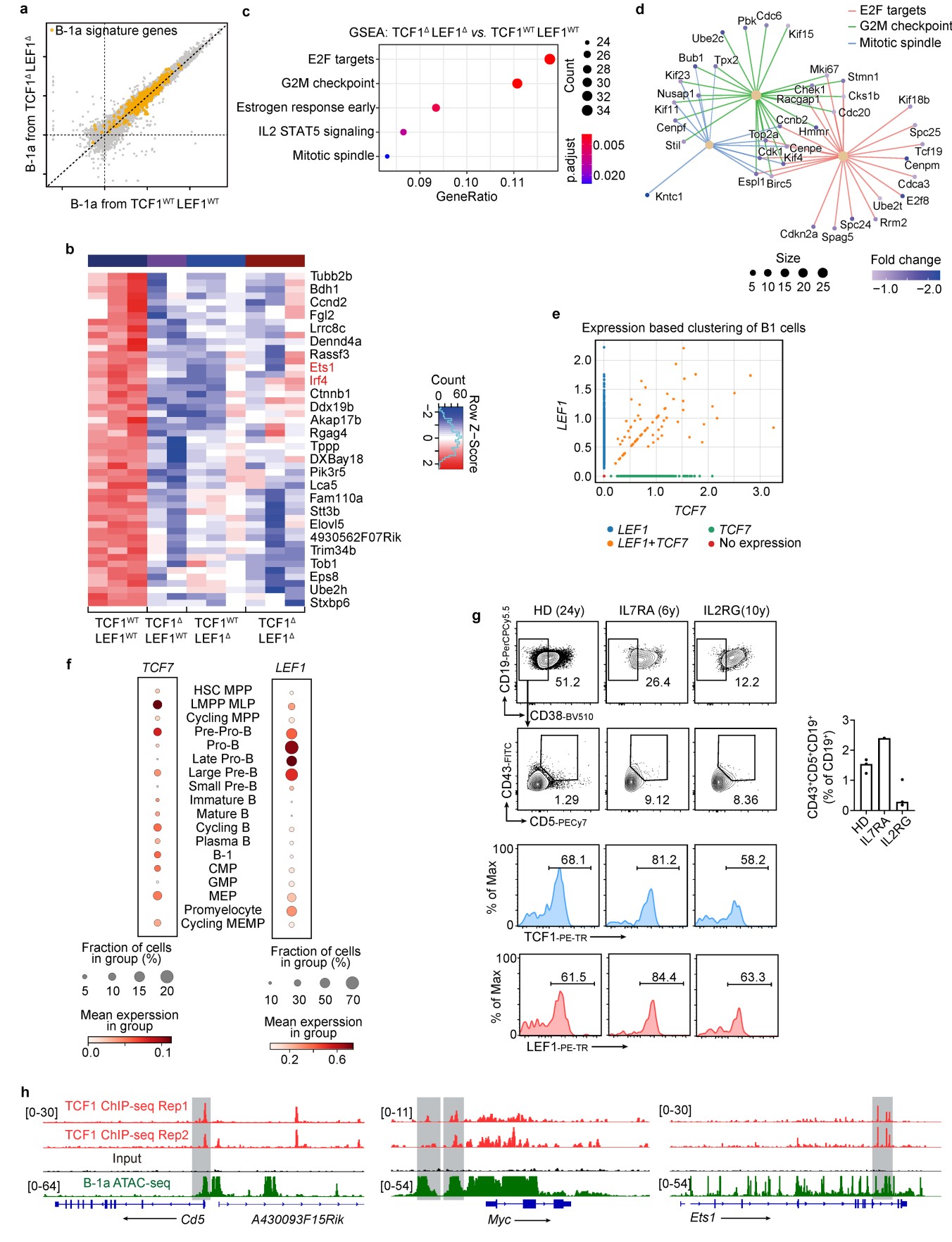

**Extended Data Fig. 6 | Signalling pathways of TCF1 and LEF1 on mice and human B-1 cells.** (a) Expression-expression plot of transcriptomes of B-1a cells isolated from TCF1$^{WT}$LEF1$^{WT}$ and TCF1$^{\Delta}$LEF1$^{\Delta}$ mice. Superimposed B-1a signature genes from published RNA-seq data which over- or under-represented in peritoneal B-1a versus B-1b cells[34]. (b) Expression heatmap of B-1a signature genes, with the colour representing normalized RPM for each gene across different samples. (c) Dot plot presentation of Hallmark gene sets according to GSEA for differentially expressed genes in TCF1$^{WT}$LEF1$^{WT}$ and TCF1$^{\Delta}$LEF1$^{\Delta}$ B-1a cells. (d) Cnet plot presentation of enriched Hallmark pathways (P < 0.05, FDR < 0.05) upon GSEA. Size = the number of differentially expressed gene which belongs to the enriched Hallmark pathway. Fold change difference between TCF1$^{WT}$LEF1$^{WT}$ and TCF1$^{\Delta}$LEF1$^{\Delta}$ B-1a cells is shown. (e) Frequency of TCF1 and LEF1 expressed cells in human B-1 clusters[23]. (f) Dot plot representation of *Tcf7* and *Lef1* mRNA expression in different subsets of human prenatal tissues. Panels **e,f** were adapted from ref. 23, AAAS. (g) Flow cytometry plots and quantification of B-1-enriched cells from PBMCs from 4 SCID patients and 3 HD. (h) ChIP-seq analysis of TCF1 binding in CD4$^+$CD8$^+$ thymocytes[42] and ATAC-seq analysis at *Cd5*, *Myc* and *Ets1* on B-1a cells from *Vh12/Vk4* transgenic mice[17]. Panel **h** was adapted from refs. 17,42, Springer Nature Ltd. The significant hallmark gene sets in MSigDB (adjustment p-value < 0.05) for each contrast based on the hypergeometric testing using cluster profiler package (c).

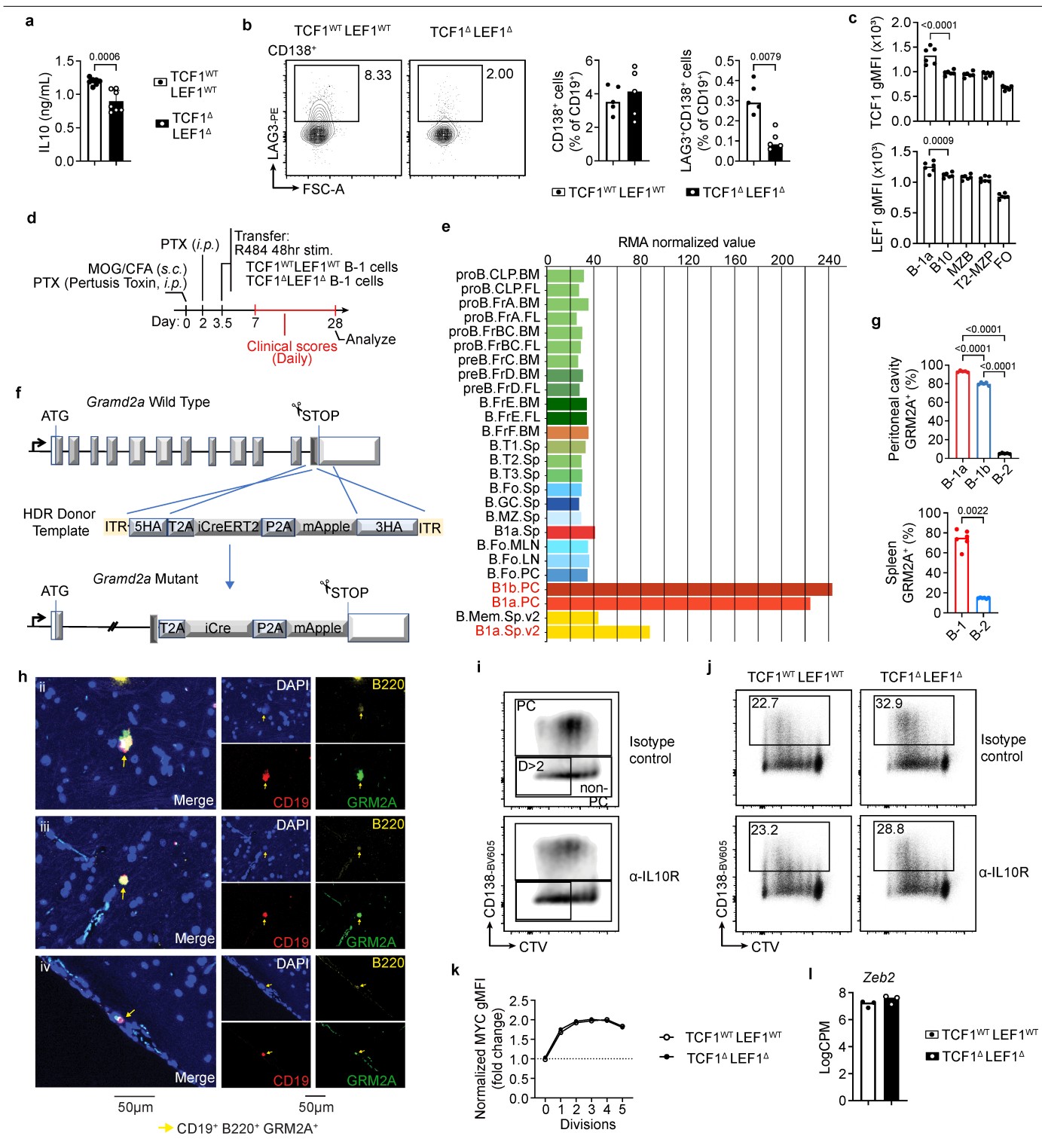

**Extended Data Fig. 7 |** See next page for caption.

**Extended Data Fig. 7 | TCF1 and LEF1 effect on regulatory function of B-1 cells.** (a) IL-10 secretion measured by cytometric bead Arrays (CBA) from TCF1$^{WT}$LEF1$^{WT}$ (n = 7) and TCF1$^{\Delta}$LEF1$^{\Delta}$ (n = 7) B-1 cells cultured in media with LPS for 3 days. (b) Flow cytometry plots and quantification of percentage of CD138$^+$Lag3$^+$ cells in the spleen of TCF1$^{WT}$LEF1$^{WT}$ (n = 5) or TCF1$^{\Delta}$LEF1$^{\Delta}$ (n = 5) mice 3 days after injection with LPS. (c) gMFI quantification of TCF1 and LEF1 in B cells subsets from B-1a (CD19$^+$B220$^-$CD5$^+$CD23$^-$), B10 (CD1d$^{hi}$CD5$^+$), immature transitional-2 (T2-MZP) (B220$^+$IgM$^{hi}$CD21$^{hi}$CD23$^+$), marginal zone B cells (MZB) (CD19$^+$B220$^+$CD23$^-$CD21$^+$), Follicular B cells (FO) (CD19$^+$B220$^+$CD23$^+$CD21$^-$) from spleen of C57BL/6J mice (n = 6). (d) Flowchart of EAE immunization. (e) Expression of *Gramd2a* across the ImmGen Database. The figure was created using ImmGen (https://immgen.org). (f) Schematics of *Gramd2a*-T2A-iCre-P2A-mApple strain. (g) Frequency of GRM2A positive cells within B-1a, B-1b and B-2 populations from peritoneal cavity (n = 5) and B-1 and B-2 cells from spleen (n = 6). (h) Identification of GRM2A positive B-1 cells in EAE mouse brain. representative cells from each labelled areas from Fig. 4o (i) Flow cytometry plots of plasma cell (PC) (CD138$^+$) and non-plasma cell (CD138$^-$) cells from purified peritoneal cavity TCF1$^{WT}$LEF1$^{WT}$ B-1 cells cultured with LPS and isotype control or anti-mouse IL-10R for 3 days. (j) Flow cytometry plots of CD138$^+$ cells after magnetically purified splenic TCF1$^{WT}$LEF1$^{WT}$ or TCF1$^{\Delta}$LEF1$^{\Delta}$ B-2 cells were cultured with LPS and anti-mouse IL-10R or isotype control for 3 days. (k) Quantification of gMFI of MYC in each division of B-2 cells stimulated with LPS for 3 days. (l) Quantification of *Zeb2* expression in TCF1$^{WT}$LEF1$^{WT}$ and TCF1$^{\Delta}$LEF1$^{\Delta}$ B-1a cells by RNA-seq. Each symbol represents an individual mouse, Data are presented as mean valued +/− S.D. Data are representative of n = 2 experiments (b-c,g), peritoneal cavity cells from three mice are pooled together (a,k). Images are representative of n = 3 mice (l). Statistical analysis was performed using two-tailed Mann-Whitney U test (a-b,l) and one way ANOVA (c,g). The exact P values are shown.

**Extended Data Table 1 | Clinical characteristics for each of the patients with pleural infection**

| GEN | Donor ID | Age | Sex | Urea (serum, mmol/L) | Albumin (serum, g/L) | CRP (serum, mg/L) | LDH (fluid, int Unit/L) | Glucose (fluid, mmol/L) | Antibiotics | RAPID Score |
|---|---|---|---|---|---|---|---|---|---|---|
| 12230 | PI-001 | 55 | F | 24 | 3.1 | 385.5 | 687 | 3.0 | IV Co-amoxiclav and Oral Metronidazole | Medium |
| 12402 | PI-002 | 67 | F | 4.3 | 18 | 471.1 | 1,738 | <0.28 | IV Co-amoxiclav and Oral Metronidazole | Low |
| 12429 | PI-003 | 78 | F | 4.2 | 23 | 73.7 | 255 | 2.5 | Oral courses of Amoxicillin and Doxycycline, then Co-amoxiclav IV Ceftriaxone, Co-amoxiclav and Metronidazole | Medium |
| 12430 | PI-004 | 76 | M | 5.7 | 35 | 60.4 | 442 | <0.28 | Oral courses of Amoxicillin and Doxycycline | Medium |

The table presents the clinical characteristics of each of the donors who provided specimens for the study. All donors had confirmed bacterial pleural infection. Sex M = Male, F = Female, CRP = C-reactive protein, LDH = Lactate dehydrogenase, IV = Intravenously.

**Extended Data Table 2 | Clinical characteristics for each of the CLL patients**

| Donor ID | Sex | WCC (x10⁹/L) | NGS TP53 | FISH (13q- / 17p-/11q- /t12) | IGHV (Unmutated/mutated) | Binet Stage | Treated CLL (Y/N) |
|---|---|---|---|---|---|---|---|
| 1 | M | 19.2 | WT | Normal | Unmutated | A | N |
| 2 | M | 22.8 | Unknown | Normal | Mutated | A | N |
| 3 | M | 11.39 | WT | 13q- | Unmutated | B | N |
| 4 | F | 25.8 | Unknown | Unknown | Unknown | A | N |
| 5 | M | Unknown | WT | 13q- | Mutated | A | N |
| 6 | F | 12.2 | Mutated | Normal | Unmutated | B | N |
| 7 | M | 29.6 | WT | Normal | Mutated | A | N |
| 8 | M | 18.56 | WT | Normal | Unknown | A | N |
| 9 | M | 19.6 | Not Checked | Normal | Unknown | A | N |
| 10 | F | 68.6 | Not Checked | Normal | Unknown | A | N |
| 11 | M | 22 | WT | Normal | Mutated | A | N |

**Extended Data Table 3 | Clinical characteristics for each of the SCID patients**

| Donor ID | Sex | Age | Mutation | Clinic/LAB |
|---|---|---|---|---|
| IL7RA_3 | F | 6y | IL7R (NM_002185) c.82G>A p.G28R | recurrent respiratoryu tract infections, BCGosis, varicella. CD3 305, CD4 70; CD8 145; CD19 80; CD16/56 333 |
| IL2RG_1 | M | 38y | IL2RG c.206T>C, p.L87P with revertants | diagnosed as CVID at 5v y; granulomatous disease, recurrent respiratory tract infections with bronchiectasis, osteomyelitis, CD3 504, CD4 150, CD8 243, CD19 19, NK 17 |
| IL2RG_2 | M | 14mo | IL2RG (NM_000206) c.720G>C p.W240C | leaky SCID, recurrent infections, chronic siarrhea, thrush, CD3 490, CD4 44 (99.7% R0), CD8 400 (94.5% memory), CD19 989, CD16/56 240 |
| IL2RG_3 | M | 10y | IL2RG c568C>T, pArg190Trp (mosaic, 20% of reads, de novo); IL2RG c.589T>C, p.Trp197Arg (mosaic, 85% of reads, maternally inherited) | mosaic XSCID, bronchiectasis, recurrent infections, s/p MMRD HSCT complicated by adenoviremia, now A&W post HSCT. Initial flow (age 8 yrs): CD3 3241 cells/mcl, CD4 753, CD8 2127, B. cells 979, NK 769, CD4/CD45RA 12%, CD4/CD45RO 88% |

# Reporting Summary

## Statistics

For all statistical analyses, confirm that the following items are present in the figure legend, table legend, main text, or Methods section.

| n/a | Confirmed | |
|---|---|---|
| ☐ | ☒ | The exact sample size (*n*) for each experimental group/condition, given as a discrete number and unit of measurement |
| ☐ | ☒ | A statement on whether measurements were taken from distinct samples or whether the same sample was measured repeatedly |
| ☐ | ☒ | The statistical test(s) used AND whether they are one- or two-sided <br> *Only common tests should be described solely by name; describe more complex techniques in the Methods section.* |
| ☐ | ☒ | A description of all covariates tested |
| ☐ | ☒ | A description of any assumptions or corrections, such as tests of normality and adjustment for multiple comparisons |
| ☐ | ☒ | A full description of the statistical parameters including central tendency (e.g. means) or other basic estimates (e.g. regression coefficient) AND variation (e.g. standard deviation) or associated estimates of uncertainty (e.g. confidence intervals) |
| ☐ | ☒ | For null hypothesis testing, the test statistic (e.g. $F$, $t$, $r$) with confidence intervals, effect sizes, degrees of freedom and $P$ value noted <br> *Give P values as exact values whenever suitable.* |
| ☒ | ☐ | For Bayesian analysis, information on the choice of priors and Markov chain Monte Carlo settings |
| ☒ | ☐ | For hierarchical and complex designs, identification of the appropriate level for tests and full reporting of outcomes |
| ☐ | ☒ | Estimates of effect sizes (e.g. Cohen's *d*, Pearson's *r*), indicating how they were calculated |

*Our web collection on statistics for biologists contains articles on many of the points above.*

## Software and code

Policy information about availability of computer code

**Data collection**   TCF1 and LEF1 RNA expression in mouse B cell subsets were collected from ImmGen. Cellular phenotype data was collected on a Fortessa, Fortessa X-20, LSR II or FACSAria II and software (CellQuest and FACSDiva version 8.0 and 9.2; BD). ELISA data was collected by using an Infinite® 200 PRO plate reader (Tecan) equipped with i-control™ version 1.9 software. RNA-seq data was collected on a HiSeq2000 machine with a depth of more than 30 million reads per samples. scRNA-seq data was collected using the NovaSeq 6000 (Illumina) system or NovaSeq S2. scRNA-seq on human prenatal B-1 were downloaded from the developmental cell atlas portal (https://developmentcellatlas.cellgeni.sanger.ac.uk/fetal-immune/lymphoid/). ChIP-seq data were downloaded from NCBI's GEO database for TCF1 and LEF1 CHIP-seq data (SRP142342).ATAC-seq data were downloaded from NCBI's GEO database for ATAC-seq samples in B-1a cells (GSM2461745). Immunohistochemistry images were collected on the Phenolmager HT (Akoya Biosciences).

**Data analysis**   Comparison between groups was performed using parametric t test, two-tailed Welch's t test or Mann-Whitney t test, one way ANOVA with Tukey multiple-comparison test, two-way ANOVA, two-tailed Pearson correlation analysis from GraphPad Prism10 (GraphPad Software, USA). Sample sizes are provided in the figures and statistically significant differences are indicated as exact P value.  Flow cytometry data was analysed using the Flowio software v10 (Flowio LLC). RNA-seq analysis were aligned to the mm10 (GRC38) genome assembly using hisat2 and the mapped reads were assigned with FeatureCounts [v2.4].  Differential expression analyses were performed with voom-limma, after removal of lowly expressed genes and normalized using the trimmed mean of M-values method.  Significantly differentially expressed genes were identified by applying a Benjamini–Hochberg adjusted P value threshold of 0.05. Gene set enrichment or pathway analysis were performed using clusterProfiler and Camera against the gene ontology (GO) database, KEGG database and HALLMARK, C2 and C7 gene sets in the MSigDB (v7.5). scRNA-seq data was analysed using Cell Ranger pipeline v.6.0.1 or v7.0.1. Data were processed and analysed using the Seurat package (v5.0.3) in R (v4.4.1). Raw gene expression matrices were filtered to exclude low-quality cells based on thresholds for mitochondrial gene content (set at 5%) and the number of detected genes per cell, using the PercentageFeatureSet() and subset() functions. For multiplexed samples labelled with cell hashing antibodies, demultiplexing was performed using the MULTIseqDemux() function in Seurat.

nature portfolio | reporting summary

Hashtag oligonucleotide (HTO) counts were normalised using centred log-ratio (CLR) transformation via the NormalizeData() function with normalisation.method = "CLR", and cells were classified as singlets, doublets, or negatives based on HTO signal. Only singlets were retained for downstream analysis. Normalisation and integration were performed using the scTransform() workflow, followed by the identification of highly variable features. Dimensionality reduction was performed using principal component analysis (PCA) via the RunPCA() function, and the top 20–30 principal components were selected to construct a shared nearest neighbour (SNN) graph using FindNeighbours(). Clustering was conducted using the FindClusters() function. Clusters were visualised using Uniform Manifold Approximation and Projection (UMAP) via the RunUMAP() function. Differential expressed genes for each cluster were identified using the FindAllMarkers() function with test.use set as either "wilcox" or "MAST". Where applicable, cell types were annotated based on canonical marker expression or informed by publicly available datasets. Cell trajectory analysis was performed using Monocle2 v2.32.0 in R v4.4.1. The annotated Seurat object was first transformed into a CellDataSet object using the as.CellDataSet() function from the Seurat package. To filter relevant genes, those expressed in at least 10 cells and with a mean expression value of 0.1 or greater were selected. Next, highly variable genes across cell subsets were identified using the differentialGeneTest() function with the formula fullModelFormulaStr = '~cell_subset'. Based on their q-value, the top 2000 genes were selected, and setOrderingFilter() was applied to prioritize them for ordering in pseudo time analysis. Dimensionality reduction was performed using reduceDimension() with the parameter reduction_method = 'DDRTree'. Cells were subsequently ordered along the trajectory using the orderCells() function and visualized with plot_cell_trajectory().B-cell receptor (BCR) repertoire analysis was conducted with the Immcantation framework, following the guidelines provided by its developers (http://immcantation.org/). This was implemented in a Python environment (v.3.11.10). We began with 10x Genomics BCR sequencing data processed through the CellRanger pipeline (v7.0.1) to obtain annotated clonotype and contig sequences. For V(D)J gene assignment, the AssignGenes.py command from the Change-O toolkit was employed, referencing the mouse IMGT and IgBLAST (v1.22.0) databases. MakeDb.py was used to standardize the data to AIRR format. Clonotypes were assigned using the DefineClones.py tool based on heavy chain sequences, employing the default Hamming distance substitution model. Subsequently, light_cluster.py was applied to refine clonotype grouping by incorporating light chain information. Finally, the resulting datasets were merged with the annotated Seurat file and key metrics, including V(D)J gene usage, CDR and FWR sequences, lengths, and mutational content were extracted for downstream analysis and visualization.Cell trajectory analysis was performed using Monocle2 v2.32.0 in R v4.4.1. The annotated Seurat object was first transformed into a CellDataSet object using the as.CellDataSet() function from the Seurat package. To filter relevant genes, those expressed in at least 10 cells and with a mean expression value of 0.1 or greater were selected. Next, highly variable genes across cell subsets were identified using the differentialGeneTest() function with the formula fullModelFormulaStr = '~cell_subset'. Based on their q-value, the top 2000 genes were selected, and setOrderingFilter() was applied to prioritize them for ordering in pseudo time analysis. Dimensionality reduction was performed using reduceDimension() with the parameter reduction_method = 'DDRTree'. Cells were subsequently ordered along the trajectory using the orderCells() function and visualized with plot_cell_trajectory(). B-cell receptor (BCR) repertoire analysis was conducted with the Immcantation framework, following the guidelines provided by its developers (http://immcantation.org/). This was implemented in a Python environment (v.3.11.10). We began with 10x Genomics BCR sequencing data processed through the CellRanger pipeline (v7.0.1) to obtain annotated clonotype and contig sequences. For V(D)J gene assignment, the AssignGenes.py command from the Change-O toolkit was employed, referencing the mouse IMGT and IgBLAST (v1.22.0) databases. MakeDb.py was used to standardize the data to AIRR format. Cells expressing more than one heavy chain or only light chains were removed. Clonotypes were assigned using the DefineClones.py tool based on heavy chain sequences, employing the default Hamming distance substitution model. Subsequently, light_cluster.py was applied to refine clonotype grouping by incorporating light chain information. Finally, the resulting datasets were merged with the annotated Seurat file and key metrics, including V(D)J gene usage, CDR and FWR sequences, lengths, and mutational content were extracted for downstream analysis and visualization.For diversity analysis, we used the iNEXT package (v3.0.1) to perform coverage-based rarefaction and extrapolation analyses. We computed the three most common Hill numbers: species richness (q = 0), Shannon diversity (q = 1) and Simpson diversity (q = 2). The resulting sampling curves, which illustrate diversity estimates with respect to sample coverage, were visualized using ggiNEXT() function. Additionally, we constructed circle packing plots using the packcircles package (v0.3.6) to visualize the immune repertoire composition of individual samples. These analyses were conducted in R v4.4.1.FASTQ files were aligned to Enembl's mouse GRCm38 genome using BWA version 0.7.15. The resulting BAM files were sorted, duplicates marked and indexed using Picard version 2.1.1. Peaks were called using MAC2 version 2.1.1 that were enriched in TCF1 or LEF1 relative to input using default parameters. Peaks were annotated using homer version 4.8. BAM files were normalised to 10 million reads and IGVTools version 2.3.75 was used to generate coverage files. Slides were counterstained with DAPI and imaged on the PhenoImager HT (Akoya Biosciences).

For manuscripts utilizing custom algorithms or software that are central to the research but not yet described in published literature, software must be made available to editors and reviewers. We strongly encourage code deposition in a community repository (e.g. GitHub). See the Nature Portfolio guidelines for submitting code & software for further information.

# Data

Policy information about availability of data

All manuscripts must include a data availability statement. This statement should provide the following information, where applicable:
- Accession codes, unique identifiers, or web links for publicly available datasets
- A description of any restrictions on data availability
- For clinical datasets or third party data, please ensure that the statement adheres to our policy

The authors declare that data supporting the findings of this study are available within the paper or its supplementary information. Datasets that support the findings of this study are accessible at the following repositories: RNA-seq data (GSE290505), scRNA-seq and scBCR-seq data (GSE294717 and GSE298030). ChIP-seq and ATAC-seq data are obtained from publicly available dataset (SRP142342 and GSM2461745). Processed and annotated single cell RNA sequencing data of foetal immune cells, specifically the HSPCs and the B cell lineages, in Suo et al, Science 2022 23 were downloaded from the developmental cell atlas portal (https://developmentcellatlas.cellgeni.sanger.ac.uk/fetal-immune/lymphoid/).

# Research involving human participants, their data, or biological material

Policy information about studies with human participants or human data. See also policy information about sex, gender (identity/presentation), and sexual orientation and race, ethnicity and racism.

| Reporting on sex and gender | The study involves both male and female participants. |
| --- | --- |

April 2023

| Reporting on race, ethnicity, or other socially relevant groupings | This study does not include any variables on race, ethnicity, or other socially relevant groups. |
|---|---|
| Population characteristics | Individuals were either healthy controls, or patients who were diagnosed with bacterial pleural infection, chronic lymphocytic leukaemia (CLL), severe combined immunodeficiency (SCID), Otitis media with effusion (OME) and sleep disordered breathing (SDB) by treating physicians Individuals known medical treatments and clinical diagnosis are provided in Extended Table 1-3 |
| Recruitment | Participants were recruited by their referring medical practitioners:<br>1.Pleural infection specimens: Patients with confirmed bacterial pleural infection were invited to donate samples (Extended Table 1); all patients had pleural neutrophil counts > 10 million/L and neutrophil/lymphocyte ratio (NLR) >4 and were on antibiotic treatment. All clinical specimens (pleural fluid and blood) were collected for Oxford Radcliffe Pleural Biobank .<br>2. CLL patients and healthy donors were also recruited following favourable study approval on 04Mar2024 by the London-Brent Research Ethics Committee and as part of the CAPTURE (NCT03226886) study, a prospective longitudinal cohort study investigating Covid vaccine responses in cancer patients (Extended Table 2).<br>3.SCID patients and healthy donors were obtained by by the Institutional Review Boards of Children's National Hospital (Washington DC) and of the National Institute of Health (Bethesda, MD) .<br>4.Adenoid tissue samples were collected from children (aged 2 — 6 years, 2 females and 4 males) undergoing elective adenoidectomy surgery for either otitis media with effusion (OME) or sleep disordered breathing (SDB) at the Great North Children's Hospital, Newcastle Upon Tyne, UK. |
| Ethics oversight | The study was approved by and complies with all relevant ethical regulations of:<br>1. All clinical specimens (pleural fluid and blood) were collected for Oxford Radcliffe Pleural Biobank (Ethical approval reference: 19/SC/0173, South Central - Oxford C Research Ethics Committee) and the study was approved by the Oxford Radcliffe Biobank Tissue Access Committee (reference number: 22/A093).<br>2. All CLL patients were approved by favourable study approval on 04Mar2024 by the London- Brent Research Ethics Committee (REC Reference 24/PR/0218, IRAS 330505) and as part of the CAPTURE (NCT03226886) study, a prospective longitudinal cohort study investigating Covid vaccine responses in cancer patients. CAPTURE is sponsored by The Royal Marsden Hospital and was approved as a substudy of TRACERx Renal (NCT03226886) by the National Research Ethics Service Committee London, Fulham on 01/05/2020 (REC 11/LO/1996).EXACT is sponsored by the Royal Marsden Hospital and was approved by the National Research Ethics Service Committee, West Midlands - Black Country on 22/11/2021 (REC/ WM/0251).<br>3. Blood fro SCID patients is approved by the Institutional Review Boards of Children's National Hospital (Washington DC) and of the National Institute of Health (Bethesda, MD) and 93-1-0119, approved by the Institutional Review Board of the National Institutes of Health (Bethesda, MD).<br>4. Adenoid tissue samples were approved by Newcastle University biobank (project code: NRTB-1).<br>5. Ethical approval to obtain blood from healthy individuals was provided by the London-Brent Regional Ethics Committee (REC:21/LO/0682). |

Note that full information on the approval of the study protocol must also be provided in the manuscript.

# Field-specific reporting

Please select the one below that is the best fit for your research. If you are not sure, read the appropriate sections before making your selection.

☒ Life sciences ☐ Behavioural & social sciences ☐ Ecological, evolutionary & environmental sciences

For a reference copy of the document with all sections, see nature.com/documents/nr-reporting-summary-flat.pdf

# Life sciences study design

All studies must disclose on these points even when the disclosure is negative.

| Sample size | For each experiment we estimated the expected change between experimental and control groups (e.g. at least a 20% change and SD at most half the magnitude of the minimum effect size we were interested in). With those assumptions we used power analysis to estimate the group size that would provide at least 80% power to detect statistically-significant difference (with p < 0.05 considered significant). |
|---|---|
| Data exclusions | No data was excluded from the analysis |
| Replication | Fig 1a No experimental replication was feasible<br>Fig 1b-d are representative of 2 experiments<br>Fig 1e-f are representative of 5 experiments<br>Fig 1g are  3 patients (pleural infection), 11 patients (CLL patients), 7 healthy donors, 6 patients (SDB/OME).<br>Fig 1h is representative of 3 healthy donors<br>Fig.1i is representative for 11 patients<br>Fig.1j-m are representative for 3 experiments<br>Fig.1n is representative for 4 experiments<br>Fig 2a-d are representative of 2 experiments<br>Fig 2e-f No experimental replication were feasible<br>Fig 2g-h are representative of 2 experiments<br>Fig 3a No experimental replication is feasible |

Fig 3b is representative of 2 experiments
Fig 3c-h No experimental replication are feasible .
Fig 4a-b No experimental replication is feasible.
Fig 4c-h are representative of 2 experiments.
Fig 4i-j are representative of 2 experiments.
Fig 4k is representative of 2 experiments.
Fig 4l-n are representative of 2 experiments.
Fig 4o are representative of 3 mice.
Fig 5a-k are representative of 2 experiments.
Fig 5l No experimental replication is feasible
Extended Fig 1a-b No experimental replication was feasible.
Extended Fig 1c-g are representative of 2 experiments
Extended Fig 1h is representative of 3 patients (pleural infection)
Extended Fig 1i is representative of 3 patients (pleural infection) and 6 patients (SDB /OME)
Extended Fig 1j is representative of 6 healthy donors
Extended Fig 1k is representative of 3 healthy donors
Extended Fig 2a-c is representative of 3 experiments
Extended Fig 2d-e is representative of 4 experiments
Extended Fig 3a No experimental replication was feasible.
Extended Fig3b-i are representative of 2 experiments
Extended Fig 4a-c are representative of 2 experiments
Extended Fig 5a-b No experimental replication was feasible
Extended Fig 5c pooled data from 3 experiments
Extended Fig 5d are representative of 3 experiments
Extended Fig 5e is either representative of 3 experiments (top) or pooled data from 3 experiments (bottom)
Extended Fig 5f is representative of 3 experiments
Extended Fig 6a-f No experimental replication was feasible
Extended Fig 6g is representative of 7 donors.
Extended Fig 6h No experimental replication was feasible
Extended Fig 7a is representative of 2 experiments
Extended Fig 7b-c are representative of 2 experiments
Extended Fig 7d-f No experimental replication was feasible
Extended Fig 7g is representative of 2 experiments
Extended Fig 7h is representative of 5 mice.
Extended Fig 7i-k are representative of 2 experiments
Extended Fig 7l  is No experimental replication was feasible

| | |
|---|---|
| Randomization | For in vitro experiments, randomization was not required given there were no relevant covariates (i.e. cells from littermate mice came from the same cage, all wells treated simultaneously using multi-channel pipettes, on the same day, in the same single plate, analysed in the same machine, handled by the same investigator). This was not relevant to the analysis of human PBMC samples, as no experimental interventions were undertaken on these samples. |
| Blinding | Investigators planned mouse experiments based on genotype and grouping, but during performance of experiments mice were identified only by randomly assigned number with investigators blind to group allocation. This was not relevant to our human study, as group allocation was not based on any interventions, but the clinical phenotype of human sample donors. CLL samples in particular were impossible to blind given their composition of predominantly malignant cells, which are immediately apparent in cytometric analysis. |

# Reporting for specific materials, systems and methods

We require information from authors about some types of materials, experimental systems and methods used in many studies. Here, indicate whether each material, system or method listed is relevant to your study. If you are not sure if a list item applies to your research, read the appropriate section before selecting a response.

## Materials & experimental systems

| n/a | Involved in the study |
|---|---|
| ☐ | ☒ Antibodies |
| ☒ | ☐ Eukaryotic cell lines |
| ☒ | ☐ Palaeontology and archaeology |
| ☐ | ☒ Animals and other organisms |
| ☒ | ☐ Clinical data |
| ☒ | ☐ Dual use research of concern |
| ☒ | ☐ Plants |

## Methods

| n/a | Involved in the study |
|---|---|
| ☐ | ☒ ChIP-seq |
| ☐ | ☒ Flow cytometry |
| ☒ | ☐ MRI-based neuroimaging |

## Antibodies

| | |
|---|---|
| Antibodies used | All antibodies used are commercially availiable and extensively used. We have listed all antibodies and their clone names in the materials section but given the large number of antibodies used over the breadth of the work we did not note all their lot numbers. B220-BUV737 (RA3-6B2, BD Horizon #612838), CD19-BV605(6D5, Biolegend #115540), CD23-BV421(B3B4, Biolegend #101621), CD5-APC (53-7.3,ebioscience #17-0051-82), IgM-PE-Cy7 (II/41,ebioscience, #25-5790-82), CD19-BUV395(1D3, BD Horizon, #563557), CD3- |

BV650 (17A2, Biolegend, #100229), CD21-BV605(7G6, BD #747763), IgD-PerCP/Cyanine5.5(11-26c.2a, Biolegend, #405710), CD93-BV480(AA4.1, BD, #746239), CD93-PE/Cyanine7(AA4.1, Biolegend, 136506), CD24-Pacific Blue(M1/69, Biolegend #101820), CD43-BV605(S7, BD, #747726), CD43-BV711 (S7, BD, #740668), CD3-APC-Cy7(17A2,Biolegend #100222), CD4-APC-Cy7(GK1.5, Biolegend, #100414), CD11b-APC-Cy7(M1/70, Biolegend,#101226), CD11b-AF700(M1/70,Biolegend,#101222), TER119-APC-Cy7(TER-119, BD, #560509), Gr1-APC-Cy7(RB6-8C5, Biolegend, #108424), Sca-1-BV421(Ly-6A/E,Biolegend, # 108128), c-Kit-APC(2B8, Biolegend, #105812), CD127-PE-Cy7(A7R34, Biolegend, 135014), CD16/32-PerCP/Cyanine5.5(93, Biolegend, #101324), CD135-PE(A2F10, Biolegend, #135306), CD45.1-BV711(A20, Biolegend, #110739), CD45.2-BUV737 (104, BD, #612779), CD274-BV711(B7-H1, Biolegend, # 124319), CD86-PE-Cy7(GL-1, Biolegend, #105014), FCRL5-AF488(biotechne, FAB6757G), BLNK phosphorylated at Tyr84 (J117-1278, BD, #558442), Erk1/2 phosphorylated at Thr202/Tyr204 (20A, BD, #561991), phosphorylated PLCγ2 (K86-1161, BD, #560134), Syk phosphorylated at Tyr525/526 (C87C1, Cell Signaling Technology, #6485S) and Btk phosphorylated at Tyr551 (M4G3LN, eBioscience, #12-9015-42), c-Myc-AF647(Y69, abcam, ab190560), TCF1/TCF7-AF488(C63D9, Cell Signaling Technology, #6444S),LEF1-AF488(C12A5, Cell Signaling Technology, #8490S), DOPC/CHOL/Fluorescein-DHPE (54:45:1); Rhodamine-DH PE Liposomes(DOPC/CHOL/Rhodamine-DHPE(54:45:1) were used (FormumMax, Sunnyvale,CA, F60103F2-R). IL10-PE (JES5-16E3, Biolegend, #505008), IL10-BV421 (JES5-16E3, BD, #563276), CD223(LAG-3)-PE(C9B7W, Biolegend, #125208), CD138-BV605(281-2, Biolegend, #142515),CD138-PE(281-2, Biolegend, #142504), CD43-BV711 (S7, BD, #740668), CD1d-PerCP/Cy5.5 (1B1, Biolegend, #123513), Ki-67-PerCP-eFluorTM710 (SolA15, eBioscience, #46-5698-82), Ig light chain κ-AF700 (RMK-45, Biolegend, #409508),Ig light chain λ-FITC (R26-46, BD,#553434), IgG3-Biotin (RMG3-1, Biolegend, #406803); For human sample staining: TruStain FcX CD16/32/64 antibodies (Biolegend, #422302), CD3-BV785 (HIT3a, BD Bioscience,#740961), CD19-BUV737(SJ25C1, BD Bioscience, #612756), CD19-PerCP/Cyanine5.5 (SJ25C1, Biolegend, #363016), CD27-PECy7(M-T271, BD Bioscience, #560609), CD38-BV605 (HIT2, BD Bioscience, #569699 ), CD43-BV510(1G10, BD Horizon, #563377), CD43-FITC(1G10, BD, #555475), CD38-BV510 (HIT2, Biolegend, #303540), CD5-APC (UCHT2, Biolegend, #300612), CD5-PE/Cyanine7 (UCHT2, Biolegend, #300622), IgA-PerCP-Vio700 (IS11-8E10, MACS, #130-113-478), CD24-BV711 (ML5, BD, #563401), IgM-BUV395 (G20-127, BD, #563903), TCF1/TCF7(C63D9, Cell Signaling Technology,#2203), LEF1(EPR2029Y, Abcam. #ab137872), Alexa Fluor 488 Donkey anti-rabbit IgG (minimal x-reactivity Poly4064, Biolegend, #406416), Alexa Fluor 647 Donkey anti-rabbit IgG (minimal x-reactivity Poly4064, Biolegend, #406414), Streptavidin-BV605 (Biolegend, #405229).

Validation | All antibodies used were commercial antibodies and had been previously validated by the manufacturing companies. We provide the clones used for each antibody. Antibody titrations and dilutions used in each experiment are only relevant to the specific batch used, which change over time and therefore not useful.

# Animals and other research organisms

Policy information about studies involving animals; ARRIVE guidelines recommended for reporting animal research, and Sex and Gender in Research

Laboratory animals | C57BL/6 mice were used in this study. Both male and female mice were used. Mice were used at 8-12 weeks for phenotyping and in vitro experiments, except for aging assessment (4-21 weeks). The animals are held within a Techniplast IVC green line system, the air movement in the cages is regulated by an Techniplast air management unit on negative pressure, 75 ACH/-20%, and all cages are placed on automatic watering, which is chlorinated to 2.45%. The humidity and temperature are regulated at room level, set at code of practice standard levels, 20-24C and 55% +/- 10%. The light cycle is 7am-7pm including dawn and dusk settings of 15 minutes (6:45-7 am and pm). The animals are kept on Datesand Eco Pure Chips sawdust, Bed'r'Nest nesting and smart homes enrichment, with Teklad Global Rodent Diet Sterilised 2018S 18% Protein.

Wild animals | The study did not involve wild animals

Reporting on sex | The study involves both male and female.

Field-collected samples | The study did not involve field animals.

Ethics oversight | Animal experimentation was performed according to the regulations approved by UK Home Office under project license (PP2867252),Australian National University's Animal Experimentation Ethics Committee and University at Buffalo,

Note that full information on the approval of the study protocol must also be provided in the manuscript.

# Plants

Seed stocks | The study did not involve plants.

Novel plant genotypes | The study did not involve plants.

Authentication | The study did not involve plants.

# ChIP-seq

## Data deposition

☒ Confirm that both raw and final processed data have been deposited in a public database such as GEO.

☒ Confirm that you have deposited or provided access to graph files (e.g. BED files) for the called peaks.

| | |
|---|---|
| Data access links<br>*May remain private before publication.* | FASTQ files were downloaded from NCBI's GEO database for TCF1 CHIP-seq data (SRP142342) |
| Files in database submission | This study analyzed published ChIP data (SRP142342) |
| Genome browser session<br>(e.g. UCSC) | This analysis doesn't have Genome browser session. |

## Methodology

| | |
|---|---|
| Replicates | 2 replicates |
| Sequencing depth | Information at SRP142342 |
| Antibodies | TCF1 |
| Peak calling parameters | Peaks were called using MAC2 version 2.1.1 that were enriched in TCF1 relative to input using default parameters. Peaks were annotated using homer version 4.8. |
| Data quality | FASTQ files were aligned to Enembl's mouse GRCm38 genome using BWA version 0.7.15. The resulting BAM files were sorted, duplicates marked and indexed using Picard version 2.1.1. Peaks were called using MAC2 version 2.1.1 that were enriched in TCF1 relative to input using default parameters. Peaks were annotated using homer version 4.8. BAM files were normalised to 10 million reads. |
| Software | IGVTools version 2.3.75 was used to generate coverage files. |

# Flow Cytometry

## Plots

Confirm that:

☒ The axis labels state the marker and fluorochrome used (e.g. CD4-FITC).

☒ The axis scales are clearly visible. Include numbers along axes only for bottom left plot of group (a 'group' is an analysis of identical markers).

☒ All plots are contour plots with outliers or pseudocolor plots.

☒ A numerical value for number of cells or percentage (with statistics) is provided.

## Methodology

| | |
|---|---|
| Sample preparation | Pleural fluid and blood specimens were transferred and processed in the lab within hours after collection. Pleural fluid specimens were centrifuged at 800g for 10 minutes. The acellular supernatant was removed and stored in -80oC. Then the cellular pellet was resuspended in red blood cell lysis buffer (5-10 ml, J62990.AK Thermoscientific) and incubated for 5 minutes. The sample was centrifuged at 500g for 5 minutes and the supernatant was discarded. If there were red blood cells the process was repeated, otherwise the cells were washed in 5 ml of PBS and centrifuged at 500g for 5 minutes. The supernatant was discarded and the cells were resuspended in 5ml of RPMI enriched with 10% FBS, passed through a 70um filter and counted. Whole blood was collected in EDTA tubes (VWR) and stored at 4°C until processing or processed freshly. All samples were processed within 24 hours. Prior to processing, tubes were brought to room temperature (RT). PBMC and plasma were isolated by density-gradient centrifugation using centrifugation tubes (SepMateTM, STEMCELL) per manufacturer's instructions. PBMCs were either proceed to be stained with antibodies directly or resuspended in Recovery cell culture freezing medium (Fisher Scientific) containing 10% DMSO, placed overnight in CoolCell freezing containers (Corning) at -80°C and then transferred to liquid nitrogen for longtime storage. Adenoid tissue samples were collected in phosphate buffered solution and immediately mechanically dissociated into a single cell suspension using scalpel blades and a 100 µm filter. Cells were then cryopreserved in freezing media (10% DMSO and 90% FBS) in liquid nitrogen prior to flow cytometry experiments.<br>Adenoid tissue samples were collected in phosphate buffered solution and immediately mechanically dissociated into a single cell suspension using scalpel blades and a 100 µm filter. Cells were then cryopreserved in freezing media (10% DMSO and 90% FBS) in liquid nitrogen prior to flow cytometry experiments.<br>Single cell suspensions were prepared from mice peritoneal cavity and spleens and B-1 cells and B-2 cells were magnetically purified using mouse B Cell Isolation Kit (Miltenyi Biotec) with anti-mouse CD45R (B220) antibody (Miltenyi Biotec, 130-110-707), labeled with Cell Trace Violet (CTV, Thermo Fisher) and cultured for 72 hours in complete RPMI 1640 media (Sigma-Aldrich) supplemented with 2mM L-Glutamine (GIBCO), 100 U penicillinstreptomycin (GIBCO), 0.1 mM nonessential amino acids (GIBCO), 100 mM HEPES (GIBCO), 55 mM 0- |

mercaptoethanol (GIBCO) and 10% FBS (GIBCO) followed by stimulating with or without 5μg ml-1 LPS (O111:B4, Sigma, L4391), 0.4 μg ml-1 IL-10 (Thermo Fisher, 210-10-10UG), 10 μg ml-1 InVivoMAb anti-mouse IL-10R (CD210) (BioXCell, BE0050) and its isotype control, InVivoMAb rat IgG1 isotype control, anti-horseradish peroxidase (BioXCell, BE0088) at 37°C in 5% CO2. Peritoneal B-1a cells are obtained from mice, the Fc receptors blocked and cells stained and sorted.

For cytokine intraceullular staining, the cells were stimulated with PMA, ionomycin and brefeldin A for the terminal 5h of culture. Cells were harvested and stained for surface markers, including ebioscience Fixable eFluor780 viability dye (Invitrogen) to exclude dead cells before cells were fixed. Intracellular staining was performed with Cytofix/Cytoperm kit (BD) with IL10-PE (JESS-16E3, Biolegend) as recommended. For transcription factors, eBioscience Foxp3/transcription factor staining buffer set (Invitrogen) was used per manufacturer's instructions.

| | |
|---|---|
| Instrument | Cells were sorted on a FACS Aria II, splenocytes and human PBMC samples were acquired on a Fortessa or Fortessa X-20 cytometer. |
| Software | FACS data was analyzed using FlowJo software v10.10.0 (FlowJo LLC). |
| Cell population abundance | Sorted sample purity was based on flow cytometry sorting analysis and stringent gating. Abundance of populations are indicated in the gating figures of the manuscript. |
| Gating strategy | SC-H/FSC-A (cells were gated along a diagonal gating strategy to eliminate cells with disproportional FSC-H and FSC-A size), SSC-W/SSC-H (cells with large SSC-W from scatter were eliminated), FSC-A/Live dead (cells staining negative for the live dead marker were selected as "live") and FSC-A/SSC-A (Cells were gated as lymphocytes if they had a lower size and granularity relative to other signals detected). Once cells were established as singlets, live and lymphocytes analysis was completed as described in the manuscript, where possible biphasic populations were used to identify positive and negative populations. |

☒ Tick this box to confirm that a figure exemplifying the gating strategy is provided in the Supplementary Information.

