## [Peer Review file · Nature]

TCF1 and LEF1 promote B-1a cell homeostasis and regulatory function.

Corresponding Author^s: Professor Carola Vinuesa, Qian Shen

Version 0:

Reviewer comments:

Referee #1

(Remarks to the Author)

This study by Shen and colleagues describes work on mice single and double-deficient in TCF-1 and LEF1. The investigators used these mice to assess the developmental path and functionality of B cell subsets, concluding that the two transcriptional regulators are required to drive CD5+ B-1 cells to develop, expand and function, but that other B cell subsets are unchanged.

TCF-1 has been shown previously to work in conjunction with LEF1 to regulate transcription and its effect in T cells has been explored. Relevant for this study Tcf1 in B cells has been shown to be regulated by EBF1, thus there is evidence of regulation in B cells. Its function in T cells demonstrated among other observations their negative regulation of Blimp-1 expression and support of Bcl6, factors that are critical also for the regulation of B cell functions, and perhaps could explain some of their data demonstrating a reduction in IL10 production by B-1 cells, although this has not been pursued. They also suggest that TCF-1/LEF1 regulates oxidative and fatty acid metabolism and that without these transcriptional regulators Ets1/myc expression in B-1 cells thereby allowing these cells to self-renew, a property of fetal-derived B-1 cells that has not been elucidated before. Finally, they show that the double-deficient B-1a cells show reduced functionality, especially a failure to strongly induce IL10 production.

The study uses flow cytometry, adoptive transfer studies, single cell RNA sequencing and various omics approaches for their analyses of single and double transcription factor knock out mice. Experiments show a sufficient number of replicates, and statistical analysis appears appropriate for the data shown. The study shows a reduction of CD5+ B-1 cells in the body cavities and spleen both in the double-KO animals and following adoptive transfer of fetal liver and bone marrow transfer. They also indicate a direct role for these transcriptional regulators in the expression of CD5 itself and demonstrate a failure of B-1a cells to be activated by LPS to generate IL10, although steady-state IL10 production appeared unaffected.

Despite these far-reaching conclusions, and many interesting and novel data provided, there are incongruencies in the manuscript, inconsistencies, a failure to appropriately integrate all their findings, and a lack of supporting details, that make it difficult to interpret all their data and reconcile some of their findings. Ultimately, I find the study confusing, even upon re-reading, and I am not convinced that they can make the claims with the data they show. I will highlight some of those inconsistencies and omissions below.

We have gained a good understanding that B-1 cell development is regulated by regulators of fetal hematopoiesis. No attempts have been made to integrate their data with those on Lin28/Let7 findings.

1) Figure 1 claims to show the expression of tcf7 (encoding tcf1) and Lef1 in CD5+ B-1 cells. Yet, confusingly, and not explained at all, Tcf7 and lef1 expression are mutually exclusively expressed in their scRNAseq data and only tcf7 expression seems to fall within the CD5+ cluster. They brush that aside by showing histogram plots of TCF1 and LEF1 staining that shows some shifts in histogram plots I do not find at all convincing. Moreover, they show similar "enhanced" expression in B-1a and B-1b compared to B-2 cells, although all their data then go on to show the effect of these transcription factors only on B-1a but not B-1b.

Fig. 1e shows data on total B-1 (unclear what this is) and then B-2 and B-1a and B-1b in body cavity but only B-1a in the spleen. If they can identify total B-1 as well as B-1a, why do they not also show B-1b data for the spleen?

Inconsistent with that discrepancy in the scRNAseq data are their own data that show distinct effects of tcf1 and Lef1. While

Lef1 deficiency reduces CD5 B-1 cells in both spleen and body cavity, tcf1 deficiency affects spleen but not body cavity CD5+ B-1 cells and the Lef1 single KO mice on are largely recapitulating what they see in the double KO. Despite that they claim throughout the need for both factors (consistent with the literature perhaps but not with their data). This is confusing and needs to be clarified, as both their transcriptional data and functional data show differential effects, yet they claim that these factors act in a common fashion.

2) Identification of “B-1 cell precursors” in the bone marrow and spleen is very challenging and it is very unclear how they derived these data, particularly given that both mature B-1 cells and B-1 derived antibody secreting cells as well multiple stages of B-2 cells show overlapping phenotypes. Flow cytometry data to support each claim of their manuscript stating reductions in precursor populations must be included and must convincingly demonstrate that these cells are indeed B-1 cells. As written, I do not find their claim that TCF1 and LEF1 promote B-1a cell development from the earliest precursors (line 136/137) substantiated with sufficient data.

That said, their fetal liver transfer data are perhaps most consistent with their claims and go a long way in convincing that there are indeed defects on CD5+ B-1 cells. Yet, whether this is truly a developmental defect is difficult to discern and in contrast to the title and abstract, and some of the figures (which make those claims), their own conclusion on line 66 states that these transcription factors regulate self-renewal and thus perhaps enhances expansion.

3) The investigators show in Fig. 2a a differentiation pathway of B-1 cells that is supported by the literature. Yet, in their text in multiple parts of the manuscript (for example line 48) they imply that peritoneal cavity CD5+ B-1 cells give rise to “their splenic offspring”. In contrast, existing evidence suggest that splenic B-1 cells precede in development peritoneal cavity B-1 cells and that elimination of the spleen (see Carsetti et al) will cause a disappearance of B-1 cells in the body cavity over time, but not the other way around. Only in response to cytokines and/or TLR do B-1 cells accumulate in the spleen, largely as effectors. Clarifications of these statements must be done.

4) The staining for Lef1 and tcf1 by flow cytometry must be shown. Their data reported gMFI (Fig. 2f) suggests at best extremely minor differences in expression. This might be biologically meaningful, but as shown I find such data unconvincing, and they fuel suspicion of overstatements throughout the paper that are troubling. Similarly, Ki67 expression is induced in cells entering G1. How meaningful is a shift in staining so small that gating cannot be done and expression of the entire population is shown as small changes in gMFI. Ki67 staining is usually giving good distinctions between positive and negative cells. As such their data are unconvincing as shown. FACS gating to transitional B-1a cells and the facs plots showing Ki67 staining should be shown to clarify, because in 2f they then state that nearly 100% of B-1a cells are Ki67+, but the tissue of B-1 origin is unclear.

5) Their statement that B-1 cells are the normal counterparts of CLL in humans is an unsubstantiated claim and the papers they are referencing do not demonstrate that.

6) They claim to find reductions also in “human B-1 cells”, using gating strategies that have been largely dismissed by the field as inappropriate. Indeed, no convincing human orthologues of B-1 cells have been isolated from adult humans to-date, while two recent studies with human fetal tissue identify potential orthologues. This data should be removed.

7) Line 147 “failure to accumulate” seems an overstatement. Instead, a convincing reduction in B-1a cell expansion seems to occur. Suggest modifying this statement.

8) Line 185 states unchanged frequencies of B-1 cell-derived plasma cells. However, extended Fig. 2b indicates significant enhanced plasma cell frequencies in the double KO mice?

9) The investigators state that the repertoire of B-1a cells is more diverse in mice lacking the 2 transcription factors (Fig. 2k). Yet this analysis is done (if I understand correctly) of total peritoneal cavity B cells, not B-1(a) cells and include B-2? Is that correct? Please clarify

10) If indeed the diversity of the clones were to expand as shown in Fig. 2k, it seems inconceivable that the % of PtC binders among B-1a cells would not at all be affected. The authors should clarify this point and how shifts in diversity would not affect % of PtC binders (which they show in their hands to be >40% of all B-1a cells).

11) Fig. 3a shows a very pronounced decrease in IgM-BCR expression among double KO peritoneal B-1a cells. While they seem to suggest effects of class-switching, the reduction seems to affect the entire cell pool. Would this reduction not explain perhaps the changes in signaling etc they are seeing in their data?

12) The direct effects of the transcriptional regulators on cd5 is important to understand the selective effect on the CD5+ B-1 cell populations and their facs-plots clearly support their transcriptional data. If CD5 expression is affected, given it is the sole means of identifying B-1 cells in this study – would that not alone then explain the differences they see, given the known role of CD5 as a regulator of TCR/BCR signaling strengths?

13) The expression of CTLA4 and PD-L1 by B-1 cells was shown previously to have significant functional effects by the Herzenberg group (Yang et al.) and studies by the Haas group. These manuscripts should be cited.

Referee #2

(Remarks to the Author)

There has been much interest in the molecular understandings that govern different B and T cell subsets ability to self-renew or differentiate. Here, Shen et al., demonstrate this ability in a subset of self-renewing B cells, B-1a cells. TCF1, a

transcription factor known for sustaining precursor cells in CD8 and CD4 T cells, and LEF1 deficiency were shown to specifically decrease the B-1a population using conditional knockout mice. This decrease in B-1a cells was shown to be due to an inability of B-1a precursors (B-1P) to differentiate into B-1a cells as well as an intrinsic proliferation defect of B-1a cells. Transcriptional and ATAC-seq analysis of TCF1/LEF1 double knockout or B cell transgenic B-1a cells revealed direct regulation of *Myc* and altered metabolic pathways governed by the two transcription factors. Furthermore, B-1a cells production of IL-10 was found to be promoted by TCF1/LEF1, and transfer of these B cells into EAE mice resulted in worse clinical scores. TCF1/LEF1 deficient B-1a take on an “exhausted phenotype” and failed to properly differentiate into plasma cells. While the general finding of TCF1 function is not novel, this paper expands this knowledge to include these transcription factors to self-renewing B-1 cells. However there are several points that are not properly addressed and require more clarification.

Comments:

Their data suggests that of *Tcf1* and *Lef1* may have different roles contributing to B-1a B cell development and differentiation. Yet this critical point is not addressed and remains unclear as most of this study was performed using *Tcf7* and *Lef1* knockout mice. Fig 1a shows that there is not much overlap in cells expressing both transcripts and *Lef1* does not appear to overlap with the B-1a subset, based on Cd5. The single knockout of *Lef1* has a more drastic effect on B-1a frequency in the peritoneal cavity than *Tcf1* single knockouts. Could *Tcf7* and *Lef1* be expressed at different stages of B-1a development, or even promote different subsets (such as a Mychi and Myclo). Is there a difference in the kinetics of TCF1 and LEF1 expression during B-1a development? This premise leads to the possibility that *Tcf1* may regulate self-renewal and *Lef1* may regulate function. Parsing out the difference between these factors may provide better understanding of the transcriptional differences dictating self-renewal and function of B-1a cells, and how this effects populations in the peritoneal cavity versus the spleen.

The double knockout mice exhibit functional differences to wild type B-1a cells in terms of IL-10 production and PD-L1 expression. This idea is tested in the EAE model and the abstract claims “TCF1 and LEF1 are also essential for IL-10 and PD-L1 expression upon B-1 cell activation and repression of CNS inflammation.” While the clinical score is clearly impacted by transfer of the different B-1 B cells there is no data to demonstrate the mechanism (i.e role of IL-10 or PD-L1) behind this result. Do the transferred B cells maintain PD-L1 and IL-10 production. Since the figure and legend state B-1 B cells are transferred and not just B-1a are they differentiating into other B cells that impact the clinical score. Are they secreting systemic amounts of IL-10, entering the CNS or is this effect promoted in secondary lymphoid organs. Further assessment of the B-1a cell’s role in providing protection should be assessed.

The results are difficult to follow as the experiments oscillate between different tissues such as peritoneal cavity, spleen and bone marrow as well as different precursor and mature (B-1a vs B-1). While B-1a are typically thought to have more impact in the peritoneal cavity than spleen, the rationale at times for looking in different tissue is hard to follow. Fig 2a does a good job of explaining this point, but it does not get into why the authors look in the spleen in reference to the B-1a cells in figs 1, 2, and 4. It is also not made clear if there are difference in B-1a cells that arise from conventional fetal liver pathway versus bone marrow precursors.

Minor:

The concluding paragraph is focused on asymmetric cell division, which parallels these findings with that of CD8 T cells. However, their data does not investigate their role of asymmetric cell division and their final conclusion should not assume this is true.

Fig 1e,f: condensing the graphs so all the groups can be shown together would be ideal. Or else, a more salient determinant of the single deficiency groups.

Fig 2k demonstrates that KO cells are more diverse as adults, while being similar to WT in neonates. What is the speculation of why this is?

In figure 5a the authors show that blocking IL-10 inhibits the PC:non-PC ratio implying it is due to lack of IL-10. Fig 4e shows reduced IL-10, not an ablation, so if the IL-10 is added back will it restore the phenotype. Is it possible that while IL-10 is reduced, TCF1/*Lef1* also impacts the ability of these B cells to respond to IL-10 and inhibit their PC differentiation?

Referee #3

(Remarks to the Author)

Qian Shen and co-authors used double-knockout mice and adoptive transfers (bone marrow and fetal liver chimeras) and a series of cellular and molecular readouts to define the role of the transcription factors TCF1 and LEF1 in B-1a cell development, self-renewal and regulatory functions. The authors concluded that TCF1 & LEF1 specifically drive the formation and self-renewing of B-1a cells. They also showed that TCF1/LEF1 deficiency down-regulates *Ets1* and *Myc* targets, leading to the expression of an exhausted B-1a cell phenotype similar to age-associated B cells.

Although the observations in the TCF1/LEF1 double-knockout (DKO) mice are interesting, particularly the exhausted B-cell phenotype, the rather overall modest effect on B-1a, including minimal effect in the development of the archetypal B-1a repertoire (VH11/VH12/anti-PtC), natural antibody levels, and splenic B-1a numbers, does not support the main premise of this study. Much of the observed phenotype and putative decrease in B-1a numbers in the TCF1/LEF1 DKO mice could be attributed to their effect on down-regulating the expression levels (MFI) of CD5, CD19 and IgM instead of a direct effect on B-1a development.

Major concerns:

1. The authors claim the mechanisms underpinning B-1a development and self-renewal are unknown. However, several groups have shown the role of *Bhlhe41* (Kreslavsky et al. Nat Immunol, 2017), *Bmi-1* (Kobayashi et al., JI 2020), CD19

(Rickert et al., Nature 1995; Haas et al., Immunity 2005), Btk and other genes in B-1a development and self-renewal. These previous studies show a stronger effect on B-1a development and self-renewal than the TCF1/LEF1 DKO shown here. More importantly, in these previous studies, the targeted knockout gene directly affected the development of canonical B-1a repertoire (including VH11, VH12, and/or anti-PtC). In contrast, the TCF1/LEF1 DKO did not significantly affect the development of the archetypal B-1a repertoire.

2. Authors claim that TCF1 and LEF1 are highly expressed in B-1 cells. Although this is true for TCF1, their data, including from ImmGen, shows LEF1's highest expression on follicular B-2, memory B, and B-2 transitional cells, not B-1a (Fig. 1a, Ext Fig. 1b). This discrepancy was not directly addressed. Although TCF1 is mainly expressed in B-1a, the TCF1-single KO does not significantly affect B-1a development; only the DKO with LEF1 shows a mild decrease in B-1a. However, LEF1 is highly expressed in B-2 cells, not B-1a (Fig. 1e).

3. CD5 expression levels (measured by flow cytometry MFI) are drastically decreased in TCF1/LEF1 DKO B-1a cells (Figs. 2g, 3e). Given that CD5 MFI is critical to identify, gate, and quantify B-1a cells in this study, the decrease in the CD5 MFI on the cell surface of B-1a could have confounded their quantification in this study, potentially invalidating their premise. The lower expression of CD5 in the DKO mice may have "merged" the B-1a (CD5^{low}) with B-1b (CD5^{neg}) in the flow cytometry gating strategy used to quantify B-cell subsets. Indeed, the fact that DKO mice develop archetypal B-1a cells (VH11/VH12/anti-PtC) at levels similar to wild-type suggests that B-1a cells can develop in TCF1/LEF1 DKO mice but have decreased CD5 expression.

4. Similarly, TCF1/LEF1 DKO B-1a cells have lower expression of CD19 (Ext. Fig. 2a) and IgM (Fig. 3a), which are known to affect B-1a development. Hence, TCF1/LEF1 transcription factors may affect B-1a development only indirectly through the expression of CD19, IgM, and CD5, as already shown in previous studies.

5. B-1 cell progenitors (B-1P) were rather increased in TCF1/LEF1 DKO mice (Ext. Figs.), contradicting the main premise of this study. Moreover, the authors speculated about a putative B-1a progenitor characterized as CD5⁺ without proper citation for this or a functional assay to validate their potential to generate B-1a. These cells expressing CD19⁺/B220⁻/CD5⁺ could be mature B or plasma cells instead of canonical B-1P.

6. In adoptive transfer chimeras, DKO B-1a cells developed and reconstituted up to 50% of the B-1a compartment in the peritoneal cavity, while it reconstituted up to 100% of splenic B-1a (Fig. 1g). Hence, TCF1/LEF1 DKO shows only mild effect on B-1a development. Previous studies (see above) have shown a more robust effect, including loss of the archetypal B-1a repertoire.

7. TCF1/LEF1 DKO mice show no significant changes in natural antibody production and B-1 plasma cells, challenging the main premise of the study.

8. Reduction in transitional B-1a was minimal (~30%), followed by a comparable reduction in B-2 cells (~20%), raising questions about the magnitude of the effect on B-1a cells.

9. The flow cytometry and gating strategy used to identify B-1a self-renewal in Fig. 2h lacks proper control and thus can be misleading. The authors use a quadrant gate that cuts through the middle of a normally-distributed cluster (Fig. 2h, top left quadrant). This plot needs a negative control that includes BrDU staining in cells from mice that did not receive BrDU treatment.

10. TCF1/LEF1 deficiency leads to less than 50% reduction in IL-10 production. Therefore, one cannot conclude that these transcription factors promote IL-10 production while basal IL-10 production levels are conserved in splenic DKO B cells.

11. Human B-1a is not well characterized in humans, and the gating strategy used here is known to include cell doublets and plasma cells (as discussed in earlier publications related to human B-1). Most importantly, no significant difference exists in TCF1/LEF1 expression levels in human memory B cells and the so-called B-1a, opposing or weakening the study's premise. It would be important to evaluate the role of these transcription factors in human fetal liver B cells.

In conclusion, although the TCF1/LEF1 DKO mice give rise to some interesting B-cell phenotypes, their effect on B-1a development and self-renewal is small compared to previous studies on Bhlhe41, Bmi-1, CD19, Btk, and others, which have not been directly compared and accommodated to the new TCF1/LEF1 DKO phenotype presented here. Moreover, these transcription factors are commonly expressed on various B-cell subsets in mice and humans – not specific to B-1a. Thus, the data provided here does not support the main interpretation that the transcription factors TCF1 and LEF1 promote B-1a development.

Minor concerns:

1. B-1 cells also make IgG, particularly IgG3 (and not only IgM and IgA as described in line 25) (see Yang Yang et al., PNAS 2012)

2. The term EAE used in lines 50 and 275 might refer to "Experimental Autoimmune Encephalomyelitis" and not "Experimental Allergic Encephalomyelitis"? If so, there is missing a key reference on the role of B-1a cells in EAE (Kurnellas et al., PNAS 2015)

3. The reference cited in lines 55 and 162 is not related to normal B-1a development but to mature B cells. Instead, please cite the reference from Kyoko Hayakawa on Thy-1+ B-1a cells when referring to the role of BCR signaling in normal B-1a development.
4. Missing references supporting the statement about the difference between progenitors that give rise to B-1 and B2 in line 87 and the putative CD5+ B-1P.
5. It is unclear how transitional B-1a cells were gated, and other gating strategies need proper FMO or other negative controls to define gating boundaries.
6. PDL1-L2 expression on B cells, including B-1, has been previously published and not cited (Rothstein's group and others).
7. The rationale for blocking IL-10R signaling was unclear, which is already known to have an autocrine function in B cells, and the effect here was mild.
8. Given the recently published role of ZEB2 in driving the formation of age-associated B cells, which were increased in the DKO mice here, it would be important to investigate the link between TFC1/LEF1 and ZEB2.
9. There is no clear justification for using geometric mean fluorescence (gMFI) instead of median MFI for the well-resolved clusters. Will the statistical significance change?
10. Fig. 3d has a typo "f" on top of the heatmap.
11. The blue label in Fig. 4d might refer to B-2 cells instead of CD19+/B220-?
12. The phenotype of B cells in Fig. 4f might represent Marginal Zone B cells due to high expression of CD21 and B220.

Version 1:

Reviewer comments:

Referee #1

(Remarks to the Author)

This manuscript resubmission by Shen et al. includes significant additional data, some reorganization and clarification of statements and additional changes to the revised manuscript. The paper is a tour-de-force of analyses on the role of two transcriptional regulators (TCF1 and Lef1) conducted in both mouse strains and human samples, suggesting their distinct involvement in B-1 cell development and regulation following activation.

The authors have addressed with these changes many of my prior concerns. Most significantly for that is the inclusion of human data that seem to suggest B-1 like populations among CD38- CD43+ CD5+ human B cells. Their additional data expand prior studies including peritoneal cavity efflux of humans with bacterial peritoneal cavity sepsis that are impressive additions to the study, with some limitations. They show a correlation between the transcriptional regulators and these B cell populations in humans. Since expression of these regulators is not restricted to B-1 cells, I remain concerned about the term human B-1, given prior work having shown that this population, identified based on their phenotype, is heterogeneous including many class-switched cells etc.

That said, their expression data are convincing and may help to further clarify functional versus developmental relationships between mouse and human B cells.

There are new data and interesting results in the study expanding the role of TCF1 as an inducer of stem-ness from CD8 T cells, to also B-1, perhaps one of the most significant conclusions.

A couple of minor concerns should be addressed:

- 1) Line 324 refers to data in Extended Fig. 4h that are at best confusing. The text states an 80% reduction in patients with IL2RG deficiency and the bar chart in that fig suggests that is the case. However, the flow plots providing an example (I assume) of these patients showed a 7-fold increase in the subset compared to HD (8% versus 1%)?
- 2) The authors should indicate for Figure 3 scRNAseq from which tissue these cells were taken. Its neither stated in the text nor the figure legend.
- 3) Fig. 4n is cited prior to Figs 4k-m

Referee #2

(Remarks to the Author)

The manuscript was greatly improved following revision, and the authors nicely addressed the reviewer's concerns. The

authors provided several additional experiments that strengthen the study, including assessing human B-1a cells from serosal cavities and CLL patients, further single-cell analysis of distinct developmental stages of B-1a, and generating a B-1a reporter mouse to track these cells in their EAE model. While this manuscript has many interesting and novel findings on B-1a cell development and function, a few issues still require further clarification.

1. Figure 1B shows a clear TCF1+ B-1a and a LEF1+ B-1a population by flow cytometry. Based on the distinct populations of either Tcf7 or Lef1 in Figure 1A and most of the mouse data is generated using a double knockout, why do the authors not look for the frequency of TCF1+ and LEF1+ double-positive B-1a cells by flow. Their trajectory data suggests that these transcription factors may have a role at different stages of B-1a development and function; determining if B-1a are all double positive or can be broken down into different populations would help understand these cells.

2. IL-10 was important in B-1a function to prevent excessive proliferation and promote the stemness of B1 cells via TCF1/LEF1. However, there appears to be a discrepancy between mice and humans. The TCF1/LEF1 populations in human B-1a cells express minimal IL10 (Figure 3h). Can the authors elaborate on whether this suggests different mechanisms between the mouse and human functions of B-1a cells.

3. While the concluding paragraph expands on some interesting ideas of exploring the effects of B-1a cell regulation in different diseases and their dysregulation into an exhausted phenotype, it does not summarize the global context of why these findings are important.

4. B-1a cells regulating disease through IL10 and PD-L1 are interesting. Is it known what the targets of these mediators are? In addition, in the EAE model (Figure 4n and o), the authors asked if B-1a cells exerted their regulatory effects locally or distally (line 386). However, their data demonstrate that B-1a cells were present in the brain, which also doesn't rule out a distal influence, perhaps in SLOs. This may not be within the scope of this paper, but some insights into their regulation mechanisms in affected organs vs SLOs would place these findings into a broader, more meaningful context.

Referee #3

(Remarks to the Author)

This is a resubmission from Qian Shen, Carola Vinuesa's team, and collaborators of the manuscript "The transcription factors TCF1 and LEF1 promote B-1a cell homeostasis and regulatory function".

I appreciate the additional experiments the authors performed to revise their important work. I agree that knocking out the two genes simultaneously moderately impacts B-1a cells. However, the new experiments and their interpretations did not address the major concerns raised in round one by the reviewers. Indeed, the authors also acknowledge that their results cannot be interpreted to conclude that TCF1/LEF1 are required for B-1a development.

Overall, the effects of TCF7-knockout (KO), LEF1-KO, or the double-KO (DKO) on B-1a cells were only partial and moderate compared to the more significant effects of other previously published transcription factors/genes. For example:

1. Transfers of DKO progenitors led to the reconstitution of roughly 50% of the normal/wild-type (WT) levels of B-1a in the PerC. In the spleen, the effect was minimal, if any. In fact, one outlier animal contributed to the statistical significance, and most recipients seem to have reconstituted most of their normal levels of splenic B-1a (Fig. 1).

2. TCF7-KO did not affect B-1a percentages in the PerC (Fig. 1j). And in the DKO mice, the effect was only moderate. There was a decrease from ~30% of B-1a in wild-type (WT) to ~10% of B-1a in DKO. However, the data showed a large spread in B-1a percentages, with some DKO animals showing a percent of B-1a cells similar to their wild-type (WT) controls (Fig. 1). For example, in Fig. 1i, the two bottom right panels show 45% (18%+26.6%) of B-1 cells in the LEF1-KO versus 35% (19.3%+16.4%) of B-1 cells in the DKO. These ranges of B-1 cells are similar to the WT shown in the first top left panel of Fig. 1i (46.6% of B-1).

3. The DKO mice develop B-1a cells with the same canonical (stereotypic) B-1a BCR repertoire as the WT mice. The decrease in the B-1a repertoire showed by the authors reflects only a global reduction in B-1a numbers, but there was no direct effect on the development of the canonical B-1a BCR repertoire, including the B-1a-specific anti-PtC repertoire.

4. Most of the moderate reduction in B-1a percentages in DKO mice can be directly attributed to the decrease of IgM and CD5 on their surface, as previously described. For example, it is already established that changes in the BCR/IgM signaling have a stronger effect on B-1a cells than B-2 cells. These effects (i.e., decrease in B-1a numbers) can be seen in CD19-KO, Btk-KO, Nur77, and other molecules downstream BCR signaling. Although it did not achieve a significant p-value, the IgM expression on B-2 cells from DKO mice also decreased, suggesting a role for TCF1/LEF1 in BCR/IgM rather than on B-1a. Since B-1 is more sensitive to BCR changes compared to B-2, the B-1 cells were affected more significantly in the DKO mice. Showing the role of TCF1/LEF1 on Ig/BCR (and CD5) homeostasis might shed light on their mechanisms.

5. Given that other genes and transcription factors (TFs), including ARID3a, Lin28b, Bhlhe41, NFATc1, have been shown to regulate B-1a development and homeostasis more robustly than TCF1/LEF1, it would be reasonable to expect a similar or at least more substantial effect of TCF1/LEF1 DKO on B-1a. Thus, it seems unlikely that TCF1/LEF1 represent key TFs or master regulators of B-1a cells in mice and humans. Such a statement in a major publication may add more confusion to the field.

6. The authors show that LEF1 (an enhancer) and TCF1 (a transcription factor) affect different signaling pathways and are expressed at different B-cell development and maturation stages. Hence, there is no logical reason for focusing the study on the DKO instead of each gene individually, which did not significantly affect B-1a numbers.

7. The effect of DKO on IL-10 occurred only for the “induced” IL-10 and did not affect the constitutive production of IL-10 by B-1a-like cells, suggesting a lack of a direct causal effect. Indeed, it is known that CD5 expression promotes IL-10 production, and a decrease of CD5 on B-1a is expected to decrease IL-10 as previously shown by Gary-Gouy et al. in 2002 (Human CD5 promotes B-cell survival through stimulation of autocrine IL-10 production. *Blood*, 2002). Thus, the reduced induced IL-10 observed here might be due to decreased CD5 expression on B-1a cells.

8. Finally, if not later confirmed, classifying human adult B-1 cells solely based on their expression of CD5+/CD43+ can be harmful to the field. Mouse B-1 cells are defined based on their separate developmental origins and functions, confirmed by lineage/genetic tracing and transfer studies. There is no current way to assess the origin of the CD5+ cells in the infected patients they describe here. In fact, many groups have already tried to characterize human B-1 based on CD5 and failed. They failed because human adult B cells (including adult bone marrow B cells) can express CD5 during development and activation, making identifying human B-1 cells challenging.

Other major concerns include:

- The overall quality of the flow plots needs improvement. The scales (and ticks) on the X/Y axis are sometimes missing or show errors (log vs bioexponential), and the gates are arbitrary (particularly for mouse LEF1 and also for human samples), missing FMO gates, percentage values, etc. (see below).
- The expression of TCF1 (flow) and TCF7 (scRNA-seq) in B-1 is negligible and cannot be quantified compared to B-2 cells. The only protein data for TCF1/LEF1 is based on a flow plot likely containing artifacts of data generation and analysis: for example, the “positive” gates are based on CD23- cells (i.e., B-1 cells), which intentionally exclude all B-2 cells and include only the tip of a rather normally distributed population that spread on the Y axis. This spreading of the population is common in flow cytometry due to spillover spreading and cannot support the claim that the tip-top of the spread contains positive cells and the bottom part includes negative cells—they may all represent the same population/phenotype, but appear spread due to spillover spreading and autofluorescence. For example, if the authors move the gate down along the population spread, they might find that those cells at the middle and bottom of the spread contain the same B-1a cells (same phenotype) as the top part. There might be no differences between the top and bottom of the spread. Hence, one cannot conclude from this flow plot that B-1a express high LEF1 and TCF1 proteins. They will need to titrate the reagent, analyze the titrated data, then perform FMO on the optimal titration to confirm the expression of these proteins on B-1a, and perhaps perform another orthogonal assay for protein detection to confirm.
- The same problem with the spleen flow plot. There are no cells inside the LEF1^{hi} gate and the only population that shows a “bump” indicating a positive expression are CD19^{low} cells, which are not B-1a. Indeed, the arbitrary gates show that most of the cells (in red) represent the B220^{hi} (i.e., B-2/MZB) population, not only B-1a.
- The authors claim that DKO increases age-associated B cells (ABC), which was recently shown to require ZEB2 and express CD11c. However, the authors did not see a difference in ZEB2 and CD11c in DKO mice and no explanation was provided. Thus, perhaps we should avoid calling them ABCs before further characterization.
- In Fig. 1a UMAP (and extended 1b), LEF1 appears to be expressed on B-2, not B-1
- ARID3a and LIN28b are the main transcription factors known to regulate B-1a cells, and this was confirmed by the authors’ scRNA-seq data. However, they did not describe how LEF1/TCF1 are mechanistically related to these B-1a master regulators (and others such as Bhlhe41, NFATc1) since the DKO didn’t show the same strong effect on B-1a as these previously published genes/TFs.
- The entire data interpretation is based on the “percentage” of B cells in DKO vs WT, rather than their absolute numbers. However, the percentage of B-1a cells will change depending on what other types of B cells are included in the CD19+ gate for each organ in the WT versus DKO. For example, if the number of CD19+ B-cell progenitors increases, the B-1a percentage would decrease even if the actual absolute numbers of B-1a remain somewhat stable. All data should have been reported in absolute numbers by counting the total number of cells in the PerC, spleen, etc.
- The authors rely on CD5 expression to identify B-1 progenitors in the spleen. However, it is known that splenic B-1 progenitors can be identified before the CD5 expression (Ghosn et al. Distinct progenitors for B-1 and B-2 cells are present in adult mouse spleen. *PNAS*, 2011).
- In this revision, the authors now suggest DKO might play a role in self-renewal rather than early differentiation of B-1 cells. However, they did not provide experimental evidence of how TCF1/LEF1 relates to Bmi1 gene, which has already been shown to control B-1a self-renewal. The BrDU data was not conclusive. For example, in Fig. 2g there are two nicely distinct populations of BrDU expression for the Ly5a+ cells. However, for the Ly5a-negative cells there is only one normally-distributed population that fluctuates around the gate drawn for the Ly5a+ pop. That population may represent autofluorescence in the WT but not in the DKO cells, which we know show less IgM/CD5 and might be less autofluorescent. Thus, based on these flow plots and gates, the authors cannot make definitive conclusions for the Ly5a-neg population without FMO and other controls for Ly5a-neg cells.

- It is misleading to say that the B cells' BCR repertoire diversity increased in the PerC of DKO mice. As the authors confirmed during their revisions, the BCR diversity is likely due to increased B-2 cells in the PerC of DKO mice. Hence, TCF1/LEF1 may not affect B-1a repertoire. Indeed, the percentage of B-1a expressing the stereotypic anti-PtC repertoire hasn't changed between WT and DKO. Thus, sequencing the repertoire of total peritoneal B cells (instead of the B-1a vs B-2) can confuse the reader.

- Fig. 1 (j and l) show a significant variability between mice and some values are inconsistent, suggesting they might have been incorrectly plotted? For example, the WT mice in the TCF1-KO group (yellow) show a large spread in the percentages of B-1, but not B-2. Since the data is reported in percentage, a large decrease in B-1 should have resulted in a large increase in B-2, but that was not the data plotted in Fig. 1j-l. In addition, one data-point might have been dropped (one mouse) from the B-2 group, which contains only 4 mice, compared to 5 mice for all the other corresponding B-1 groups in the same Fig. 1.

- Based on scRNA-seq data and flow, LEF1 and TCF7+ cells rarely overlap (they appear almost mutually exclusive), but in human fetal tissues they appear as double-positive. In addition, IL7R are known to be expressed on progenitors, which according to Fig.3f (scrRNA-seq), should express LEF1, not TCF7. However, the human fetal samples show IL7R expressed only on double-positive cells, suggesting a different cell type?

- In Fig. 1e, the authors show >60% of B cells in infected individuals' blood and pleural cavity are B-1 cells. This is highly unlikely. Classifying human B-1 cells solely based on CD5 and CD43 may be inaccurate as the field has already addressed this many years ago. CD5 was not a reliable marker for human B-1 because developing B cells in adults, including adult bone marrow B cells, can express CD5, CD38, CD43, etc., due to early differentiation or activation. Thus, I would be cautious claiming that 60% of human blood B cells are B-1 during infection solely based on CD5/CD43 expression.

- The human blood data in panels e and g (Fig 1) are inconsistent. In panel e, they show >60% of CD5+/CD43+, but in panel g there is virtually no cells expressing CD5 (and the percentage value is missing).

- It was unclear to me why they analyzed IL-2RG-deficient patients to quantify B-1 cells without describing whether the mechanism of TCF1/LEF1 in B-1a is through IL-2. In addition, the flow gates drawn to define the CD5+ are arbitrary and inconclusive, likely containing contamination from the main cluster. In extended Fig. 4h, since the number of cells collected for the patients was much lower than for healthy controls, the few dots inside the CD5 gate can misrepresent a high percentage, skewing the result and interpretation. There is no clear CD5+/CD43+ population, and thus the percentage reported may be meaningless.

Minor comments:

- Fig. 1c, the bottom right panel, has the axis labels reverted, I believe?
- The ref. 11 in line 216 may be incorrect. I believe they meant to describe the Thy-1-KO study from Hayakawa/Hardy et al.
- The ref. 78 in line 378 may be incorrect. I believe they meant to describe the role of B-1a in EAE by Kurnellas et al., PNAS.

Finally, I acknowledge and appreciate the important work of the authors, and thus think their findings, which show a moderate effect on B-1a cells, could be more accurately described as "TCF1/LEF1 DKO mice show decreased percentage of B-1a cells, impaired formation of a Myc-lo population, and a decrease in stimulation-induced IL-10 and PDL1 in activated B-1a cells. This phenotype is likely due, in part, to the decrease of CD5 and IgM expression on B-1a cells." This conclusion seems reasonable based on the data presented here.

Version 2:

Reviewer comments:

Referee #1

(Remarks to the Author)

I wished the authors would be a little more careful then stating "we identified...human B-1 cells" in the abstract. I understand its to get published in Nature, but the phenotype of their human cells have been reported previously to be B-1 just to be shown wrong. As I stated before these cells (when sorted) are heterogeneous, not just with regards to Ig subset, I am well aware of the ability of B-1 cells to class switch.

A more careful-statement, such as B-1 like cells does not distract from their excellent work, but given how few people work on B-1, the authors statements suggest an absoluteness that I do not feel is warranted, perhaps also reflected in the many concerns of Reviewer #3.

The body of work provided in this manuscript adds substantially to our understanding of B cell subsets in mice and humans and I have no further comments.

Referee #2

(Remarks to the Author)

The authors have fully addressed my questions and concerns during this past revision. I have no additional questions and believe this study greatly expands the mechanism of B1-a cell development and function and is suitable for publication.

Referee #3

(Remarks to the Author)

I appreciate the author's efforts to address my suggestions and concerns. Here are additional comments/clarifications regarding some of the author's responses to my previous review. I divided my review into three parts: A) Mouse B-1 cells, B) Technical/data analysis, and C) Human B-1 cells.

A) Regarding Mouse B-1 cells:

1. The authors have shown compelling B-1a phenotype and function on the double TCF1/LEF1-KO (DKO) mice/cells. I also appreciate the authors' careful searching of the literature for other transcription factors (TFs) and molecules that affect B-1a development, as I suggested during my previous review. Based on how the authors addressed my comments, I find it helpful to clarify that every experimental model has caveats. My point in bringing up previous studies on B-1a development is not to undermine the extensive and important work from the authors here, but to bring to their attention possible mechanistic explanations for the effects they observe. It is also to highlight that other genes show a strong effect on B-1a development and repertoire that are desirable to consider when building a working model based on how their new data fits into the current paradigm/literature. Clarifying this, I believe, is relevant and helpful to the field and the readers. See below for some clarifications on the points I made on my initial review and additional comments.

2. My comments on the role of other TFs/genes in B-cell biology are focused on the B-1a lineage, which is known to represent a separate B-cell lineage that emerges from separate B-cell progenitors. As the authors mentioned, Arid3a deficiency leads to >80% reduction in B-1a in the peritoneal cavity. In another paper, Yan Zhou et al. (Lin28b promotes fetal B lymphopoiesis through the transcription factor Arid3a, JEM, 2015), they manipulated Lin28 and Arid3a expression in fetal vs adult B-cell progenitors in adoptive transfer experiments to confirm a robust decrease in CD5+ B-1a cells. These adoptive transfer experiments can be compared to similar experiments performed here, in Fig. 1n. Similarly, the NFATc1 study mentioned by the authors uses adoptive transfers to show 9-fold decrease in B-1a cells and >10-fold decrease in anti-PtC binders (Figs 2A,B,C). I compared that to the adoptive transfer experiments performed by the authors here in their Fig. 1n, showing ~10% B-1a reconstitution from WT versus ~4% B-1a in the TCF1/LEF1 DKO. Other studies also performed conditional NFATc1-KO using Vav or CD19 cre models (Märklin et al., 2020). In the case of Bhlhe41 deficiency, in addition to >70% decrease in B-1a, the study shows a complete absence of a key B-1 repertoire (i.e., lack of VH12). And there are the studies related to BCR/Ig/CD19 signaling, including work from Thomas Tedder's and Rajewsky's groups showing the role of CD19 in B-1a development, where CD19-/- lacks B-1a-derived natural antibodies and defects in B-1a development. An attempt to integrate these findings into TCF1/LEF1 would be helpful to the field. As I suggested in my last review, I believe some of the results observed by the authors here could be explained/linked to previous studies above—this is not a negative criticism.

3. The other misunderstanding that needs to be clarified is around Lin28b. Lin28b is a fetal-exclusive RNA-binding protein, which, in adults, is repressed by let-7 expression. Thus, Lin28b is NOT expressed in adults. The reason the authors did not find a publication on this is because it is known that Lin28b-KO in adults is uninformative, given that this gene is not expressed (instead, there is high expression of let-7, which, if removed, can promote B-1a development). Together, Lin28b, Arid3a, IGF2BP3, and others are required for B-1a development in early life, not adults. Thus, the conclusion by the authors that Lin28b expression is the same among all B-cell subsets based on the ImmGen App, is not quite correct. The problem is that all the values in the ImmGen data are likely to represent negative expression since Lin28b is not expressed in adult cells.

4. I used the word "moderate" based on the data shown in Fig. 1, which I addressed in my previous review, comment #2. For example, in Fig. 1i there are changes in the percentages of B-1 cells across WT, LEF1-KO, and DKO that appear moderate—this doesn't mean they are insignificant. But deserve attention given that they are based on total live cells, and not total CD19+ B cells (Fig. 1 j-l). For example, when B-1a decreases but B-2 does not increase, what other populations can increase (in only some mice) that affect the direct ratio of B-1/B-2? Myeloid cells?

5. The point I was trying to make in my comment #3 from the previous review is that TCF1/LEF1 does NOT seem to affect B-1a repertoire similar to how Bhlhe41 deficiency (or H-2DM deficiency) does. Instead, it decreases B-1a numbers. Hence, I would not emphasize changes in the repertoire of DKO mice if the reason for such a change is already known to be due to a decrease in B-1a. Thank you for addressing this.

6. Based on my previous comments regarding prior literature, do the authors think that TCF1/LEF1 play a role in Ig/BCR, given that DKO decreases BCR expression on all B-cell subsets, not only B-1a? Could that be part of the mechanism of B-1a effects observed here?

7. Based on the flow plots, there seems to be more than 50% decrease in the level of CD5 protein on the surface of B-1a cells. Do the authors think the decrease in IL-10 observed here is due to a significant reduction in CD5 expression on B-1a?

8. B-1 progenitors exist in the spleen before CD5 expression, which is turned on during the transitional stage before maturation. My original question was about precursors to B-1a cells, either CD5+ transitional B-1a (as the authors analyzed) or CD5-neg B-1a-specific progenitors, as I described. It would be interesting to look at their data to investigate the effect on

the previously described splenic B-1 progenitor. However, this is not necessary to validate the author's claims here.

B) Regarding Technical/data analysis:

1. The authors mention that using either log or bioexponential scales is "perfectly acceptable" to display digital flow data plotted in various figures. For example, in Fig. 1b, TCF1 and LEF1 seem each plotted on different scales, including on the bi-axial plot showing both proteins on different scales on the same plot. Log scale can generate artifacts in the lower end of the plot and has been replaced by the bio-exponential scale, which was developed to address this problem with the log scale when displaying cells with low expression values. Please, see the paper "Interpreting flow cytometry data: a guide for the perplexed" (Nature Immunol, 2006). It is good flow practice to use bio-exponential for fluorescent antibodies when analyzing markers that are not highly expressed to avoid artifacts of log. There are exceptions to this rule, but they don't seem to apply to this case. Linear scale for scatter (FSC) plots are fine and often recommended.

2. Re: BrDU, the new display using "dot plots" further supports my original comments. There are very few cells (or events) on the Ly5a-neg cells. Thus, the few dots above the "quadrant gates" could lead to the formation of contour plots above the quadrant line, even though they might not be biologically different from the main population. This difference can be appreciated when comparing the BrDU+ cells from the Ly5a+ population. There, the BrDU+ cells represent a separate population, while the Ly5a-neg cells have a global shift upwards of the entire population and may not represent two groups of BrDU+ and BrDU-. Unfortunately, the few events make it difficult to conclude whether you have a separate population of BrDU+ or whether those few dots are simply the distribution around the main population—this is the limitation of the assay. The authors' interpretation, however, is based on the quadrant gates showing percentages of 1.57 vs 0.11. These percentages can be skewed based on the other 3 populations that make up the quadrant gates, which when increased or decreased, affect the percentage of cells on that top left quadrant. Again, my comments are intended to improve clarity and reproducibility for the field/reader. Please see "Interpreting flow cytometry data: a guide for the perplexed" (Nature Immunol, 2006) for more details on the caveats for using quadrant gating and making gating decisions on a few events.

C) Regarding Human B-1 cells:

1. I appreciate all the human data presented here—they are indeed interesting. However, although possible, it seems unlikely that some adult humans would have >60% of B-1 circulating in blood. It is possible that the authors may be correct about the human B-1 classification in adult tissues and the diseases described. However, the proof is not contained within the current data, which relies on a set of surface markers. While I truly appreciate the efforts in classifying these cells in humans (much needed in the field!), I would be cautious about how to describe this classification to leave room for future studies that may provide more definitive evidence based on the developmental origin of these cells. Thus, personally, I wouldn't state that Human B-1 cells in blood/pleural were identified "with confidence".

2. The authors argument on the lack of CD38 expression is reasonable. While I agree that CD38 is generally expressed on activated, transitional, or plasma cells, the kinetics of CD38 can be complex. This surface protein is expressed during B-cell development, it is turned off during final maturation, and is only later re-expressed on plasma cells. Is it possible that blood cells expressing TCF1/LEF1 are activated cells that have been blocked from plasma cell development? For example, would TLR activation lead to IL-10 secretion while a fraction of these cells remain CD38 negative?

Referees' comments:

Referee #1 (Remarks to the Author):

This study by Shen and colleagues describes work on mice single and double-deficient in TCF-1 and LEF1. The investigators used these mice to assess the developmental path and functionality of B cell subsets, concluding that the two transcriptional regulators are required to drive CD5+ B-1 cells to develop, expand and function, but that other B cell subsets are unchanged.

TCF-1 has been shown previously to work in conjunction with LEF1 to regulate transcription and its effect in T cells has been explored. Relevant for this study Tcf1 in B cells has been shown to be regulated by EBF1, thus there is evidence of regulation in B cells. Its function in T cells demonstrated among other observations their negative regulation of Blimp-1 expression and support of Bcl6, factors that are critical also for the regulation of B cell functions, and perhaps could explain some of their data demonstrating a reduction in IL10 production by B-1 cells, although this has not been pursued. They also suggest that TCF-1/LEF1 regulates oxidative and fatty acid metabolism and that without these transcriptional regulators Ets1/myc expression in B-1 cells thereby allowing these cells to self-renew, a property of fetal-derived B-1 cells that has not been elucidated before. Finally, they show that the double-deficient B-1a cells show reduced functionality, especially a failure to strongly induce IL10 production.

The study uses flow cytometry, adoptive transfer studies, single cell RNA sequencing and various omics approaches for their analyses of single and double transcription factor knock out mice. Experiments show a sufficient number of replicates, and statistical analysis appears appropriate for the data shown. The study shows a reduction of CD5+ B-1 cells in the body cavities and spleen both in the double-KO animals and following adoptive transfer of fetal liver and bone marrow transfer. They also indicate a direct role for these transcriptional regulators in the expression of CD5 itself and demonstrate a failure of B-1a cells to be activated by LPS to generate IL10, although steady-state IL10 production appeared unaffected.

Despite these far-reaching conclusions, and many interesting and novel data provided, there are incongruencies in the manuscript, inconsistencies, a failure to appropriately integrate all their findings, and a lack of supporting details, that make it difficult to interpret all their data and reconcile some of their findings. Ultimately, I find the study confusing, even upon re-reading, and I am not convinced that they can make the claims with the data they show. I will highlight some of those inconsistencies and omissions below.

We have gained a good understanding that B-1 cell development is regulated by regulators of fetal hematopoiesis.

We thank the referee for the comments of the many interesting and novel data provided. The referee is concerned with a lack of supporting details, that make it difficult to interpret all the data and reconcile some of our findings and finds the study confusing. We have taken these comments seriously and undertaken the required experiments to answer the questions posed.

1. No attempts have been made to integrate their data with those on Lin28/Let7 findings.

We addressed this question by comparing expression of Lin28b, Bhlhe41, Tcf7 and Lef1 at the different B-1 cell developmental stages by scRNA-seq. We individually enriched, barcoded then pooled B cells subsets including: i) foetal liver B-1 progenitors (FLB1P), ii) adult bone marrow B-1 progenitors (BMB1P), iii) peritoneal cavity B-1 cells (PCB1) and iv) spleen B-1 cells (SPB1). As shown in Fig. 3f, the results reveal differences in expression of *Lin28b* and TCF1-LEF1 during B-1 cell

ontogeny: *Lin28b* appears to be expressed in foetal liver B-1 progenitors, and nearly extinguished thereafter, or expressed at low levels to be detectable by scRNA-seq in the subsequent stages of B-1 cell development. By contrast, *Lef1* is expressed strongly in B-1 progenitor from both foetal liver and bone marrow and maintains a low but detectable level of expression in peripheral B-1 cells. *Tcf7* is very lowly expressed in the progenitor stages and is higher in peritoneal cavity and spleen B-1 cells, suggesting a more important role in these mature stages, which is consistent with our findings that TCF1 is important for the generation of the stem-like memory B-1 cells upon B-1 cell activation, to prevent exhaustion.

Figure 1 claims to show the expression of *tcf7* (encoding *tcf1*) and *Lef1* in CD5+ B-1 cells. Yet, confusingly, and not explained at all, *Tcf7* and *lef1* expression are mutually exclusively expressed in their scRNAseq data and only *tcf7* expression seems to fall within the CD5+ cluster. They brush that aside by showing histogram plots of TCF1 and LEF1 staining that shows some shifts in histogram plots I do not find at all convincing. Moreover, they show similar “enhanced” expression in B-1a and B-1b compared to B-2 cells, although all their data then go on to show the effect of these transcription factors only on B-1a but not B-1b.

The correlation between expression levels of protein and mRNA in mammals is relatively low, with a Pearson correlation coefficient of ~0.40, which means that mRNA is often not a reliable predictor of protein abundance. Indeed, there is accumulating evidence for significant regulation and control of steady-state protein abundances after mRNA is made — including post-transcriptional, translational and protein degradation regulation¹. This is often the case for transcription factors².

Following the reviewer’s suggestion we now add several different flow cytometric plots for the readers to visualize TCF1 and LEF1 expression with the different B-1a markers in 2D plots, rather than histograms (see below); these show that high TCF1/LE1 high cells are largely B-1a:

- Gating of TCF1^{high} B cells in peritoneal cavity selectively identifies CD5^{hi} CD43⁺ B-1a cells. New Fig. 1b and Extended Fig. 1c top panel
- Gating of LEF1^{high} B cells in peritoneal cavity selectively identifies CD5^{hi} CD43⁺ B-1a cells. New Fig. 1b and Extended Fig. 1c bottom panel
- The highest TCF1-expressing cells in spleen are CD19⁺B220⁻ that correspond to B-1 cells (New Extended Fig. 1d).

2. Fig. 1e shows data on total B-1 (unclear what this is) and then B-2 and B-1a and B-1b in body cavity but only B-1a in the spleen. If they can identify total B-1 as well as B-1a, why do they not also show B-1b data for the spleen?

For clarity, we now show the FACS plots used to gate these cells in the main figure. As reported by others^{3,4}, whilst the demarcation between B-1a and B-1b is clear in the peritoneal cavity, B-1b cells cannot be convincingly identified in the spleen only based on CD5^{lo}. Thus, we only refer to total B-1 cells in the spleen.

Inconsistent with that discrepancy in the scRNAseq data are their own data that show distinct effects of *tcf1* and *Lef1*. While *Lef1* deficiency reduces CD5 B-1 cells in both spleen and body cavity, *tcf1* deficiency affects spleen but not body cavity CD5+ B-1 cells and the *Lef1* single KO mice on are largely recapitulating what they see in the double KO. Despite that they claim throughout the need for both factors (consistent with the literature perhaps but not with their data). This is confusing and needs to be clarified, as both their transcriptional data and functional data show differential effects, yet they claim that these factors act in a common fashion.

We thank the reviewer for asking us to clarify this point. The explanation for the shared albeit non-completely overlapping and for some functions non-redundant effects of TCF1 and LEF1, is likely to lie in the kinetics of expression of these two transcription factors during development.

In a new experiment shown in Fig. 3c-f, we have performed single cell RNA sequencing (scRNA-seq) and lineage trajectory analysis of B-1 cells at different stages of development and assessed the expression of TCF1 and LEF1 at each stage. First, we could confirm that spleen B-1 cells harbour

some cells that co-localized with foetal liver B-1 progenitors, suggesting these are indeed splenic B-1 progenitors (see above figure) as also observed by Dear TN et al.^{5,6}. The bulk of splenic B-1 cells however cluster with peritoneal cavity B-1 cells suggesting they are mature.

Given that the different populations were barcoded, directionality can be inferred in the trajectory analysis. When we look at LEF1/TCF1 expression at the different stages and or locations, we see that B-1 progenitors from both foetal liver and bone marrow express abundant Lef1 but little Tcf7 while Tcf7 is expressed at higher amounts by B-1 cells in peritoneal cavity and spleen (new Fig. 3f), with reduced – but still present – Lef1 expression in these cells. This suggests that Lef1 is probably more important for B-1 development, which is consistent with our Lef1-deficient mouse data, and TCF1 may be more important for peripheral maintenance and formation of stem-like B-1a cells, which is consistent with our in vitro B-1 cell activation data.

These expression data support our finding that double TCF1 and LEF1 deficiency resulted in a more profound B-1 cell depletion in the spleen than TCF1 single deficiency.

We have also performed additional experiments to address potential differences that may be ascribed to either one or both TFs. RNA-seq on peritoneal B-1a cells from TCF1KO, LEF1KO and dKO now shown in Fig. 3h and Extended Fig. 4b demonstrated that both TCF1 single and TCF1/LEF1 double deficiency B-1a cells have significant effects on cell cycle and E2F pathway while LEF1 single

deficiency in B-1a cells influences IL-6-STAT3, TGF- β and TNF α (Fig.3h and Extended Fig.4b). IL-2/STAT5 signalling pathway was affected in all mutant - TCF1KO, LEF1KO and dKO - B-1a cells.

In a separate new analysis of human prenatal tissue scRNA-seq data we show that the cells that co-express TCF1 and LEF1 are not only the highest expressors of the B-1a markers CD5 and CD43 (encoded by SPN) but also express the highest amounts of IL7R and IL2RG (new Fig. 3g). This suggests that both TFs contribute to B-1a cell functions.

Identification of “B-1 cell precursors” in the bone marrow and spleen is very challenging and it is very unclear how they derived these data, particularly given that both mature B-1 cells and B-1 derived antibody secreting cells as well multiple stages of B-2 cells show overlapping phenotypes. Flow cytometry data to support each claim of their manuscript stating reductions in precursor populations must be included and must convincingly demonstrate that these cells are indeed B-1 cells. As written, I do not find their claim that TCF1 and LEF1 promote B-1a cell development from the earliest precursors (line 136/137) substantiated with sufficient data.

Prompted by this comment from the reviewer we have edited the nomenclature to be concordant with that in the literature, naming the B1 precursor subsets from foetal and neonatal BM “B-1 progenitor” and we now show the gating strategies for B-1 progenitors as described in the literature in Extended Fig. 2a. Although our data showing a block in differentiation of B-1 progenitor, we cannot definitely conclude that TCF1/LEF1 are required for B-1a cell differentiation from foetal liver progenitors given the identification of both bone marrow and splenic B-1a precursors require CD5 expression and we have shown that TCF1/LEF1 deficiency causes a reduction in CD5 expression.

That said, their fetal liver transfer data are perhaps most consistent with their claims and go a long way in convincing that there are indeed defects on CD5+ B-1 cells. Yet, whether this is truly a developmental defect is difficult to discern and in contrast to the title and abstract, and some of the figures (which make those claims), their own conclusion on line 66 states that these transcription factors regulate self-renewal and thus perhaps enhances expansion. (development or expansion?)

The reviewer asks the important question of whether TCF1 affects both development and self-renewal to achieve a replete pool, or only the latter. In other words, is self-renewal required for B-1 cell maintenance or also required for the development of a mature B-1a compartment? Whist we

cannot definitively conclude that TCF1/LEF1 are required for development of B-1a cells from foetal liver progenitors as mentioned above, there is evidence that TCF1/LEF1 are required for the formation of a replete peripheral mature B-1a cell pool: Foetal liver cells and adult bone marrow cells have reduced capacity to reconstitute the B-1a repertoire of Rag1-/- recipients.

We have been cautious and reworded the text throughout to emphasize the perceived developmental defects may be due to defects in self-renewal rather than early differentiation of B-1 cells.

3) The investigators show in Fig. 2a a differentiation pathway of B-1 cells that is supported by the literature. Yet, in their text in multiple parts of the manuscript (for example line 48) they imply that peritoneal cavity CD5+ B-1 cells give rise to “their splenic offspring”. In contrast, existing evidence suggest that splenic B-1 cells precede in development peritoneal cavity B-1 cells and that elimination of the spleen (see Carsetti et al) will cause a disappearance of B-1 cells in the body cavity over time, but not the other way around. Only in response to cytokines and/or TLR do B-1 cells accumulate in the spleen, largely as effectors. Clarifications of these statements must be done.

Firstly, we thank the reviewer for pointing the need for us to clarify this. We meant that peritoneal B-1a cells migrate to the spleen after stimulation, where their offspring plasma cells secrete higher amount of protective antibodies and cytokines. We have made sure to cite the work by Carsetti et al.

In terms of development, our lineage trajectory by pseudotime analysis reveals that foetal liver, bone marrow, spleen, and peritoneal cavity all harbour cells that co-localize with foetal B-1 progenitors (Fig. 3e), although this does not prove that their presence is sufficient to repopulate the overall B-1a cell population.

The staining for Lef1 and tcf1 by flow cytometry must be shown. Their data reported gMFI (Fig. 2f) suggests at best extremely minor differences in expression. This might be biologically meaningful, but as shown I find such data unconvincing, and they fuel suspicion of overstatements throughout the paper that are troubling.

We now show all the FACS plots (Fig. 1b and Extended Fig. 1c-d). The reader will be able to see that TCF1 and LEF1 expression is a feature of CD23-negative B cells in the peritoneal cavity (B-1 cells), and that the high expressors are virtually all CD5+, and thus B-1a cells. The picture is similar in the spleen as high TCF1 and LEF1 expressing cells are B-1 cells, albeit LEF1 expression is lower.

We now have also added stains of human B cells from both pleural fluid drained from patients with bacterial pleural infection and also their peripheral blood. We show that CD43+CD5+ cells stain positive for both TCF1 and LEF1 (new Fig. 1e and Extended Fig. 1i) are highly enriched in pleural cavity, and also in blood from patients with bacterial pleural infection, strongly supporting these are indeed B-1a cells.

Similarly, Ki67 expression is induced in cells entering G1. How meaningful is a shift in staining so small that gating cannot be done and expression of the entire population is shown as small changes in gMFI. Ki67 staining is usually giving good distinctions between positive and negative cells. As such their data are unconvincing as shown. FACS gating to transitional B-1a cells and the facs plots

showing Ki67 staining should be shown to clarify, because in 2f they then state that nearly 100% of B-1a cells are Ki67+, but the tissue of B-1 origin is unclear.

We now show the FACS plots so that the reviewer and reader can easily see the positive and negative populations. It now becomes clear that practically all Ki67 positive cells are found within the CD5+ (B-1a) subset (new Fig. 2d). We show that Ki67 positive B-1a cells decline with age, which parallels a decline in TCF1 and LEF1 expression (new Fig. 2e).

With regards to the transitional B-1a cells, we now also show the FACS plots so readers and the reviewer will be able to clearly see that again, the vast majority of CD5+ cells are Ki-67 positive (Extended Fig. 2f).

5) Their statement that B-1 cells are the normal counterparts of CLL in humans is an unsubstantiated claim and the papers they are referencing do not demonstrate that.

There is indeed controversy regarding the existence of B-1 cells in humans⁷⁻⁹. Having established that the CD19⁺ CD43⁺ CD38⁻ CD5⁺ combination identifies B-1 cells in human pleural cavity and in the blood of patients with bacterial pleural infection, we have applied this gating strategy to the blood of patients with B cell chronic lymphocytic leukaemia (CLL). We show that in most patients, the neoplastic B cells co-express CD43, CD5 and high amount of TCF1 and LEF1 compared to the residual normal B cells in the same patients (Fig. 1e & h). There are patients however with CLL B cells only expressing one or the other transcription factor, which may reflect the stage of B-1 cell development at which the leukaemia arose. By contrast, B-1 cells are not seen in peripheral blood of healthy subjects. We are cautious with the conclusions and are not claiming that these are B-1 cells, but it is fair to say that these results support a B-1 cell origin of CLL.

6) They claim to find reductions also in “human B-1 cells”, using gating strategies that have been largely dismissed by the field as inappropriate. Indeed, no convincing human orthologues of B-1 cells have been isolated from adult humans to-date, while two recent studies with human fetal tissue identify potential orthologues. This data should be removed.

As described above, we now provide stronger evidence supporting the existence of human B-1 cells:

- We show human pleural cavity B cells (above and in new Fig. 1e-h) in which there is a large subset with a phenotype identical to that of mouse peritoneal cavity B-1a cells.
- In patients with bacterial pleural infection, cells with a B-1a phenotype expand in the blood
- Amongst blood cells with a B-1a cell phenotype, 6-13% of cells bind phosphatidyl choline (PtC) – a feature of mouse B-1a cells, whereas less than 0.5% of other blood B cells do
- Also, only recently, our co-authors Muzlifah Haniffa, Vijaya Baskar Mahalingam Shanmuglah identified human B-1 cells based on RNA profiles in human prenatal tissues and described them as such¹⁰. Reanalysis of those B-1 cells identified four subsets based on expression of TCF1 and LEF1. We found that TCF1 and LEF1 double positive cells are the highest expressors of CD5 and SPN (CD43) (Fig. 3g).

7) Line 147 “failure to accumulate” seems an overstatement. Instead, a convincing reduction in B-1a cell expansion seems to occur. Suggest modifying this statement.

We have now modified this statement: “In line with a failure to expand in adult life, peritoneal B-1a cells lacking TCF1 and LEF1 failed to expand.”

8) Line 185 states unchanged frequencies of B-1 cell-derived plasma cells. However, extended Fig. 2b indicates significant enhanced plasma cell frequencies in the double KO mice?

We wonder whether the reviewer was looking at a different figure. In our original extended Fig.3b (currently Extended Fig. 3e), the frequency of B-1 plasma cells (B-1PC) is comparable between WT and double KO mice.

9) The investigators state that the repertoire of B-1a cells is more diverse in mice lacking the 2 transcription factors (Fig. 2k). Yet this analysis is done (if I understand correctly) of total peritoneal cavity B cells, not B-1(a) cells and include B-2? Is that correct? Please clarify

We thank the reviewer for this comment, and we apologise for the oversight, we have corrected this to say total peritoneal cavity B cells.

10) If indeed the diversity of the clones were to expand as shown in Fig. 2k, it seems inconceivable that the % of PtC binders among B-1a cells would not at all be affected. The authors should clarify this point and how shifts in diversity would not affect % of PtC binders (which they show in their hands to be >40% of all B-1a cells).

First, for clarification, the total number of PtC-binding peritoneal B cells was significantly reduced consistent with a decrease in total B-1a cells in dKO mice (now shown in Fig. 2j), but the percentage of PtC binders amongst B-1a cells was maintained (extended Fig. 3a).

To get a more complete picture we have now performed scBCR-seq on peritoneal B cells. We find an overall a more diverse repertoire in total peritoneal B cells from TCF1/LEF1 deficient mice probably due to the overall decrease in the more clonal B -1a cells and overrepresentation of B-1b and B-2 cells. By contrast, the B-1 cell BCR repertoire appears less diverse in TCF1/LEF1 in dKO mice, which suggests that TCF1 and LEF1 may promote B-1a clonal expansion. The mechanisms underpinning these effects of TCF1 and LEF1 are currently unknown.

11) Fig. 3a shows a very pronounced decrease in IgM-BCR expression among double KO peritoneal B-1a cells. While they seem to suggest effects of class-switching, the reduction seems to affect the entire cell pool. Would this reduction not explain perhaps the changes in signalling etc they are seeing in their data?

Yes, the reviewer is correct, and we thank them for raising this interesting point. Indeed, the mean fluorescence intensity of Ig light chain expression (Ig κ or Ig λ) within B-1a cells is lower, regardless of the isotype that is expressed; these differences are not seen in B-2 cells. This is now included in new Extended Fig. 2j. As the reviewer suggests, this decrease in BCR expression caused by TCF1/LEF1 deletion is likely to contribute to the signalling changes and we have now explicitly mentioned this.

12) The direct effects of the transcriptional regulators on cd5 is important to understand the selective effect on the CD5+ B-1 cell populations and their facs-plots clearly support their

transcriptional data. If CD5 expression is affected, given it is the sole means of identifying B-1 cells in this study – would that not alone then explain the differences they see, given the known role of CD5 as a regulator of TCR/BCR signaling strengths?

It is indeed interesting that TCF1 and LEF1 regulate CD5 expression, and as the reviewer points out this is likely to contribute in some degree to changes in BCR signaling. The regulation of CD5 is relatively modest (see CD5 MFI of B-1a PtC-binding B cells below) so it is unclear that it can explain the entire phenotype. In T cells, TCF1 and LEF1 are known to regulate multiple pathways beyond CD5/TCR signaling, which is likely to be the case in B cells. We have nevertheless been cautious in the text and appropriately mentioned a likely important contribution of CD5 downregulation.

13) The expression of CTLA4 and PD-L1 by B-1 cells was shown previously to have significant functional effects by the Herzenberg group (Yang et al.) and studies by the Haas group. These manuscripts should be cited.

We thank the reviewer's suggestion, and we have cited these papers (references ¹¹⁻¹⁴).

Referee #2 (Remarks to the Author):

There has been much interest in the molecular understandings that govern different B and T cell subsets ability to self-renew or differentiate. Here, Shen et al., demonstrate this ability in a subset of self-renewing B cells, B-1a cells. TCF1, a transcription factor known for sustaining precursor cells in CD8 and CD4 T cells, and LEF1 deficiency were shown to specifically decrease the B-1a population using conditional knockout mice. This decrease in B-1a cells was shown to be due to an inability of B-1a precursors (B-1P) to differentiate into B-1a cells as well as an intrinsic proliferation defect of B-1a cells. Transcriptional and ATAC-seq analysis of TCF1/LEF1 double knockout or B cell transgenic B-1a cells revealed direct regulation of Myc and altered metabolic pathways governed by the two transcription factors. Furthermore, B-1a cells production of IL-10 was found to be promoted by TCF1/LEF1, and transfer of these B cells into EAE mice resulted in worse clinical scores. TCF1/LEF1 deficient B-1a take on an "exhausted phenotype" and failed to properly differentiate into plasma cells. While the general finding of TCF1 function is not novel, this paper expands this knowledge to include these transcription factors to self-renewing B-1 cells. However there are several points that are not properly addressed and require more clarification.

We thank the reviewer for their overall positive comments.

Comments:

1) Their data suggests that of Tcf1 and Lef1 may have different roles contributing to B-1a B cell development and differentiation. Yet this critical point is not addressed and remains unclear as most of this study was performed using Tcf7 and Lef1 knockout mice. Fig 1a shows that there is not much overlap in cells expressing both transcripts and Lef1 does not appear to overlap with the B-1a subset, based on Cd5. The single knockout of Lef1 has a more drastic effect on B-1a frequency in the peritoneal cavity than Tcf1 single knockouts. Could Tcf7 and Lef1 be expressed at different stages of B-1a development, or even promote different subsets (such as a Mychi and Myclo). Is there a

difference in the kinetics of TCF1 and LEF1 expression during B-1a development? This premise leads to the possibility that Tcf1 may regulate self-renewal and Lef1 may regulate function. Parsing out the difference between these factors may provide better understanding of the transcriptional differences dictating self-renewal and function of B-1a cells, and how this effects populations in the peritoneal cavity versus the spleen.

The reviewer raises a very important point that we have addressed with several new experiments:

- i. In order to assess whether LEF1 and TCF1 were expressed at different stages in B-1 cell development, we performed scRNA-seq from individually enriched, barcoded then pooled B cells subsets including: i) foetal liver B-1 progenitors (FLB1P), ii) adult bone marrow B-1 progenitors (BMB1P), iii) peritoneal cavity B-1 cells (PCB1) and iv) spleen B-1 cells (SPB1). Given that the different populations were barcoded, directionality could be inferred in the trajectory analysis.

Analysis of LEF1/TCF1 expression at the different stages and or locations, reveals that B-1 progenitors from both foetal liver and bone marrow express abundant Lef1 but little Tcf7 while Tcf7 is expressed at higher amounts by B-1 cells in peritoneal cavity and spleen (new Fig. 3f), with reduced – but still present – Lef1 expression in these cells. This suggests that Lef1 is probably more important for B-1 development, which is consistent with our Lef1-deficient mouse data, and TCF1 may be more important for peripheral maintenance and formation of stem-like B-1a cells, which is consistent with our in new in vitro B-1 cell activation data (below).

- ii. These expression data support our finding that double TCF1 and LEF1 deficiency resulted in a more profound B-1a cell depletion in the spleen than TCF1 single deficiency:

- iii. We have also performed additional experiments to address potential differences that may be ascribed to either one or both TFs. RNA-seq on peritoneal B-1a cells from TCF1KO, LEF1KO and dKO now shown in Fig. 3h and Extended Fig. 4b demonstrated that both TCF1 single and TCF1/LEF1 double deficiency have significant effects on cell cycle and E2F pathway while LEF1 single deficiency in B-1a cells influences IL-6-STAT3, TGF- β and TNF α (Fig.3h and Extended Fig.4b). Intriguingly, IL-2/STAT5 signalling pathway was affected in all - TCF1KO, LEF1KO and dKO - B-1a cells.

- iv. In a separate new analysis of human prenatal tissue scRNA-seq data we show that the cells that co-express TCF1 and LEF1 are not only the highest expressors of the B-1a markers CD5 and CD43 (encoded by SPN) but also express the highest amounts of IL7R and IL2RG (new Fig. 3g). This suggests that both TFs contribute to B-1a cell functions.

- v. With regards to the function of these transcription factors during B-1 cell activation, we have repeated the experiments using single LEF1 and TCF1-deficient B-1 cells. In new Fig. 5h we show that TCF1-deficient B-1 cells are less capable of generating Myc^{lo} cells in the third division compared to LEF1-deficient B-1 cells, suggesting that TCF1 is more important than LEF1 in formation of stem-like B-1 cells.

2). The double knockout mice exhibit functional differences to wild type B-1a cells in terms of IL-10 production and PD-L1 expression. This idea is tested in the EAE model and the abstract claims "TCF1 and LEF1 are also essential for IL-10 and PD-L1 expression upon B-1 cell activation and repression of CNS inflammation." While the clinical score is clearly impacted by transfer of the different B-1 B cells there is no data to demonstrate the mechanism (i.e role of IL-10 or PD-L1) behind this result. Do the transferred B cells maintain PD-L1 and IL-10 production. Since the figure and legend state B-1 B cells are transferred and not just B-1a are they differentiating into other B cells that impact the clinical score. Are they secreting systemic amounts of IL-10, entering the CNS or is this effect promoted in

secondary lymphoid organs. Further assessment of the B-1a cell's role in providing protection should be assessed.

These are tricky experiments to do. Whilst the role of B cells in EAE has been studied by many groups, very few have managed to identify the B-1a cells after transfer in peripheral lymphoid tissues or brain ¹⁵. This may be because as we and others have shown ¹⁶, once B-1a cells become activated, they change their phenotype quite profoundly including changes in B220 and CD5 expression (and thus commonly used congenic markers such as CD45.1/2). To overcome this problem, we have generated a new mouse model in which B-1 cells can be tracked due to expression of a fluorochrome driven by a marker we have found to be only expressed in peripheral B-1 cells: Gramd2a

We confirmed that Gramd2a identified B-1 cells comprising the majority of B-1a and B-1b cells (>90% and >75% respectively) in the peritoneal cavity.

We adoptively transferred activated B-1 cells from *Gramd2^{mApple-Cre}* and did immunofluorescence of brain sections 21 days after EAE induction. We observed the donor derived B-1 cells (mApple+) in the brain which predominantly located underneath the meninges or in the nearby parenchyma (new Fig. 4o and Extended Fig. 5e-f).

Analysis of B-1a-derived IL-10 in the circulation of recipient mice remains challenging and can be the subject of future studies.

3) The results are difficult to follow as the experiments oscillate between different tissues such as peritoneal cavity, spleen and bone marrow as well as different precursor and mature (B-1a vs B-1). While B-1a are typically thought to have more impact in the peritoneal cavity than spleen, the rationale at times for looking in different tissue is hard to follow. Fig 2a does a good job of explaining this point, but it does not get into why the authors look in the spleen in reference to the B 1-a cells in figs 1, 2, and 4. It is also not made clear if there are difference in B-1a cells that arise from conventional fetal liver pathway versus bone marrow precursors.

We have performed a number of additional experiments to address this that have considerably improved our results:

- i) We now show the FACS plots/gating strategies to convincingly demonstrate that TCF1 and LEF1 are highly expressed in B-1a cells in mouse peritoneal cavity, with B-1 cells present in the spleen:

- ii) We also performing stains (in the same tube) of peripheral blood mononuclear cells and pleural effusion B cells from human patients with bacterial pleural infection, revealing a clear CD43⁺CD5⁺ population that has phosphatidyl-choline (PtC) reactivity and expresses high amounts of TCF1 and LEF1 in both compartments. This population is thus very likely to be B-1 cells. By contrast, in healthy individuals there are very few of these B-1 cells circulating in the blood. We also show that B-1 cells are also rare, albeit visible, in human mucosal associated lymphoid tissues like adenoids. Furthermore, we show that the malignant expanded B cells in chronic lymphocytic leukaemia (CLL) also share a B-1 cell phenotype, with the majority co-expressing TCF1 and LEF1 (new Fig. 1e-h).

As explained above, we also looked for transcriptional differences by scRNA-seq in B-1 cell progenitors in foetal liver vs bone marrow, and peripheral peritoneal cavity vs spleen B-1 cells (new Fig. 3c,d), and found that the progenitors cluster close together without a complete overlap, and the

peripheral spleen and peritoneal cavity cells also cluster close together, indicative of shared but not identical transcriptomes (new Fig. 3c). Trajectory analysis (new Fig. 3d) suggests a shared developmental pathway, but further work would be needed to understand the differences.

Minor:

The concluding paragraph is focused on asymmetric cell division, which parallels these findings with that of CD8 T cells. However, their data does not investigate their role of asymmetric cell division and their final conclusion should not assume this is true.

This is a good point: we have now removed discussion about asymmetric cell division so that there is no confusion on this point.

Fig 1e,f: condensing the graphs so all the groups can be shown together would be ideal. Or else, a more salient determinant of the single deficiency groups.

Thank you for this suggestion, which has now been done in new Fig. 1j-l:

Fig 2k demonstrates that KO cells are more diverse as adults, while being similar to WT in neonates. What is the speculation of why this is?

Clonality had been performed in total peritoneal CD19+ B cells, rather than only in B-1a cells. As B-1a cells – which are clonally restricted - decline with age in the KO, B-2 cells which are more diverse, are overrepresented, thus the overall diversity of the total peritoneal B cell pool increases. We hope this is more easily visualized in new Fig. 2h:

In figure 5a the authors show that blocking IL-10 inhibits the PC:non-PC ratio implying it is due to lack of IL-10. Fig 4e shows reduced IL-10, not an ablation, so if the IL-10 is added back will it restore the phenotype. Is it possible that while IL-10 is reduced, TCF1/Lef1 also impacts the ability of these B cells to respond to IL-10 and inhibit their PC differentiation?

We thank for the reviewer's interesting suggestion. We have now added back IL-10 in both TCF1^{WT}LEF1^{WT} and TCF1^ΔLEF1^Δ B-1 cells and find that IL-10 can restore plasma cell differentiation of B-1 cells in the absence of TCF1 and LEF1, suggesting an autocrine/paracrine role of TCF1/LEF1-driven IL-10 secretion in B-1a cells (new Fig. 5d).

Referee #3 (Remarks to the Author):

Qian Shen and co-authors used double-knockout mice and adoptive transfers (bone marrow and fetal liver chimeras) and a series of cellular and molecular readouts to define the role of the transcription factors TCF1 and LEF1 in B-1a cell development, self-renewal and regulatory functions. The authors concluded that TCF1 & LEF1 specifically drive the formation and self-renewing of B-1a cells. They also showed that TCF1/LEF1 deficiency down-regulates Ets1 and Myc targets, leading to the expression of an exhausted B-1a cell phenotype similar to age-associated B cells.

Although the observations in the TCF1/LEF1 double-knockout (DKO) mice are interesting, particularly the exhausted B-cell phenotype, the rather overall modest effect on B-1a, including minimal effect in the development of the archetypal B-1a repertoire (VH11/VH12/anti-PtC), natural antibody levels, and splenic B-1a numbers, does not support the main premise of this study. Much of the observed phenotype and putative decrease in B-1a numbers in the TCF1/LEF1 DKO mice could be attributed to their effect on down-regulating the expression levels (MFI) of CD5, CD19 and IgM instead of a direct effect on B-1a development

We are glad that the referee found our finding interesting. We have performed a series of experiments to address their concerns.

Major concerns:

1. The authors claim the mechanisms underpinning B-1a development and self-renewal are unknown. However, several groups have shown the role of Bhlhe41 (Kreslavsky et al. Nat Immunol, 2017), Bmi-1 (Kobayashi et al., JI 2020), CD19 (Rickert et al., Nature 1995; Haas et al., Immunity 2005), Btk and other genes in B-1a development and self-renewal. These previous studies show a stronger effect on B-1a development and self-renewal than the TCF1/LEF1 DKO shown here. More importantly, in these previous studies, the targeted knockout gene directly affected the development of canonical B-1a repertoire (including VH11, VH12, and/or anti-PtC). In contrast, the TCF1/LEF1 DKO did not significantly affect the development of the archetypal B-1a repertoire.

In terms of transcription factors, Bhlhe41 and Bmi-1 have indeed been reported to regulate B-1a self-renewal; we had cited Kreslavsky's paper and we now also cite Kobayashi's paper. We have changed our statement about the mechanisms as being "incompletely understood". Our discovery that two transcription factors, TCF1 and LEF1, until now only described to function in T cells and never known to exert a function in B cells, play important roles in the maintenance of a normal sized B-1a cell pool, in B-1a self-renewal and in B-1a cell regulatory function is indeed novel. TFC1/LEF1 deficiency reduces all clones proportionally (those that are archetypal and those that are not), impairs formation of a B-1 stem-like Myc^{lo} population likely to be key to B-1 self-renewal, causes exhaustion of B-1 cells, and impairs B-1a cell expression of IL-10 and PD-L1, reducing their regulatory potential and ability to repress CNS inflammation. We believe these are important and original findings in their own right, that do not diminish all the previous work in the field.

2. Authors claim that TCF1 and LEF1 are highly expressed in B-1 cells. Although this is true for TCF1, their data, including from ImmGen, shows LEF1's highest expression on follicular B-2, memory B, and B-2 transitional cells, not B-1a (Fig. 1a, Ext Fig. 1b). This discrepancy was not directly addressed. Although TCF1 is mainly expressed in B-1a, the TCF1-single KO does not significantly affect B-1a development; only the DKO with LEF1 shows a mild decrease in B-1a. However, LEF1 is highly expressed in B-2 cells, not B-1a (Fig. 1e).

The correlation between expression levels of protein and mRNA in mammals is relatively low, with a Pearson correlation coefficient of ~ 0.40 , which means that mRNA is often not a reliable predictor of protein abundance. Indeed, there is accumulating evidence for significant regulation and control of steady-state protein abundances after mRNA is made — including post-transcriptional, translational and protein degradation¹ regulation. This is often the case for transcription factors².

Following the reviewer's suggestion we now add several different flow cytometric plots for the readers to visualize TCF1 and LEF1 expression with the different B-1a markers in 2D plots, rather than histograms (see below), that shows high TCF1/LEF1 high cells are largely B-1a:

- Gating of TCF1^{high} B cells in peritoneal cavity selectively identifies CD5^{hi} CD43⁺ B-1a cells. New Fig. 1b and Extended Fig. 1c top panel
- Gating of LEF1^{high} B cells in peritoneal cavity selectively identifies CD5^{hi} CD43⁺ B-1a cells. New Fig. 1b and Extended Fig. 1c bottom panel
- The highest TCF1-expressing cells in spleen are CD5⁺ that correspond to B-1 cells (New Extended Fig. 1d).

The explanation for the shared albeit non-completely overlapping functions and non-redundant effects of TCF1 and LEF1, is likely to lie in the kinetics of expression of these two transcription factors during development.

In a new experiment shown in Fig. 3c-f, we have performed single cell RNA sequencing (scRNA-seq) and lineage trajectory analysis of B-1 cells at different stages of development and assessed the expression of TCF1 and LEF1 at each stage. First, we could confirm that spleen B-1 cells harbour some cells that co-localize with foetal liver B-1 progenitors, suggesting these are indeed splenic B-1 progenitors (see above figure) as also observed by Dear TN et al.^{5,6}. The bulk of splenic B-1 cells however cluster with peritoneal cavity B-1 cells suggesting they are mature.

Given that the different populations were barcoded, directionality can be inferred in the trajectory analysis. When we look at LEF1/TCF1 expression at the different stages and or locations, we see that B-1 progenitors from both foetal liver and bone marrow express abundant Lef1 but little Tcf7 while Tcf7 is expressed at higher amounts by B-1 cells in peritoneal cavity and spleen (new Fig. 3f), with reduced – but still present – Lef1 expression in these cells. This suggests that Lef1 is probably more

important for B-1 development, which is consistent with our *Lef1*-deficient mouse data, and *TCF1* may be more important for peripheral maintenance and formation of stem-like B-1a cells, which is consistent with our in vitro B-1 cell activation data.

These expression data support our finding that double *TCF1* and *LEF1* deficiency resulted in a more profound B-1a cell depletion in the spleen than *TCF1* single deficiency (New Fig. 1j,l). In this figure, one can also appreciate that *TCF1*-deficiency does affect significantly the numbers of B-1 cells in the spleen.

We have also performed additional experiments to address potential differences that may be ascribed to either one or both TFs. RNA-seq on peritoneal B-1a cells from *TCF1*KO, *LEF1*KO and dKO now shown in Fig.3h and Extended Fig.4b demonstrated that both *TCF1* single and *TCF1*/*LEF1* double deficiency B-1a cells have significant effects on cell cycle and E2F pathway while *LEF1* single deficiency in B-1a cells influences IL-6-STAT3, TGF- β and TNF α (Fig.3h and Extended Fig.4b). Intriguingly, IL-2/STAT5 signalling pathway was affected in all - *TCF1*KO, *LEF1*KO and dKO-B-1a cells.

In a separate new analysis of human prenatal tissue scRNA-seq data we show that the cells that co-express TCF1 and LEF1 are not only the highest expressors of the B-1a markers CD5 and CD43 (encoded by SPN) but also express the highest amounts of IL7RA and IL2RG (new Fig. 3g). This suggests that both TFs contribute to B-1a cell functions.

3. CD5 expression levels (measured by flow cytometry MFI) are drastically decreased in TCF1/LEF1 DKO B-1a cells (Figs. 2g, 3e). Given that CD5 MFI is critical to identify, gate, and quantify B-1a cells in this study, the decrease in the CD5 MFI on the cell surface of B-1a could have confounded their quantification in this study, potentially invalidating their premise. The lower expression of CD5 in the DKO mice may have “merged” the B-1a (CD5^{low}) with B-1b (CD5^{neg}) in the flow cytometry gating strategy used to quantify B-cell subsets. Indeed, the fact that DKO mice develop archetypal B-1a cells (VH11/VH12/anti-PtC) at levels similar to wild-type suggests that B-1a cells can develop in TCF1/LEF1 DKO mice but have decreased CD5 expression.

Whilst there is decreased CD5, the reviewer will appreciate the B-1a population is still identifiable in the TCF1/LEF1 KO mice, shown in Fig. 1i-k (below). It is possible that some B-1a cells with marginally low CD5 expression may be present in the B-1b gate. Given B-1b cells did not significantly increase in TCF1/LEF1 deficient mice, we think it is reasonable to conclude that loss of B-1a cells accounts for most of the loss of total B-1 cells.

Also, the PtC-binding population, despite expressing lower amount of CD5, can be identified as a completely discrete population as shown in Fig. 1l (below). This PtC+ population is decreased in total numbers, reflecting the reduction in total B-1a cells.

When plotted instead as a proportion of B-1a cells, the PtC-binding B-1a cells remain as a comparable fraction of the total (reduced) B-1a cell pool, which in itself indicates that we are able to reliably gate B-1a cells despite their reduced CD5 MFI:

4. Similarly, TCF1/LEF1 DKO B-1a cells have lower expression of CD19 (Ext. Fig. 2a) and IgM (Fig. 3a), which are known to affect B-1a development. Hence, TCF1/LEF1 transcription factors may affect B-1a development only indirectly through the expression of CD19, IgM, and CD5, as already shown in previous studies.

Prompted by the reviewer, we have now quantified CD19 expression in both B-1 progenitors (B1P) in E18.5 foetal liver and mature B-1a cells from peritoneal cavity. There is no difference of CD19 expression between TCF1^{WT}LEF1^{WT} and TCF1^ΔLEF1^Δ mice (new Extended Fig. 2j, see right).

The reviewer is correct that TCF1-LEF1-deficiency causes a reduction total BCR expression: we now show this formally by measuring Ig kappa and lambda light chain expression (Extended Fig.2j). This adds to the novelty of our manuscript showing that that transcription factors TCF1 and LEF1 – not known before to exert functions in B cells - act upstream to regulate many important molecules for B-1 cell maintenance, self-renewal and function.

5. B-1 cell progenitors (B-1P) were rather increased in TCF1/LEF1 DKO mice (Ext. Figs.), contradicting the main premise of this study. Moreover, the authors speculated about a putative B-1a progenitor characterized as CD5+ without proper citation for this or a functional assay to validate their potential to generate B-1a. These cells expressing CD19+/B220-/CD5+ could be mature B or plasma cells instead of canonical B-1P.

We thank for the reviewer for their suggestions. It is true that defects in factors required for cell differentiation very often manifest with an increase in the progenitor population that cannot differentiate into the next stage. A comparable accumulation of B-1P was seen in Bmi KO mice, in which B-1a cells also were severely decreased (Kobayashi et al., 2021). However, given that CD5 expression is key to identify both bone marrow and splenic B-1a precursors and we have shown reduction of CD5 expression in dKO mice, we agree with the reviewer that we cannot definitively conclude that TCF1 and LEF1 are required for B-1a development. Hence, we have edited the text accordingly throughout to confine our conclusions on the effects of these transcription factors on B-1a cell expansion / self-renewal and regulatory function.

In terms of the putative B-1a progenitor characterized as CD5+, we cite the work of Li, H. et al: "Identification of novel B-1 transitional progenitors by B-1 lymphocyte fate-mapping transgenic mouse model Bhlhe41 dTomato-Cre" ¹⁷.

6. In adoptive transfer chimeras, DKO B-1a cells developed and reconstituted up to 50% of the B-1a compartment in the peritoneal cavity, while it reconstituted up to 100% of splenic B-1a (Fig. 1g). Hence, TCF1/LEF1 DKO shows only mild effect on B-1a development. Previous studies (see above) have shown a more robust effect, including loss of the archetypal B-1a repertoire.

In repeat and more robust chimera experiments with increased mouse numbers and increased numbers of donor cells, we show that mice reconstituted with donors lacking either one or both transcription factors have a ~ 50% reduction in B-1a cells in peritoneal cavity and B-1 cells in spleen (Extended Fig. 1n).

7. TCF1/LEF1 DKO mice show no significant changes in natural antibody production and B-1 plasma cells, challenging the main premise of the study.

To date, it is not known how many B-1a cells are needed to establish the natural antibody repertoire in a mouse. Our study shows that TCF1/LEF1 deficiency cause an up to 70% reduction in peritoneal B-1a cells, which may be sufficient to produce enough IgM to fill that niche. The main premise of our study is to show novel and important functions of TCF1/LEF1 in B-1a cell renewal and maintenance as well as regulatory capacity. In the absence of these transcription factors, B-1a cells are more easily exhausted, fail to give rise to a resting “stem-like” population upon stimulation, express reduced IL-10 and PD-L1, and fail to repress CNS inflammation.

8. Reduction in transitional B-1a was minimal (~30%), followed by a comparable reduction in B-2 cells (~20%), raising questions about the magnitude of the effect on B-1a cells.

This may be an oversight by the reviewer: Transitional B-1a cells were ~15% in WT and ~5% in TCF1/LEF1 KO. This represents a 66% reduction (Extended Fig. 2c).

9. The flow cytometry and gating strategy used to identify B-1a self-renewal in Fig. 2h lacks proper control and thus can be misleading. The authors use a quadrant gate that cuts through the middle of a normally-distributed cluster (Fig. 2h, top left quadrant). This plot needs a negative control that includes BrdU staining in cells from mice that did not receive BrdU treatment.

The negative control from the same experiment (mice not receiving BrdU water) has now been included in new Fig.2g.

10. TCF1/LEF1 deficiency leads to less than 50% reduction in IL-10 production. Therefore, one cannot conclude that these transcription factors promote IL-10 production while basal IL-10 production levels are conserved in splenic DKO B cells.

Again, there seems to be a misunderstanding of our data, and we have tried to make it more clear in this version. We found two subsets of splenic B cells (blue and red gates) each producing IL-10 in resting vs activated states respectively (Fig. 4f):

- i) The red gate identifies a B-1a-like subset - CD19^{hi} CD5⁺ CD21^{lo} CD1d^{lo} - that produces IL-10 after activation. In DKO B cells, these IL-10-producing cells are reduced by 75%.
- ii) The blue gate identifies a likely non-B-1a derived B cell subset - CD19^{lo} CD5⁻ CD21^{hi} CD1d⁺ - that produces basal IL-10, which does not change in DKO mice.

The conclusion is that TCF1/LEF1 deficiency significantly reduces IL-10 production by splenic B-1a-like cells.

11. Human B-1a is not well characterized in humans, and the gating strategy used here is known to include cell doublets and plasma cells (as discussed in earlier publications related to human B-1). Most importantly, no significant difference exists in TCF1/LEF1 expression levels in human memory B cells and the so-called B-1a, opposing or weakening the study's premise.

We have dedicated considerable effort to characterize B-1 cells in humans. For this, we have performed stains of pleural cavity B cells from human patients with bacterial pleural infection. In order to compare like with like, pleural cavity B cells were stained in the same tube with CTV-labelled PBMCs from the same donor. This revealed a clear and expanded population in pleural cavity bearing the markers that identify B-1 cells in mice CD19⁺ CD43⁺ CD5⁺ and lacking the plasmablast marker CD38. These human pleural cavity B-1 cells also express high amounts of TCF1 and LEF1 as shown in mouse peritoneal cavity B-1a cells and have phosphatidyl-choline (Ptc) reactivity. Notably, unlike healthy blood donors that have few B-1 cells circulating in the blood, patients with bacterial pleural infection have a sizeable circulating B-1 population. We also show that B-1 cells are also rare, albeit visible, in human mucosal associated lymphoid tissues like adenoids. Furthermore, we show that the malignant expanded B cells in chronic lymphocytic leukemia (CLL) also share a B-1 cell phenotype, with the majority co-expressing TCF1 and LEF1.

Minor concerns:

1. B-1 cells also make IgG, particularly IgG3 (and not only IgM and IgA as described in line 25) (see Yang Yang et al., PNAS 2012)

Corrected and citation included.

2. The term EAE used in lines 50 and 275 might refer to “Experimental Autoimmune Encephalomyelitis” and not “Experimental Allergic Encephalomyelitis”? If so, there is missing a key reference on the role of B-1a cells in EAE (Kurnellas et al., PNAS 2015)

Both terms are used interchangeably in the literature¹⁸. EAE is not strictly speaking an autoimmune disease given the need to immunize animals in the presence of a CFA-related adjuvant, which is more akin to a delayed type of hypersensitivity (DTH) reaction. In any case, we now refer to Experimental Autoimmune Encephalomyelitis and have now included the additional reference suggested by the reviewer¹⁹

3. The reference cited in lines 55 and 162 is not related to normal B-1a development but to mature B cells. Instead, please cite the reference from Kyoko Hayakawa on Thy-1+ B-1a cells when referring to the role of BCR signaling in normal B-1a development.

Both papers by Hayakawa, with Hardy and Herzenberg are now included^{20,21}

4. Missing references supporting the statement about the difference between progenitors that give rise to B-1 and B2 in line 87 and the putative CD5+ B-1P.

These references are now included^{17,22,23}

5. It is unclear how transitional B-1a cells were gated, and other gating strategies need proper FMO or other negative controls to define gating boundaries.

The transitional B-1a gating strategy is included in Fig. 2d, and other gating strategies are shown in supplementary figures.

6. PDL1-L2 expression on B cells, including B-1, has been previously published and not cited (Rothstein's group and others).

We have added these citations ^{12,14}

7. The rationale for blocking IL-10R signaling was unclear, which is already known to have an autocrine function in B cells, and the effect here was mild.

Whilst it is known that IL-10R signalling limits B-1 cell growth, we sought to clarify whether the observed TCF1/LEF1-mediated break in excessive B-1 cell division to prevent exhaustion depends on IL-10 production, thus the rationale of the IL-10 blockade experiment.

8. Given the recently published role of ZEB2 in driving the formation of age-associated B cells, which were increased in the DKO mice here, it would be important to investigate the link between TCF1/LEF1 and ZEB2.

We thank the reviewer for their suggestion. We checked *Zeb2* expression in TCF1/LEF1 deficient B-1a cells by RNAseq and observed no change of *Zeb2* expression between TCF1^{WT}LEF1^{WT} and TCF1^ΔLEF1^Δ mice (figure below, now shown in Extended Fig. 5k).

9. There is no clear justification for using geometric mean fluorescence (gMFI) instead of median MFI for the well-resolved clusters. Will the statistical significance change?

Geometric mean fluorescence intensity (gMFI) is effectively median MFI, and is less affected by outliers than mean fluorescence intensity (MFI); however, the trends for gMFI or median MFI tend to be the same ²⁴. In any case, we have double checked and the statistical significance remains the

same.

10. Fig. 3d has a typo "f" on top of the heatmap.

Thank you, corrected.

11. The blue label in Fig. 4d might refer to B-2 cells instead of CD19+/B220-?

It did refer to B-1 cells (CD19+/B220-) that were subdivided into those expressing higher vs lower TCF1. To avoid confusion with the previous blue/red colour scheme for B-2 vs B-1 cells, we have changed it as below:

12. The phenotype of B cells in Fig. 4f might represent Marginal Zone B cells due to high expression of CD21 and B220.

Yes, the so called B10 cells share surface markers with marginal zone B cells, plasmablasts etc besides expressing intracellular IL-10 staining²⁵. We are being cautious and only referring to them as CD19^{low} IL-10⁺ cells.

References:

1. Vogel, C. & Marcotte, E. M. Insights into the regulation of protein abundance from proteomic and transcriptomic analyses. *Nat Rev Genet* **13**, 227–32 (2012).
2. Blais, J. D. *et al.* Activating transcription factor 4 is translationally regulated by hypoxic stress. *Mol Cell Biol* **24**, 7469–7482 (2004).
3. Kreslavsky, T. *et al.* Essential role for the transcription factor Bhlhe41 in regulating the development, self-renewal and BCR repertoire of B-1a cells. *Nat Immunol* **18**, 442–455 (2017).
4. Haas, K. M., Poe, J. C., Steeber, D. A. & Tedder, T. F. B-1a and B-1b cells exhibit distinct developmental requirements and have unique functional roles in innate and adaptive immunity to *S. pneumoniae*. *Immunity* **23**, 7–18 (2005).
5. Carsetti, R., Rosado, M. M. & Wardemann, H. Peripheral development of B cells in mouse and man. *Immunol Rev* **197**, 179–191 (2004).
6. Dear, T. N. *et al.* The Hox11 gene is essential for cell survival during spleen development. *Development* **121**, 2909–2915 (1995).
7. Dilillo, D. J. *et al.* Chronic lymphocytic leukemia and regulatory B cells share IL-10 competence and immunosuppressive function. *Leukemia* **27**, 170–182 (2012).
8. Saulep-Easton, D. *et al.* The BAFF receptor TACI controls IL-10 production by regulatory B cells and CLL B cells. *Leukemia* **30**, 163–172 (2015).
9. Darwiche, W., Gubler, B., Marolleau, J. P. & Ghamlouch, H. Chronic lymphocytic leukemia B-cell normal cellular counterpart: Clues from a functional perspective. *Front Immunol* **9**, 345407 (2018).
10. Suo, C. *et al.* Mapping the developing human immune system across organs. *Science* **376**, (2022).
11. Yang, Y. *et al.* CTLA-4 expression by B-1a B cells is essential for immune tolerance. *Nat Commun* **12**, 1–17 (2021).
12. Khan, A. R. *et al.* PD-L1hi B cells are critical regulators of humoral immunity. *Nat Commun* **6**, 1–16 (2015).
13. Zhong, X. *et al.* PD-L2 expression extends beyond dendritic cells/macrophages to B1 cells enriched for VH11/VH12 and phosphatidylcholine binding. *Eur J Immunol* **37**, 2405–2410 (2007).
14. Spurrier, M. A., Jennings-Gee, J. E., Daly, C. A. & Haas, K. M. The PD-1 Regulatory Axis Inhibits T Cell-Independent B Cell Memory Generation and Reactivation. *The Journal of Immunology* **207**, 1978–1989 (2021).
15. Choi, J. K. *et al.* IL-27-producing B-1a cells suppress neuroinflammation and CNS autoimmune diseases. *Proc Natl Acad Sci U S A* **118**, e2109548118 (2021).
16. Kreuk, L. S. M. *et al.* B cell receptor and toll-like receptor signaling coordinate to control distinct B-1 responses to both self and the microbiota. *Elife* **8**, (2019).
17. Li, H. *et al.* Identification of novel B-1 transitional progenitors by B-1 lymphocyte fate-mapping transgenic mouse model Bhlhe41 dTomato-Cre. *Front Immunol* **13**, 946202 (2022).
18. Stromnes, I. M. & Goverman, J. M. Passive induction of experimental allergic encephalomyelitis. *Nat Protoc* **1**, 1952–60 (2006).
19. Kurnellas, M. P. *et al.* Amyloid fibrils activate B-1a lymphocytes to ameliorate inflammatory brain disease. *Proc Natl Acad Sci U S A* **112**, 15016–15023 (2015).

20. Hayakawa, K. *et al.* Early generated B1 B cells with restricted BCRs become chronic lymphocytic leukemia with continued c-Myc and low Bmf expression. *Journal of Experimental Medicine* **213**, 3007–3024 (2016).
21. Hayakawa, K., Hardy, R. R., Stall, A. M., Herzenberg, L. A. & Herzenberg, L. A. Immunoglobulin-bearing B cells reconstitute and maintain the murine Ly-1 B cell lineage. *Eur J Immunol* **16**, 1313–1316 (1986).
22. Montecino-Rodriguez, E. & Dorshkind, K. B-1 B Cell Development in the Fetus and Adult. *Immunity* **36**, 13–21 (2012).
23. Pedersen, G. K. *et al.* B-1a transitional cells are phenotypically distinct and are lacking in mice deficient in I κ BNS. *Proc Natl Acad Sci U S A* **111**, E4119–E4126 (2014).
24. El-Hajjar, L., Ali Ahmad, F. & Nasr, R. A Guide to Flow Cytometry: Components, Basic Principles, Experimental Design, and Cancer Research Applications. *Curr Protoc* **3**, e721 (2023).
25. Kalampokis, I., Yoshizaki, A. & Tedder, T. F. IL-10-producing regulatory B cells (B10 cells) in autoimmune disease. *Arthritis Res Ther* **15**, 1–12 (2013).

Referee #1 (Remarks to the Author):

This manuscript resubmission by Shen et al. includes significant additional data, some reorganization and clarification of statements and additional changes to the revised manuscript. The paper is a tour-de-force of analyses on the role of two transcriptional regulators (TCF1 and Lef1) conducted in both mouse strains and human samples, suggesting their distinct involvement in B-1 cell development and regulation following activation. The authors have addressed with these changes many of my prior concerns. Most significantly for that is the inclusion of human data that seem to suggest B-1 like populations among CD38⁻ CD43⁺ CD5⁺ human B cells. Their additional data expand prior studies including peritoneal cavity efflux of humans with bacterial peritoneal cavity sepsis that are impressive additions to the study, with some limitations. They show a correlation between the transcriptional regulators and these B cell populations in humans. Since expression of these regulators is not restricted to B-1 cells, I remain concerned about the term human B-1, given prior work having shown that this population, identified based on their phenotype, is heterogeneous including many class-switched cells etc. That said, their expression data are convincing and may help to further clarify functional versus developmental relationships between mouse and human B cells. There are new data and interesting results in the study expanding the role of TCF1 as an inducer of stem-ness from CD8 T cells, to also B-1, perhaps one of the most significant conclusions.

We appreciate the reviewer's positive comments about our work and value their concern about the heterogeneity of the human B-1 population (CD19⁺CD38⁻CD43⁺TCF1⁺LEF1⁺ population). The literature supports the notion that B-1 cells are also heterogeneous in mice, and that they can undergo Ig class switch recombination: B-1a cells producing IgA have been documented¹ and isotypes like IgG3 can be produced in response to infection². In the figure below we show that a small fraction of mouse peritoneal cavity CD19⁺ PtC⁺ B-1a cells have switched to IgA and IgG. Nevertheless, following the reviewers' suggestion, we have cautiously described the human CD19⁺ CD38⁻ CD43⁺ TCF1⁺ LEF1⁺ cells as a "B-1-enriched" (rather than just B-1), to account for the potential heterogeneity and inclusion of some non-B-1 cells.

A couple of minor concerns should be addressed:

1) Line 324 refers to data in Extended Fig. 4h that are at best confusing. The text states an 80% reduction in patients with IL2RG deficiency and the bar chart in that fig suggests that is the case. However, the flow plots providing an example (I assume) of these patients showed a 7-fold increase in the subset compared to HD (8% versus 1%)?

Thank you for pointing out the need to clarify this. The flow plots in Extended Fig. 6g had been pre-gated on CD19⁺ CD38⁻ cells whereas the bar chart in the main figure showed the percentage out of total CD19⁺ cells. We have clarified this in the text and legends.

2) The authors should indicate for Figure 3 scRNAseq from which tissue these cells were taken. Its neither stated in the text nor the figure legend.

We now indicate the tissues (foetal liver, adult bone marrow, spleen and peritoneal cavity) in the legend (we had described the tissues used to purify each subset in lines 223-225).

3) Fig. 4n is cited prior to Figs 4k-m.

Thank you. We have now updated the correct order of Fig.4 in both text and figure.

Referee #2 (Remarks to the Author):

The manuscript was greatly improved following revision, and the authors nicely addressed the reviewer's concerns. The authors provided several additional experiments that strengthen the study, including assessing human B-1a cells from serosal cavities and CLL patients, further single-cell analysis of distinct developmental stages of B-1a, and generating a B-1a reporter mouse to track these cells in their EAE model. While this manuscript has many interesting and novel findings on B-1a cell development and function, a few issues still require further clarification.

1. Figure 1B shows a clear TCF1+ B-1a and a LEF1+ B-1a population by flow cytometry. Based on the distinct populations of either Tcf7 or Lef1 in Figure 1A and most of the mouse data is generated using a double knockout, why do the authors not look for the frequency of TCF1+ and LEF1+ double-positive B-1a cells by flow. Their trajectory data suggests that these transcription factors may have a role at different stages of B-1a development and function; determining if B-1a are all double positive or can be broken down into different populations would help understand these cells.

2. IL-10 was important in B-1a function to prevent excessive proliferation and promote the stemness of B1 cells via TCF1/LEF1. However, there appears to be a discrepancy between mice and humans. The TCF1/LEF1 populations in human B-1a cells express minimal IL-10 (Figure 3h). Can the authors elaborate on whether this suggests different mechanisms between the mouse and human functions of B-1a cells.

Thank you for asking us to elaborate on this, now included in the text. The tissues used in Figure 3h are of foetal (i.e. prenatal) origin, thus sterile and prior to interaction of B cells with the microbiome. It has been reported that human newborn cord blood B cells (thought to be enriched in B-1 cells) can produce IL-10 in response to TLR9 ligands, which are abundant in microbial DNA⁴. Thus, we expect post-natal human B-1 cells to be IL-10 producers, particularly upon encountering TLR ligands. Unfortunately, the low numbers of B-1 cells obtained from the highly purulent pleural fluid lavage precluded investigating IL-10 production.

3. While the concluding paragraph expands on some interesting ideas of exploring the effects of B-1a cell regulation in different diseases and their dysregulation into an exhausted phenotype, it does not summarize the global context of why these findings are important.

We thank the reviewer for this suggestion. We have summarized the global context of our findings in the last paragraph, which now reads as follows:

“Our findings suggest that TCF1 and LEF1 are important for generation of IL-10–producing regulatory B-1 cells in the periphery. Since the major wave of B-1 cell expansion occurs postnatally upon microbiome exposure⁵⁸ and bacterial CpG is a potent inducer of IL-10 secretion⁵⁹, B-1a regulatory function is likely key to help prevent inappropriate immune responses to commensal bacteria and viruses. Gut IgA plasma cells have been shown to arise from the same hematopoietic progenitors as B-1a cells during ontogeny and produce IgA clones in response to neonatal gut viral infection⁶⁰. Whether early life B-1a education and their anti-microbial IgA production determines the outcome of protective vs pathogenic/autoimmune reactions to microbial exposures later in life remains to be determined. Our identification of expanded B-1 cells in the blood of some patients with pleural infection based on TCF1, LEF1 and CD43 expression suggests that these cells may also serve as a biomarker for early detection of serosal infections (e.g. pleuritis, peritonitis) and possibly sepsis. Finally, our data suggests that chronic activation of B-1 cells - known to be selected for self-reactivities early in life³² - or loss of IL-10 or PD-L1-mediated regulatory function as observed in the absence of TCF1/LEF1 may contribute to autoimmunity. This work paves the way for studies to elucidate the protective vs pathogenic roles of human B-1 cells in autoimmune diseases, immunodeficiency, and cancer, and their therapeutic manipulation.”

4. B-1a cells regulating disease through IL10 and PD-L1 are interesting. Is it known what the targets of these mediators are? In addition, in the EAE model (Figure 4n and o), the authors asked if B-1a cells exerted their regulatory effects locally or distally (line 386). However, their data demonstrate that B-1a cells were present in the brain, which also doesn't rule out a distal influence, perhaps in SLOs. This may not be within the scope of this paper, but some insights into their regulation mechanisms in affected organs vs SLOs would place these findings into a broader, more meaningful context.

We appreciate the reviewer's insightful comments. In mice, IL-10 produced by B-1a cells has been reported to inhibit proinflammatory responses⁵, limit the phagocytic ability of macrophages⁶ and promote regulatory T cell responses⁵. PD-L1^{hi} B cells have been shown to restrict T-cell differentiation⁷. This has now been mentioned in the text.

We agree with the reviewer's point about B-1a cells also possibly exerting a distal influence in SLOs. We have added this as follows (Line 316-318):

“Whilst this suggests that B-1a cells can enter the brain and may thus exert local regulatory effects, we cannot exclude additional distal effects in secondary lymphoid organs.”

Referee #3 (Remarks to the Author):

This is a resubmission from Qian Shen, Carola Vinuesa's team, and collaborators of the manuscript “The transcription factors TCF1 and LEF1 promote B-1a cell homeostasis and regulatory function”.

I appreciate the additional experiments the authors performed to revise their important work.

I agree that knocking out the two genes simultaneously moderately impacts B-1a cells. However, the new experiments and their interpretations did not address the major concerns raised in round one by the reviewers. Indeed, the authors also acknowledge that their results cannot be interpreted to conclude that TCF1/LEF1 are required for B-1a development.

Overall, the effects of TCF7-knockout (KO), LEF1-KO, or the double-KO (DKO) on B-1a cells were only partial and moderate compared to the more significant effects of other previously published transcription factors/genes.

The reviewer brings up their main concern repeatedly: there have been reports of other transcription factors and molecules such as Arid3a and Lin28b that the reviewer believes exert stronger influence on B-1a cells than TCF1/LEF1, and this diminishes the reviewer's interest in our findings. We have carefully searched the literature, and conclude that the reviewer's statements of the "modest", "moderate", "minimal" effects of TCF1/LEF1 compared with the other molecules are not backed up by the published data:

In our TCF1/LEF1 KO mice we report:

- a **45%** reduction in total peritoneal B-1 cells (Fig. 1k),
- a **71%** reduction in peritoneal B-1a cells (Fig. 1k),
- a **67%** reduction in splenic B-1 cells (Fig. 1m). The comparisons are:

Arid3a deficiency:

- In the paper by Hayakawa and Hardy using total (not B cell conditional) Arid 3a KO⁸, only a single mouse per genotype was shown. There was no quantification, no gatings shown, no percentages in gates, and no outliers shown in the contour plots; effect comparison is thus impossible.
- The paper by Stephen Malin⁹, showed B cell conditional Arid3a KO. Although values for individual mice were not shown, there was:
 - a **~22%** reduction of total PC B-1 cells,
 - an **~81%** reduction in PC-B1a cells,
 - and **0%** reduction (i.e. no reduction) in splenic B-1 cells.

LIN28b deficiency: There is only one paper showing data for total LIN28b KO mice, which only showed data for day 10 neonatal mice¹⁰. The senior author of that paper, Joan Yuan, has communicated to us that there was no reduction in B-1 cells in adult KO mice (both in spleen and peritoneal cavity). This was not published.

Nfatc1 deficiency: The published mice¹¹ were bred on a BALB/cByJ background and were complete KO. Furthermore, experiments were only performed in foetal liver chimeric mice, so the results are not directly comparable with intact mice. In these experiments, there was a **~ 60%** reduction in B-1a cells in the peritoneal cavity and **~ 80%** reduction in the spleen, although individual mice were not shown.

Bhlhe41 deficiency: The mouse model used was a complete KO (not a conditional B cell KO)¹². There was good quantification in this paper, which showed a ~73% reduction in total B-1a cells in the peritoneal cavity and ~79% in the spleen.

In summary, the published data (see table below) do not support that the effect of TCF1/LEF1 are minimal or modest compared with the effect of the preferred factors of this reviewer - Arid3a and LIN28b.

	Total B-1 Reduction in PC	B-1a cell reduction in PC	B-1a cell reduction in spleen	Reference

Arid3a B cell KO	22%	82%	0%	9
LIN28b total KO	0%	0%	-	10 and personal communication
Bhlhe41 total KO	51%	73%	79%	12
Nfatc1KO fetal liver chimeras	60%	60%	80%	11
TCF1/LEF1 B cell DKO	47% 	71% 	67% 	Figs. 1j-m in this ms.

We would also like to point out the novel findings of our manuscript:

1. We describe a novel role of TCF1 and LEF1 in B cells – to date, these transcription factors have only been shown to exert roles in T cells.
2. We show TCF1 and LEF1 regulate B-1a cell stemness /self-renewal, and this is achieved through several mechanisms beyond regulation of Myc metabolism, not described in memory T cells, including upregulation of Ets1 and production of IL-10 that allows the emergence of a Myc^{lo} population after activation.
3. We identify human B-1 cells with confidence – which have been controversial to date, by i) accessing pleural lavage from patients with bacterial pleural infection and ii) combining phenotypic markers described in mice with TCF1/LEF1 and with BCR reactivity to phosphatidyl choline (PtC) using PtC-liposomes. We show this gating strategy identifies a B-1 enriched population in human blood which is expanded in some patients with bacterial pleural infection and includes the neoplastic population in CLL patients. These results will open the field for investigation of the function of B-1 cells in humans.
4. We show that TCF1/LEF1 promote B-1 cell regulatory function, dampening EAE.
5. By generating a novel B-1a reporter mouse, we identify for the first time B-1a cell in the brain parenchyma in the context of CNS inflammation, which suggest that these cells may exert important regulatory roles at sites of inflammation. These mice will be useful for the scientific community, to track B-1 cells in infection, inflammation and cancer.

For example:

1. Transfers of DKO progenitors led to the reconstitution of roughly 50% of the normal/wild-type (WT) levels of B-1a in the PerC. In the spleen, the effect was minimal, if any. In fact, one outlier animal contributed to the statistical significance, and most recipients seem to have reconstituted most of their normal levels of splenic B-1a (Fig. 1).

These statements regarding our foetal liver and bone marrow chimera experiments are incorrect and unfairly dismissive:

- Peritoneal cavity B-1 cells were reduced by **66%** in foetal liver chimeras (Fig. 1n) and **60%** in bone marrow chimeras (Extended Fig. 2e).
- In the spleen, the reductions were **60%** in foetal liver chimeras (below left) and **52%** in bone marrow chimeras (below right) (Extended Fig 2e shown below). These experiments were repeated 3 times with similar results.

We compare medians, and use Mann-Whitney U test for statistics, to precisely avoid type I errors (false positive caused by single outlier). In U-tests, only the rank is considered, not the actual value. So the p-value of this experiment (above, right panel), would be identical if the wild type mouse that appears to have 4.6% of B-1a cells in the chimeric spleen had 2.5% instead.

2. TCF7-KO did not affect B-1a percentages in the PerC (Fig. 1j). And in the DKO mice, the effect was only moderate. There was a decrease from ~30% of B-1a in wild-type (WT) to ~10% of B-1a in DKO. However, the data showed a large spread in B-1a percentages, with some DKO animals showing a percent of B-1a cells similar to their wild-type (WT) controls (Fig. 1). For example, in Fig. 1i, the two bottom right panels show 45% (18%+26.6%) of B-1 cells in the LEF1-KO versus 35% (19.3%+16.4%) of B-1 cells in the DKO. These ranges of B-1 cells are similar to the WT shown in the first top left panel of Fig. 1i (46.6% of B-1).

Peritoneal:

Splenic:

The precise median decreases in the DKO are:

- **71%** (top left) and **66%** (bottom left) reduction in peritoneal B-1a cells
- **50%** (top right), and **67%** (bottom right) reduction in splenic B1 cells.

“This is only moderate” by comparison to what? As shown in the table (previous page), these values are in the range of those shown for other B-1 factors.

The spread per group is perfectly normal for mouse studies, and we are comparing medians to avoid a type I error.

3. The DKO mice develop B-1a cells with the same canonical (stereotypic) B-1a BCR repertoire as the WT mice. The decrease in the B-1a repertoire showed by the authors reflects only a global reduction in B-1a numbers, but there was no direct effect on the development of the canonical B-1a BCR repertoire, including the B-1a-specific anti-PtC repertoire.

We are unsure what the concern is here. We describe a significant reduction in total PtC-binding B cells. It is unclear why the reviewer thinks that we should see a bias in the reduction of some B-1a clones over others. While there are 3 clones that are more abundant amongst B-1a cells, there is a multitude of smaller clones also present that are likely to require the same cues for development. We simply show that B-1a clones are reduced proportionally when B-1a cells have an impaired self-renewal ability.

4. Most of the moderate reduction in B-1a percentages in DKO mice can be directly attributed to the decrease of IgM and CD5 on their surface, as previously described. For example, it is already established that changes in the BCR/IgM signaling have a stronger effect on B-1a cells than B-2 cells. These effects (i.e., decrease in B-1a numbers) can be seen in CD19-KO, Btk-KO, Nur77, and other molecules downstream BCR signaling. Although it did not achieve a significant p-value, the IgM expression on B-2 cells from DKO mice also decreased, suggesting a role for TCF1/LEF1 in BCR/IgM rather than on B-1a. Since B-1 is more sensitive to BCR changes compared to B-2, the B-1 cells were affected

more significantly in the DKO mice. Showing the role of TCF1/LEF1 on Ig/BCR (and CD5) homeostasis might shed light on their mechanisms.

Our manuscript adds to the exiting literature of the impact of BCR signalling and CD5 expression on B-1a cell maintenance. Our novel findings are:

1. TCF1 and LEF1 play a role in B-1 cell turnover and maintenance.
2. A site in the CD5 promoter is a direct target of TCF1 in T cells, and the DNA binding sites in T cells coincide with open chromatin sites in B-1a cells, suggesting TCF1 also directly controls CD5 expression in B-1a cells.
3. Besides the known role of CD5 and BCR signalling in promoting B-1 numbers, we show a role for Ets1 downstream of TCF1/LEF1 in B-1 cell maintenance as mice age, and critical role of IL-10 in allowing the emergence of stem-like B-1a cells: IL-10 curtails excessive B-1a proliferation driven by high Myc levels, by reducing Myc levels in a subset of activated but not terminally differentiated B cells, after the 2nd division.

5. Given that other genes and transcription factors (TFs), including ARID3a, Lin28b, Bhlhe41, NFATc1, have been shown to regulate B-1a development and homeostasis more robustly than TCF1/LEF1, it would be reasonable to expect a similar or at least more substantial effect of TCF1/LEF1 DKO on B-1a. Thus, it seems unlikely that TCF1/LEF1 represent key TFs or master regulators of B-1a cells in mice and humans. Such a statement in a major publication may add more confusion to the field.

- As explained above, the comments about other factors exerting more robust effects are not backed up by the literature. We have referred to the afore mentioned discoveries in our manuscript, referenced them, and used them as benchmarks to validate our scRNA-seq (Fig. 3f).
- Our data does suggest that TCF1 and LEF1 are key transcription factors (nowhere do we use the term “master regulator”) given the functional consequences of their absence (significant increase in CNS inflammation). We also show that in the absence of TCF1/LEF, B-1a cells not only proliferate excessively, but also acquire an exhausted phenotype, which may be an indicator of pathogenicity as opposed to regulatory potential.

6. The authors show that LEF1 (an enhancer) and TCF1 (a transcription factor) affect different signaling pathways and are expressed at different B-cell development and maturation stages. Hence, there is no logical reason for focusing the study on the DKO instead of each gene individually, which did not significantly affect B-1a numbers.

The reasons to focus on the study of the DKO are:

- Whilst the level of TCF1 and LEF1 expression change during development, both remain expressed in mature B-1 cells, as shown by the scRNA seq analysis (Fig. 3f) and protein staining (Fig.1c).
- TCF1 and LEF1 are known to directly interact and form a complex (see String and ¹³):

- TCF1 and LEF1 have been shown to co-regulate several targets in CD4+ and CD8+ T cells with some overlapping functions albeit not entirely redundant functions^{14,15}.

- We have shown shared (e.g. IL-2-STAT5 signalling) and unique pathways targeted by these transcription factors in B-1a cells. For example, TCF1 affects E2F targets and the G2M checkpoint while LEF1 affects TNF α / TGF β signalling pathways.

7. The effect of DKO on IL-10 occurred only for the “induced” IL-10 and did not affect the constitutive production of IL-10 by B-1a-like cells, suggesting a lack of a direct causal effect. Indeed, it is known that CD5 expression promotes IL-10 production, and a decrease of CD5 on B-1a is expected to decrease IL-10 as previously shown by Gary-Gouy et al. in 2002 (Human CD5 promotes B-cell survival through stimulation of autocrine IL-10 production. Blood, 2002). Thus, the reduced induced IL-10 observed here might be due to decreased CD5 expression on B-1a cells.

The lack of effect described by the reviewer prior to activation (the constitutive production) was only seen in splenic B cells, which contain several IL-10 producing B-2 subsets (e.g. transitional B cells, plasma cells, marginal zone B cells), thus not necessarily B-1 cells. In fact, these splenic B cells producing constitutive IL-10 were CD21 $^{+}$ CD5 $^{-}$ and B220 hi , as opposed to B-1 cells that are CD21 lo CD5 $^{+}$ and B220 lo . Also, we had explicitly mentioned that it is known that CD5 expression promotes IL-10 production in lines 369 and 372 of the previous version, now lines 290-291.

8. Finally, if not later confirmed, classifying human adult B-1 cells solely based on their expression of CD5 $^{+}$ /CD43 $^{+}$ can be harmful to the field. Mouse B-1 cells are defined based on their separate developmental origins and functions, confirmed by lineage/genetic tracing and transfer studies. There is no current way to assess the origin of the CD5 $^{+}$ cells in the infected patients they describe here. In fact, many groups have already tried to characterize human B-1 based on CD5 and failed. They failed because human adult B cells (including adult bone marrow B cells) can express CD5 during development and activation, making identifying human B-1 cells challenging.

We do not classify human B-1 cells “solely based on CD5 $^{+}$ /CD43 $^{+}$ cells”. We identify human B-1 cells based on their anatomical location: B cells in pleural cavity, lack of CD38 expression, and co-expression of CD43, LEF1 and TCF1. Levels of CD5 tend to be high but are indeed more variable so we have been careful with the wording, explicitly stating that it is the combination of CD43, LEF1 and TCF1 that best identifies human B-1-enriched cells. It has been difficult to find these cells in the past because, as we show, in the absence of relevant infections within serosal sites, B-1 cells are very low in the circulation. We also show that PtC reactivity is present in the B-1-enriched, but not in the B-2 cell gate regardless of the presence of pleural infection.

Other major concerns include:

- The overall quality of the flow plots needs improvement. The scales (and ticks) on the X/Y axis are sometimes missing or show errors (log vs bioexponential), and the gates are arbitrary (particularly for mouse LEF1 and also for human samples), missing FMO gates,

percentage values, etc. (see below).

We carefully checked each plot. Regarding the scales of flow plots in previous manuscript, a linear scale was only used for FSC/SSC axes (e.g. x-axis of Fig. 2c) and the y-axis of all histogram plots (e.g. Fig. 2d). Log scales were applied on CD5/TCF1/LEF1 axis of pleural infection plots in Fig. 1e (top two rows) and all the rest flow plots used a biexponential scale. Both are perfectly acceptable; depending on the voltages at the time of acquisition and brightness of the fluorochromes, one may display a clearer population. We have now added FMO gates in LEF1 and TCF1 stains (Fig.1d). Percentage values were only omitted from flow plots that depicted a gating strategy, but these have now been added (Fig. 1c and 1d).

- The expression of TCF1 (flow) and TCF7 (scRNA-seq) in B-1 is negligible and cannot be quantified compared to B-2 cells. The only protein data for TCF1/LEF1 is based on a flow plot likely containing artifacts of data generation and analysis: for example, the “positive” gates are based on CD23⁻ cells (i.e., B-1 cells), which intentionally exclude all B-2 cells and include only the tip of a rather normally distributed population that spread on the Y axis. This spreading of the population is common in flow cytometry due to spillover spreading and cannot support the claim that the tip-top of the spread contains positive cells and the bottom part includes negative cells—they may all represent the same population/phenotype, but appear spread due to spillover spreading and autofluorescence. For example, if the authors move the gate down along the population spread, they might find that those cells at the middle and bottom of the spread contain the same B-1a cells (same phenotype) as the top part. There might be no differences between the top and bottom of the spread. Hence, one cannot conclude from this flow plot that B-1a express high LEF1 and TCF1 proteins. They will need to titrate the reagent, analyze the titrated data, then perform FMO on the optimal titration to confirm the expression of these proteins on B-1a, and perhaps perform another orthogonal assay for protein detection to confirm.

We respectfully disagree. Our plots clearly show that high expression of TCF1 and LEF1 proteins is a feature of the CD23⁻ population, which are B-1 cells (CD23 is a marker of B-1 cells and is absent in B-2 cells). We routinely titrate all antibodies and routinely use FMOs (now plotted in Fig 1d). We are confident that the stain is not an “artifact due to spill over”. Transcription factors are lowly expressed and notoriously difficult to stain but we have extensive and published experience in staining lowly expressed transcription factors in T and B cells (e.g. Bcl6, Tbet, RorgT, Blimp1, etc) and can detect artifacts when they occur. We have nevertheless added additional gating strategies to make it clearer as shown here.

- The same problem with the spleen flow plot. There are no cells inside the LEF1^{hi} gate and the only population that shows a “bump” indicating a positive expression are CD19^{low} cells, which are not B-1a. Indeed, the arbitrary gates show that most of the cells (in red) represent the B220^{hi} (i.e., B-2/MZB) population, not only B-1a.

Expression in the spleen is lower than in the peritoneal cavity and combining the TCF1 or LEF1 stains with low B220 or high CD5 and CD43 helps identification. We have plotted the same cells using additional gating strategies, which make it clearer that LEF1^{hi} cells correspond to the B220^{lo} splenic B-1a cells (right). We do not see why it is a problem that some B-2 cells also express LEF1. This is also the case for all the transcription factors this reviewer mentions to be crucial for B-1a cells, see example of Arid3a (left) and LIN28b (right):

[REDACTION]

The authors claim that DKO increases age-associated B cells (ABC), which was recently shown to require ZEB2 and express CD11c. However, the authors did not see a difference in ZEB2 and CD11c in DKO mice and no explanation was provided. Thus, perhaps we should avoid calling them ABCs before further characterization.

We showed that Zeb2 and CD11c remained unchanged in the absence of TCF1 and LEF1 and we suggested that TCF1 and to a lesser extent LEF1 simply prevent exhaustion of proliferating cells. The word ABC only featured in the discussion and has now been removed.

- In Fig. 1a UMAP (and extended 1b), LEF1 appears to be expressed on B-2, not B-1

There is LEF1 expression in some B-1 and some B-2 cells. We discussed why this might be in previous versions of this manuscript – lines 95-97 in current version. This is not dissimilar for the B-1 transcription factors Arid3a and Lin28b according to Immgen (shown above). For example, germinal center (GC) and transitional 1 (T1) (both B-2 cells) express more *Arid3a* mRNA than peritoneal B-1a and B-1b and splenic B-1a cells, and Lin28b in peritoneal cavity B cells is completely comparable to the expression in every B-2 subset. Also, it is widely known that mRNA and protein correlations are not perfect due to many factors, and that scRNAseq often fails to detect lowly expressed transcription factors.

- ARID3a and LIN28b are the main transcription factors known to regulate B-1a cells, and this was confirmed by the authors' scRNA-seq data. However, they did not describe how LEF1/TCF1 are mechanistically related to these B-1a master regulators (and others such as Bhlhe41, NFATc1) since the DKO didn't show the same strong effect on B-1a as these previously published genes/TFs.

First, we have described above this reviewer's misconception about the stronger effects of the factors they describe as the "master regulators", Arid3a and LIN28b. We have shown that the RNA of these TFs does not change in TCF1/LEF1 dKO.

- The entire data interpretation is based on the "percentage" of B cells in DKO vs WT, rather than their absolute numbers. However, the percentage of B-1a cells will change depending on what other types of B cells are included in the CD19+ gate for each organ in the WT versus DKO. For example, if the number of CD19+ B-cell progenitors increases, the B-1a percentage would decrease even if the actual absolute numbers of B-1a remain somewhat stable. All data should have been reported in absolute numbers by counting the total number of cells in the PerC, spleen, etc.

This appears to be an oversight by the reviewer. We showed absolute numbers in each subset in Extended Fig.2a (see right), and the magnitude of the effects are completely comparable to the data showing percentages.

- The authors rely on CD5 expression to identify B-1 progenitors in the spleen. However, it is known that splenic B-1 progenitors can be identified before the CD5 expression (Ghosh et al. Distinct progenitors for B-1 and B-2 cells are present in adult mouse spleen. PNAS, 2011).

We do not investigate in any figure B-1 progenitors (B-1P) in the spleen. The reviewer might have overlooked that we were analysing transitional B-1a (TrB-1a) cells in the spleen that do require CD5 expression for identification. In Line140-143, we clearly stated we investigated TrB-1a which is known to exclusively generate B-1a cells¹⁷ and they're defined as CD93⁺IgM⁺CD19⁺B220^{lo}CD5⁺ cells.

- In this revision, the authors now suggest DKO might play a role in self-renewal rather than early differentiation of B-1 cells. However, they did not provide experimental evidence of how TCF1/LEF1 relates to Bmi1 gene, which has already been shown to control B-1a self-renewal.

- We showed the self-renewal data right from the first version of the ms. We have just been cautious to draw strong conclusions on the role these TFs on early differentiation due to the possible confounding effect on the gateings of the slight CD5 downregulation. Given the very early differences we see in total B-1 cells (Fig.2c, 4 weeks of age), it is very likely there is also a defect in early differentiation.
- TCF1 and LEF1 have been reported to play a role in self-renewal in T cells¹⁸, embryonic stem cells¹⁹ and hematopoietic stem cells²⁰, in a manner not described to be dependent on Bmi1 (and expression of Bmi1 remains comparable between TCF1/LEF1 WT and dKO).

- The BrdU data was not conclusive. For example, in Fig. 2g there are two nicely distinct populations of BrdU expression for the Ly5a+ cells. However, for the Ly5a-negative cells there is only one normally-distributed population that fluctuates around the gate drawn for the Ly5a+ pop. That population may represent autofluorescence in the WT but not in the DKO cells, which we know show less IgM/CD5 and might be less autofluorescent. Thus, based on these flow plots and gates, the authors cannot make definitive conclusions for the Ly5a-neg population without FMO and other controls for Ly5a-neg cells.

The concern that the Ly5b (Ly5a-) population that is BrdU positive represents “autofluorescence” because it is “normally distributed” is unreasonable. The reviewer might find easier to visualise it if we show dotplots instead of contours (Fig on the right):

- It is misleading to say that the B cells’ BCR repertoire diversity increased in the PerC of DKO mice. As the authors confirmed during their revisions, the BCR diversity is likely due to increased B-2 cells in the PerC of DKO mice. Hence, TCF1/LEF1 may not affect B-1a repertoire. Indeed, the percentage of B-1a expressing the stereotypic anti-PtC repertoire hasn’t changed between WT and DKO. Thus, sequencing the repertoire of total peritoneal B cells (instead of the B-1a vs B-2) can confuse the reader.

It was not a misleading statement because it is a fact that the BCR repertoire is more diverse (i.e. there is a higher total and relative number of unique clones) in the peritoneal cavity of DKO mice precisely due to the loss of clonal B-1a cells and overrepresentation of B-2 cells, as we had stated in our ms.

All we can do is add “whilst” at the beginning of the sentence and add “only” to qualify it further:

“Whilst the BCR repertoire within the total peritoneal CD19+ B cell population of TCF1/LEF1 deficient mice appeared more diverse (Extended Fig. 5a top panel), this was probably only due to the overall decrease in the more clonal B-1a cells, and overrepresentation of sequenced B-1b and B-2 cells”.

• Fig. 1 (j and l) show a significant variability between mice and some values are inconsistent, suggesting they might have been incorrectly plotted? For example, the WT mice in the TCF1-KO group (yellow) show a large spread in the percentages of B-1, but not B-2. Since the data is reported in percentage, a large decrease in B-1 should have resulted in a large increase in B-2, but that was not the data plotted in Fig. 1j-l. In addition, one data-point might have been dropped (one mouse) from the B-2 group, which contains only 4 mice, compared to 5 mice for all the other corresponding B-1 groups in the same Fig. 1.

- The variability observed is normal in mouse studies and within normal ranges.
- The reviewer has overlooked that we are not showing B-1 cells as % of CD19+ cells in the first (left) panel; instead, the figure shows data as “% of live cells” (so that we can show where our threshold for B220 low cells is, compared with non-B cells). There are of course differences when plotting B-1a cells as a proportion of total CD19+ cells because the latter considers the percentage of CD19+ cells in each sample, which is not shown here. We have made sure all data points are visible:

• Based on scRNA-seq data and flow, LEF1 and TCF7+ cells rarely overlap (they appear almost mutually exclusive), but in human fetal tissues they appear as double-positive. In addition, IL7R are known to be expressed on progenitors, which according to Fig.3f (scrRNA-seq), should express LEF1, not TCF7. However, the human fetal samples show IL7R expressed only on double-positive cells, suggesting a different cell type?

As highlighted in the purple boxes, there is still expression of LEF1, TCF7, CD5 and CD43 (SPN) in mature splenic (SP) and peritoneal cavity (PC) B cells, albeit for Lef1 this is less than in the progenitor stages. However, B-1P do express larger amounts of both Lef1 and IL7R. Fig.3h data shows that amongst human fetal cells that express CD5 and SPN there is a prominent LEF1 and TCF1 double positive population.

• In Fig. 1e, the authors show >60% of B cells in infected individuals' blood and pleural cavity are B-1 cells. This is highly unlikely. Classifying human B-1 cells solely based on CD5 and CD43 may be inaccurate as the field has already addressed this many years ago. CD5 was not a reliable marker for human B-1 because developing B cells in adults, including adult bone marrow B cells, can express CD5, CD38, CD43, etc., due to early differentiation or activation. Thus, I would be cautious claiming that 60% of human blood B cells are B-1 during infection solely based on CD5/CD43 expression.

We explained above that the field has found it difficult to identify human B-1 cells because they are not present at significant numbers in the blood or secondary lymphoid tissues of healthy individuals. However, they are very prominent in the pleural cavity and in the blood of some – not all – individuals with active pleural infection. We do not propose to use CD5 as the main marker of B-1 cells in the periphery. In fact, our data suggests that the combination of CD43 with LEF1 or TCF1 is a better combination and adding PtC binding increases confidence. Any concern that we are gating activated or differentiated B cells is significantly mitigated by gating on CD38- B cells, which excludes most activated B cells, transitional B cells, plasmablasts and plasma cells²¹. To be cautious, we are calling this population in the periphery “B-1 enriched”.

• The human blood data in panels e and g (Fig 1) are inconsistent. In panel e, they show >60% of CD5+/CD43+, but in panel g there is virtually no cells expressing CD5 (and the percentage value is missing).

Rather than inconsistent, this is the typical variability seen in humans, particularly in the context of infection. More work is needed to understand which bacteria induce expansion of B-1 cells in humans. As shown in the literature⁴, human newborn regulatory B cells (nBreg) from cord blood (CB) respond differently with pathogens associated with neonatal sepsis, *E. coli* and *S. agalactiae*. Only *S. agalactiae* triggers nBreg IL-10 production while *E. coli* induces IgM secretion. The kinetics and timeframe of expansion and contraction also need to be investigated.

• It was unclear to me why they analyzed IL-2RG-deficient patients to quantify B-1 cells without describing whether the mechanism of TCF1/LEF1 in B-1a is through IL-2. In addition, the flow gates drawn to define the CD5+ are arbitrary and inconclusive, likely containing contamination from the main cluster. In extended Fig. 4h, since the number of cells collected

for the patients was much lower than for healthy controls, the few dots inside the CD5 gate can misrepresent a high percentage, skewing the result and interpretation. There is no clear CD5+/CD43+ population, and thus the percentage reported may be meaningless.

- We explained the rationale for this analysis in lines 249-254:
“Given that IL-2 signalling is a TCF1/LEF1-dependent hallmark pathway in mouse B-1a cells (Fig. 3g) and that LEF1/TCF1 co-expressing human B-1 cells express high IL-2RG and IL7RA (Fig. 3h), we asked whether these receptors influence B-1 cell development in humans. We evaluated the B-1-enriched cells in the circulation of paediatric SCID patients with genetic loss-of-function (LOF) variants in IL2RG and IL7RA, and sufficient events in the CD19+ CD38- gate.”
- As expected, total B cells were significantly reduced in SCID patients, and they were significantly activated B cells (CD38+). Thus, despite some intriguing changes, we were very cautious not to draw firm conclusions and instead stated: “further work is needed to reach definitive conclusion” (Line257) in our manuscript.

Minor comments:

- Fig. 1c, the bottom right panel, has the axis labels reverted, I believe?
We updated the plot.

- The ref. 11 in line 216 may be incorrect. I believe they meant to describe the Thy-1-KO study from Hayakawa/Hardy et al.
We have updated this ref.

- The ref. 78 in line 378 may be incorrect. I believe they meant to describe the role of B-1a in EAE by Kurnellas et al., PNAS.

We did mean ref. 78 (Miles et al) and we have now also cited Kurnellas' paper.

Finally, I acknowledge and appreciate the important work of the authors, and thus think their findings, which show a moderate effect on B-1a cells, could be more accurately described as “TCF1/LEF1 DKO mice show decreased percentage of B-1a cells, impaired formation of a Myc-lo population, and a decrease in stimulation-induced IL-10 and PDL1 in activated B-1a cells. This phenotype is likely due, in part, to the decrease of CD5 and IgM expression on B-1a cells.” This conclusion seems reasonable based on the data presented here.

References:

1. Kroese, F. G. *et al.* Many of the IgA producing plasma cells in murine gut are derived from self-replenishing precursors in the peritoneal cavity. *Int Immunol* **1**, 75–84 (1989).
2. Baumgarth, N. B-1 Cell Heterogeneity and the Regulation of Natural and Antigen-Induced IgM Production. *Front Immunol* **7**, 324 (2016).
3. Jia, Y. *et al.* Thymine DNA glycosylase promotes transactivation of β -catenin/TCFs by cooperating with CBP. *J Mol Cell Biol* **6**, 231–9 (2014).
4. Gu, Q. *et al.* Intestinal newborn regulatory B cell antibodies modulate microbiota communities. *Cell Host Microbe* **32**, 1787-1804.e9 (2024).
5. Miles, K. *et al.* Immune Tolerance to Apoptotic Self Is Mediated Primarily by Regulatory B1a Cells. *Front Immunol* **8**, 1952 (2017).
6. Popi, A. F., Lopes, J. D. & Mariano, M. Interleukin-10 secreted by B-1 cells modulates the phagocytic activity of murine macrophages in vitro. *Immunology* **113**, 348–54 (2004).
7. Khan, A. R. *et al.* PD-L1hi B cells are critical regulators of humoral immunity. *Nat Commun* **6**, 5997 (2015).
8. Hayakawa, K. *et al.* Crucial Role of Increased Arid3a at the Pre-B and Immature B Cell Stages for B1a Cell Generation. *Front Immunol* **10**, 457 (2019).
9. Habir, K., Aeinehband, S., Wermeling, F. & Malin, S. A Role for the Transcription Factor Arid3a in Mouse B2 Lymphocyte Expansion and Peritoneal B1a Generation. *Front Immunol* **8**, 1387 (2017).
10. Vanhee, S. *et al.* Lin28b controls a neonatal to adult switch in B cell positive selection. *Sci Immunol* **4**, (2019).
11. Berland, R. & Wortis, H. H. Normal B-1a cell development requires B cell-intrinsic NFATc1 activity. *Proc Natl Acad Sci U S A* **100**, 13459–64 (2003).
12. Kreslavsky, T. *et al.* Essential role for the transcription factor Bhlhe41 in regulating the development, self-renewal and BCR repertoire of B-1a cells. *Nat Immunol* **18**, 442–455 (2017).
13. Jia, Y. *et al.* Thymine DNA glycosylase promotes transactivation of β -catenin/TCFs by cooperating with CBP. *J Mol Cell Biol* **6**, 231–9 (2014).
14. Yu, S. *et al.* The TCF-1 and LEF-1 transcription factors have cooperative and opposing roles in T cell development and malignancy. *Immunity* **37**, 813–26 (2012).
15. Xing, S. *et al.* Tcf1 and Lef1 transcription factors establish CD8(+) T cell identity through intrinsic HDAC activity. *Nat Immunol* **17**, 695–703 (2016).
16. Suo, C. *et al.* Mapping the developing human immune system across organs. *Science* **376**, eabo0510 (2022).
17. Pedersen, G. K. *et al.* B-1a transitional cells are phenotypically distinct and are lacking in mice deficient in I κ BNS. *Proc Natl Acad Sci U S A* **111**, E4119-26 (2014).

18. Kratchmarov, R., Magun, A. M. & Reiner, S. L. TCF1 expression marks self-renewing human CD8+ T cells. *Blood Adv* **2**, 1685–1690 (2018).
19. Kim, S. *et al.* The Distinct Role of Tcfs and Lef1 in the Self-Renewal or Differentiation of Mouse Embryonic Stem Cells. *Int J Stem Cells* **13**, 192–201 (2020).
20. Yu, S. *et al.* Hematopoietic and Leukemic Stem Cells Have Distinct Dependence on Tcf1 and Lef1 Transcription Factors. *J Biol Chem* **291**, 11148–60 (2016).
21. Perez-Andres, M. *et al.* The nature of circulating CD27+CD43+ B cells. *J Exp Med* **208**, 2565–6 (2011).

Referees' comments:

Referee #1 (Remarks to the Author):

I wished the authors would be a little more careful then stating "we identified...human B-1 cells" in the abstract. I understand its to get published in Nature, but the phenotype of their human cells have been reported previously to be B-1 just to be shown wrong. As I stated before these cells (when sorted) are heterogeneous, not just with regards to Ig subset, I am well aware of the ability of B-1 cells to class switch.

A more careful-statement, such as B-1 like cells does not distract from their excellent work, but given how few people work on B-1, the authors statements suggest an absoluteness that I do not feel is warranted, perhaps also reflected in the many concerns of Reviewer #3.

The body of work provided in this manuscript adds substantially to our understanding of B cell subsets in mice and humans and I have no further comments.

We have taken on board this important comment from the referee and have now been much more cautious in both the abstract and the text.

In the abstract, we now simply say:

"These transcription factors [TCF1/LEF1] are also expressed in human chronic lymphocytic leukaemia (CLL) B cells and in a B-1-like population that is abundant in pleural fluid and circulation of some patients with pleural infection".

Throughout the manuscript, we no longer refer to human B1-enriched cells, but more cautiously, we describe "B-1-like" cells.

We have stained additional healthy donor PBMCs and besides CD19, CD38, CD43, CD5, TCF1 and LEF1, we have included markers such as CD27, CD24, IgM and IgA that now better describe the heterogeneity of the CD43+CD5+ population as described by the reviewer (Fig. 1h and Extended Fig. 1j). The results confirm the trends we described of higher PtC, TCF1 and LEF1 amongst CD43+ CD5+ cells, which have now been quantified and statistics have been added.

Referee #2 (Remarks to the Author):

The authors have fully addressed my questions and concerns during this past revision. I have no additional questions and believe this study greatly expands the mechanism of B1-a cell development and function and is suitable for publication.

Referee #3 (Remarks to the Author):

I appreciate the author's efforts to address my suggestions and concerns. Here are additional comments/clarifications regarding some of the author's responses to my previous review. I divided my review into three parts: A) Mouse B-1 cells, B) Technical/data analysis, and C) Human B-1 cells.

A) Regarding Mouse B-1 cells:

1. The authors have shown compelling B-1a phenotype and function on the double TCF1/LEF1-KO (DKO) mice/cells. I also appreciate the authors' careful searching of the literature for other transcription factors (TFs) and molecules that affect B-1a development, as I suggested during my previous review. Based on how the authors addressed my comments, I find it helpful to clarify that every experimental model has caveats. My point in bringing up previous studies on B-1a development is not to undermine the extensive and important work from the authors here, but to bring to their attention possible mechanistic explanations for the effects they observe. It is also to highlight that other genes show a strong effect on B-1a development and repertoire that are desirable to consider when building a working model based on how their new data fits into the current paradigm/literature. Clarifying this, I believe, is relevant and helpful to the field and the readers. See below for some clarifications on the points I made on my initial review and additional comments.

Thank you for clarifying this.

2. My comments on the role of other TFs/genes in B-cell biology are focused on the B-1a lineage, which is known to represent a separate B-cell lineage that emerges from separate B-cell progenitors. As the authors mentioned, Arid3a deficiency leads to >80% reduction in B-1a in the peritoneal cavity. In another paper, Yan Zhou et al. (Lin28b promotes fetal B lymphopoiesis through the transcription factor Arid3a, JEM, 2015), they manipulated Lin28 and Arid3a expression in fetal vs adult B-cell progenitors in adoptive transfer experiments to confirm a robust decrease in CD5+ B-1a cells. These adoptive transfer experiments can be compared to similar experiments performed here, in Fig. 1n. Similarly, the NFATc1 study mentioned by the authors uses adoptive transfers to show 9-fold decrease in B-1a cells and >10-fold decrease in anti-PtC binders (Figs 2A,B,C). I compared that to the adoptive transfer experiments performed by the authors here in their Fig. 1n, showing ~10% B-1a reconstitution from WT versus ~4% B-1a in the TCF1/LEF1 DKO. Other studies also performed conditional NFATc-1-KO using Vav or CD19 cre models (Märklin et al., 2020). In the case of Bhlhe41 deficiency, in addition to >70% decrease in B-1a, the study shows a complete absence of a key B-1 repertoire (i.e., lack of VH12). And there are the studies related to BCR/Ig/CD19 signaling, including work from Thomas Tedder's and Rajewsky's groups showing the role of CD19 in B-1a development, where CD19^{-/-} lacks B-1a-derived natural antibodies and defects in B-1a development. An attempt to integrate these findings into TCF1/LEF1 would be helpful to the field. As I suggested in my last review, I believe some of the results observed by the authors here could be explained/linked to previous studies above—this is not a negative criticism.

We thank the reviewer for their comments. We believe we have made a balanced attempt to describe our findings in the context of previous work. Nevertheless, we are being cautious when comparing the effects of different molecules on B-1 cell development and natural antibodies, particularly in the absence of a fate-mapping system to establish whether circulating antibodies are of B-1 or B-2 origin. For example, CD19 is not only required for B-1 cell development; it is also required for B-2 BCR signalling and survival. The greatest antibody isotype that is reduced in unimmunised CD19^{-/-} mice is IgG1 – which is not thought to be of B-1 origin- followed by IgM (PMID:8566028), with no defect in IgA (PMID: 10940875) - despite some IgA being thought to be of B-1a cell origin (PMID:26062045).

3. The other misunderstanding that needs to be clarified is around Lin28b. Lin28b is a fetal-exclusive RNA-binding protein, which, in adults, is repressed by let-7 expression. Thus, Lin28b is NOT expressed in adults. The reason the authors did not find a publication on this is because it is known that Lin28b-KO in adults is uninformative, given that this gene is not expressed (instead, there is high expression of let-7, which, if removed, can promote B-1a development). Together, Lin28b, Arid3a, IGF2BP3, and others are required for B-1a development in early life, not adults. Thus, the conclusion by the authors that Lin28b expression is the same

among all B-cell subsets based on the ImmGen App, is not quite correct. The problem is that all the values in the ImmGen data are likely to represent negative expression since Lin28b is not expressed in adult cells.

We agree with the reviewer: we also show in Fig. 3f that Lin28b expression is restricted to fetal liver progenitor cells. We just wanted to point out that mice lacking Lin-28b can have a completely normal B-1 pool (for the reasons the reviewer suggests), whereas mice lacking TCF1 and LEF1 do not have a replete B-1 pool. Thus, we ask the reviewer to acknowledge that the effects of these transcription factors can be substantial.

4. I used the word “moderate” based on the data shown in Fig. 1, which I addressed in my previous review, comment #2. For example, in Fig. 1i there are changes in the percentages of B-1 cells across WT, LEF1-KO, and DKO that appear moderate—this doesn't mean they are insignificant. But deserve attention given that they are based on total live cells, and not total CD19+ B cells (Fig. 1 j-l). For example, when B-1a decreases but B-2 does not increase, what other populations can increase (in only some mice) that affect the direct ratio of B-1/B-2? Myeloid cells?

The referee wonders whether a relative decrease of B-1 cells when plotted as a percentage of live cells could simply be explained by increase in another population such as myeloid cells. The reviewer might have overlooked that we included total numbers in Extended Fig. 2a. These data show a clear reduction in total B-1 cells and B-1a cells.

5. The point I was trying to make in my comment #3 from the previous review is that TCF1/LEF1 does NOT seem to affect B-1a repertoire similar to how Bhlhe41 deficiency (or H-2DM deficiency) does. Instead, it decreases B-1a numbers. Hence, I would not emphasize changes in the repertoire of DKO mice if the reason for such a change is already known to be due to a decrease in B-1a. Thank you for addressing this.

We are pleased the reviewer is satisfied with our answer.

6. Based on my previous comments regarding prior literature, do the authors think that TCF1/LEF1 play a role in Ig/BCR, given that DKO decreases BCR expression on all B-cell subsets, not only B-1a? Could that be part of the mechanism of B-1a effects observed here?

The reviewer is right in that we cannot exclude a role for TCF1/LEF1 in B-2 cells. We agree that there appears to be a trend – although not statistically significant – of reduced Igk/l expression in B-2 cells as well. We had discussed a possible role of these TFs in memory B-2 cells in an earlier submission and have reintroduced it in the discussion of this version:

While TCF-1 and LEF1 deficiency had more pronounced effects on B-1 cells, it is possible that these transcription factors also contribute to the homeostasis of long-lived B-2 cell subsets. We observed a trend toward reduced B-2 cell turnover and diminished surface BCR expression in the absence of TCF-1 and LEF1. This effect may be particularly relevant for B-2 cells that rely on self-renewal for their

maintenance, such as memory B cells, which exhibit the highest expression levels of both transcription factors among B-2 cell populations.

7. Based on the flow plots, there seems to be more than 50% decrease in the level of CD5 protein on the surface of B-1a cells. Do the authors think the decrease in IL-10 observed here is due to a significant reduction in CD5 expression on B-1a?

We had already quantified the MFI of CD5 protein on B-1a cells in Extended Figure 2c, comparing 7 TCF1/LEF1 KO mice (median = 322) and 5 littermate wild type controls (median = 435). This constitutes a 26% reduction. We had already described that CD5 expression promotes IL-10 production and provided the reference (50), but we have made this even more clear in this version by saying:

“A positive correlation, albeit weak, was also seen with CD5 expression, consistent with the known role for CD5 in promoting IL-10 production⁵⁰, and the observed reduction in CD5 expression in B-1a cells lacking TCF1/LEF1 (Fig. 4i-j).”

8. B-1 progenitors exist in the spleen before CD5 expression, which is turned on during the transitional stage before maturation. My original question was about precursors to B-1a cells, either CD5+ transitional B-1a (as the authors analyzed) or CD5-neg B-1a-specific progenitors, as I described. It would be interesting to look at their data to investigate the effect on the previously described splenic B-1 progenitor. However, this is not necessary to validate the author’s claims here.

We thank reviewer for the interesting suggestion. We used gating strategy described in PMID:21282663 and found B-1 progenitors (B-1P) in spleen were modestly increased in 9-day-old DKO mice compared to WT. Unfortunately, we did not do these stains in adult mice, whose bone marrow B-1P were comparable between WT and DKO mice. This will be interesting follow up work.

B) Regarding Technical/data analysis:

1. The authors mention that using either log or bioexponential scales is “perfectly acceptable” to display digital flow data plotted in various figures. For example, in Fig. 1b, TCF1 and LEF1 seem each plotted on different scales, including on the bi-axial plot showing both proteins on different scales on the same plot. Log scale can generate artifacts in the lower end of the plot and has been replaced by the bio-exponential scale, which was developed to address this problem with the log scale when displaying cells with low expression values. Please, see the paper “Interpreting flow cytometry data: a guide for the perplexed” (Nature Immunol, 2006). It is good flow practice to use bio-exponential for fluorescent antibodies when analyzing markers that are not highly expressed to avoid artifacts of log. There are exceptions

to this rule, but they don't seem to apply to this case. Linear scale for scatter (FSC) plots are fine and often recommended.

We thank the reviewer for this explanation – we modified the relevant axes in our previous submission following their advice.

2. Re: BrDU, the new display using “dot plots” further supports my original comments. There are very few cells (or events) on the Ly5a-neg cells. Thus, the few dots above the “quadrant gates” could lead to the formation of contour plots above the quadrant line, even though they might not be biologically different from the main population. This difference can be appreciated when comparing the BrDU+ cells from the Ly5a+ population. There, the BrDU+ cells represent a separate population, while the Ly5a-neg cells have a global shift upwards of the entire population and may not represent two groups of BrDU+ and BrDU-. Unfortunately, the few events make it difficult to conclude whether you have a separate population of BrDU+ or whether those few dots are simply the distribution around the main population—this is the limitation of the assay. The authors' interpretation, however, is based on the quadrant gates showing percentages of 1.57 vs 0.11. These percentages can be skewed based on the other 3 populations that make up the quadrant gates, which when increased or decreased, affect the percentage of cells on that top left quadrant. Again, my comments are intended to improve clarity and reproducibility for the field/reader. Please see “Interpreting flow cytometry data: a guide for the perplexed” (Nature Immunol, 2006) for more details on the caveats for using quadrant gating and making gating decisions on a few events.

We agree with the reviewer's point, which we had meant to fix in the previous version. We have now gated the percentage of BrdU+ out of donor cells only (top left quadrant out of total left quadrant only). The updated figure confirms that deficiency in TCF1 and LEF1 leads to a greater reduction in the proportion of donor-derived BrdU-labelled B-1a cells ($P=0.0210$) compared to B-2 cells ($P=0.0948$) (Fig. 2d). As explained in point 6 above, we have discussed that here appears to be a trend of an effect on B-2 cells too:

While TCF-1 and LEF1 deficiency had more pronounced effects on B-1 cells, it is possible that these transcription factors also contribute to the homeostasis of long-lived B-2 cell subsets. We observed a trend toward reduced B-2 cell turnover and diminished surface BCR expression in the absence of TCF-1 and LEF1. This effect may be particularly relevant for B-2 cells that rely on self-renewal for their maintenance, such as memory B cells, which exhibit the highest expression levels of both transcription factors among B-2 cell populations.

C) Regarding Human B-1 cells:

1. I appreciate all the human data presented here—they are indeed interesting. However, although possible, it seems unlikely that some adult humans would have >60% of B-1 circulating in blood. It is possible that the authors may be correct about the human B-1 classification in adult tissues and the diseases described. However, the proof is not contained within the current data, which relies on a set of surface markers. While I truly appreciate the efforts in classifying these cells in humans (much needed in the field!), I would be cautious about how to describe this classification to leave room for future studies that may provide more definitive evidence based on the developmental origin of these cells. Thus, personally, I wouldn't state that Human B-1 cells in blood/pleural were identified "with confidence".

We have taken on board this important comment from the referees and have now been much more cautious in both the abstract and the text.

In the abstract, we now refrain from "identifying" human B-1-enriched cells, and simply describe TCF1 and LEF1 being found expressed in pleural fluid B-1-like cells:

"These transcription factors [TCF1/LEF1] are also expressed in human chronic lymphocytic leukaemia (CLL) B cells and in a B-1-like population that is abundant in pleural fluid and circulation of some patients with pleural infection".

Throughout the manuscript, we no longer refer to human B1-enriched cells, but more cautiously, we describe "B-1-like" cells.

We have stained additional healthy donor PBMCs and besides CD19, CD38, CD43, CD5, TCF1 and LEF1, we have included markers such as CD27, CD24, IgM and IgA that now better describe the heterogeneity of the CD43+CD5+ population (Fig. 1h and Extended Fig. 1j). The results confirm the trends we described of higher PtC, TCF1 and LEF1 amongst CD43+ CD5+ cells, which have now been quantified and statistics have been added.

2. The authors argument on the lack of CD38 expression is reasonable. While I agree that CD38 is generally expressed on activated, transitional, or plasma cells, the kinetics of CD38 can be complex. This surface protein is expressed during B-cell development, it is turned off during final maturation, and is only later re-expressed on plasma cells. Is it possible that blood cells expressing TCF1/LEF1 are activated cells that have been blocked from plasma cell development? For example, would TLR activation lead to IL-10 secretion while a fraction of these cells remain CD38 negative?

We thank reviewer for raising a very interesting point. We agree that CD38 – as well as CD5 and CD43 - expression patterns are complex, and further work is needed to understand the significance of their regulation and how it they may be modulated during B cell activation with different stimuli. At present, this is difficult to test in humans due to the impossibility of fate mapping in-vivo. We have nevertheless added a couple of sentences in the discussion alluding to the possibility raised by the reviewer:

Given the observed gradation in PtC reactivity across CD43⁺CD5⁺, CD43⁻CD5⁺, and CD43⁻CD5⁻ circulating populations in healthy donors — and the fact that a fraction of CD38⁺ cells also exhibit PtC reactivity — it is likely that surface markers such as CD5, CD43, and CD38 change dynamically as B-1-like cells mature. Similarly, an interrelationship may exist between CD38⁺ and CD38⁻ B-1-like cells, with CD38 expression changing as these cells mature and undergo activation.